



# Tropical tropospheric ozone and carbon monoxide distributions: characteristics, origins and control factors, as seen by IAGOS and IASI

Maria Tsivlidou[1], Bastien Sauvage[1,*], Brice Barret[1,*], Pawel Wolff[2], Hannah Clark[3], Yasmine Bennouna[1], Romain Blot[1], Damien Boulanger[2], Philippe Nédélec[1], Eric Le Flochmoën[1], and Valérie Thouret[1]

[1]Laboratoire d'Aérologie (LAERO), Université Toulouse III – Paul Sabatier, CNRS, Toulouse, France
[2]Observatoire Midi-Pyrénées (OMP-SEDOO), Université Toulouse III - Paul Sabatier, CNRS, Toulouse, France
[3]IAGOS-AISBL, 98 Rue du Trône, Brussels, Belgium
[*]These authors contributed equally to this work.

**Correspondence:** Maria Tsivlidou (maria.tsivlidou@gmail.com)

**Abstract.** The characteristics and seasonal variability of the tropical tropospheric distributions of ozone ($O_3$) and carbon monoxide (CO) were analysed based on in situ measurements provided by the In-service Aircraft for a Global Observing System (IAGOS) program since 1994 and 2002 respectively, combined with observations from the Infrared Atmospheric Sounding (IASI) instrument on board the Met-op A satellite since 2008. The SOFT-IO model, which couples back trajectories with CO emissions inventories, was used to explore the origins and sources of the tropical CO observed by IAGOS. The highest tropical $O_3$ and CO maxima occur over Northern Hemisphere (NH) Africa in the low troposphere (LT) (80 ppb and 850 ppb respectively at 2.5 km over Lagos) during the dry season (January). Despite the active local fires, local anthropogenic (AN) emissions (60 %) are dominant for the CO, and consequently the $O_3$ maxima. The importance of the local AN emissions are highlighted over Central Africa, as they cause a persistent polluted surface layer during the transition seasons (40 % in October and 86 % in April). The second highest $O_3$ and CO maxima are observed over Asia. Local or regional Asian AN emissions cause the CO maximum in the LT (0.5 km) in January, and the $O_3$ maximum in the free troposphere (at 6 km) in the post-monsoon season (April). South China is the only Asian site where $O_3$ peaks in the LT (75 ppb at 2.5 km), due to local fires (30 %) in addition to the local (52 %) and regional (15 %) AN emissions. The highest amount of transported CO in the tropics originates from Africa. The main transport pathway is from the dry-season African regions towards the wet-season ones. Contributions from the NH Africa are found over Arabia and Eastern Africa (up to 70 %), and India (40 % in the mid (MT) and 60% in the upper (UT) troposphere) during the dry season. Transport towards NH South America is found all year long, with significant contributions in the MT and UT (30–40 % over Caracas on annual basis). In contrast, the impact of the Asian emissions in the LT and MT is limited on a local or regional scale. Export of polluted Asian air masses is important in the UT during the Asian summer monsoon and post-monsoon seasons, when convection is active. The AN Asian contributions are mostly found over Arabia and Eastern Africa (up to 80 %) during the Asian summer monsoon. During the post-monsoon, CO impacted by the Indonesian fires (resp. SouthEast Asian AN emissions) are transported towards Eastern Africa (64% and 16%) due to the Tropical Easterly Jet. The lowest $O_3$ and CO levels are observed over South America, due to less strong local





emissions in comparison to Asia and Africa. The only important CO and $O_3$ enhancement is observed in the MT during the local fires (October), when $O_3$ and precursors impacted by the local AN and fire emissions are trapped in an anticyclone and

transported towards South Africa (5–10 ppb from SH and NH South America respectively).

# 1 Introduction

Tropospheric ozone ($O_3$) and carbon monoxide (CO) are key components in the atmosphere. $O_3$ has a significant impact on human health close to the surface (Curtis et al., 2006; Jerrett et al., 2009) and on climate by being a powerful greenhouse gas

(Gauss et al., 2003; IPCC, 2021). $O_3$ is a secondary pollutant produced by photochemical oxidation of precursors such as CO and volatile organic compounds (VOCs) in the presence of nitrogen oxides ($NO_x$) (Logan et al., 1981). Its distribution is controlled by: stratospheric transport (Stevenson et al., 2013); transport processes at intercontinental and hemispheric scale (Wild et al., 2004); emissions of precursors (natural and anthropogenic) and destruction processes (photochemical and depositional) (Monks et al., 2015). Due to its longer lifetime, CO is considered a powerful pollution tracer of combustion products at a

hemispheric level (Edwards et al., 2006). CO impacts the oxidation capacity of the atmosphere by being the major sink of OH radicals in non polluted atmosphere (Lelieveld et al., 2016), and the climate by producing greenhouse gases, such as $CO_2$ and $O_3$, during its oxidation (Myhre et al., 2013). The primary (resp. secondary) sources of CO include anthropogenic and biomass burning emissions (resp. oxidation of VOCs).

Recent studies (Gaudel et al., 2018, 2020; Zhang et al., 2016) have shown increasing tropospheric $O_3$ burden in the second

half of the 20th century mostly due to increase of precursors in the tropical regions. However, the global $O_3$ distribution and sources of precursors remain uncertain due to inadequate observations in the remote free troposphere, especially over developing countries in the tropics (Gaudel et al., 2018; Tarasick et al., 2019).

The tropical region is of particular interest regarding tropospheric $O_3$ and CO. It combines: i) intense photochemistry due to high UV radiation and humidity, ii) large active natural sources of CO and other $O_3$ precursors through biomass burning

(Ziemke et al., 2009), biogenic (Aghedo et al., 2007) and lighting emissions (Sauvage et al., 2007b, c), iii) increasing anthropogenic emission due to rapid industrialisation (Granier et al., 2011; Duncan et al., 2016), iv) large ozone net production potential because deep convection can transport surface emissions to higher altitudes, where their lifetime is increased due to lack of surface deposition and dilution with unpolluted background (Pickering et al., 1995) and v) dynamic processes capable of redistributing chemical species in a regional and global scale (Zhang et al., 2016). Thus, the tropics are a region where $O_3$

production is favoured.

Satellite observations from the OMI and MLS sensors (Ziemke et al., 2019) and simulations from the GEOS-Chem chemical transport model (Zhang et al., 2016, 2021) display the highest $O_3$ burden increase in the tropical region - mostly over India, East Asia and SouthEast Asia. Most studies tend to confirm an increase of $O_3$ in the tropics but they are mostly based on model





simulations, sparse ground observations or satellite data with little consistency, and it is not clear what can cause such $O_3$

increase. Indeed, the trends are attributed to different factors such as biomass burning (Heue et al., 2016), dynamics (Lu et al., 2019; Thompson et al., 2021) or anthropogenic (Zhang et al., 2016; Gaudel et al., 2020). Thus, further investigation based on *in situ* observations is required in order to better constraint models and satellite retrievals, and reduce the uncertainties in the quantification of $O_3$ and CO trends and source attribution over the tropics.

Measurements of tropical $O_3$ and CO are available by satellite observations, but they have a coarse vertical resolution (e.g.

Barret et al., 2008; Thompson et al., 2001). Several field campaigns have been carried out in the tropics. However, they provide sparse measurements in terms of temporal and spatial coverage. The Southern Hemisphere ADditional OZone Sounding (SHADOZ) program (Thompson et al., 2003a) provides long-term $O_3$ observations over the tropics using ozonesondes since 1998. Even though these measurements offered a better understanding on vertical distribution of tropical $O_3$, they are mostly limited to remote observing sites such as Ascension and Reunion island, and they under-represent the tropical upper tropo-

sphere. In addition, it is difficult to provide additional constraints regarding the relation between $O_3$ and CO in the tropics, due to a lack of simultaneous CO *in situ* observations.

The IAGOS (In-service Aircraft for a Global Observing System; (Marenco et al., 1998; Petzold et al., 2015)(Thouret et al., 2022) program has provided continuous and consistent $O_3$ (Thouret et al., 1998; Blot et al., 2021) and CO (Nédélec et al., 2015) observations over the tropics for the last 26 and 18 years respectively. It measures vertical profiles over remote (e.g.

Madras) and megacities (e.g. Lagos, Hong Kong), along with the lower part of the upper tropical troposphere. Previous studies have documented the tropical composition over Africa (Sauvage et al., 2005, 2007a, d; Lannuque et al., 2021), South America (Yamasoe et al., 2015) and South Asia (Sahu et al., 2014; Sheel et al., 2014). However, they are focused on specific regions of the tropics and have limited temporal coverage, especially for CO as fewer measurements were available at this time. Thus, the $O_3$ and CO distributions and their interlocking in the entire tropics are still not well documented.

The SOFT-IO model (Sauvage et al., 2017) has been developed to supplement the analysis of the IAGOS dataset by estimating anthropogenic (AN) and biomass burning (BB) contributions to the observed CO measurements. These measurements, along with the SOFT-IO output allow us to trace the CO origin, and establish connections with $O_3$ origin over the tropics. Further, global distributions provided by Infrared Atmospheric Sounding Interferometer (IASI)-Software for a Fast Retrieval of IASI Data (SOFRID) (Barret et al., 2011; De Wachter et al., 2012) retrievals since 2008 complement the $O_3$ and CO distri-

butions provided by IAGOS. They allow us to understand the spatial extent of pollution plumes, and explore intercontinental transport patterns.

In this article we take advantage of the unique IAGOS database to (i) document the characteristics and seasonal variability of these two atmospheric species over the whole tropical band for the last decade for the first time, (ii) explore the origin of the observed CO anomalies, (iii) investigate transport processes driving the CO and $O_3$ distribution in the tropics.

The observational (IAGOS and IASI) and model based (SOFT-IO) datasets, and methodology are introduced in Sect. 2. In Section 3, the IAGOS observations are analysed to document $O_3$/CO vertical profiles, along with the UT composition over the tropics. In addition, the sources of observed CO are explored with SOFT-IO.



## 2 Data and Methods

### 2.1 IAGOS observations

The Research Infrastructure IAGOS (Petzold et al., 2015; Thouret et al., 2022) provides *in situ* measurements of trace gases ($O_3$, CO, water vapour, $NO_y$ between 2001 and 2005 (e.g. Gressent et al., 2014), and more recently $NO_x$, $CH_4$, $CO_2$ and cloud particles, see https://www.iagos.org/iagos-data/) and meteorological parameters (temperature and winds), using equipped commercial aircraft. Full description of the instruments can be found in Nédélec et al. (2015). $O_3$ (resp. CO) is measured using a dual-beam ultraviolet absorption monitor (infrared analyser) with an accuracy of 2 ppb (resp. 5 ppb), a precision of 2 %

(resp. 5 %) and a time resolution of 4 (resp. 30) seconds (Thouret et al., 1998; Nedelec et al., 2003). IAGOS measures vertical profiles during ascend and descend phases, and the upper troposphere (between 9 and 12 km; 300–185 hPa) during cruise phases. Considering the aircraft speed (7–8 m s$^{-1}$ during ascent/descent; 900 km h$^{-1}$ during cruise), the time resolution of the instruments corresponds to a vertical resolution of 30 m (resp. 225 m), and a horizontal resolution of 1 km (resp. 7.5 km) for $O_3$ (resp. CO).

$O_3$ (resp. CO) observations have been collected since 1994 (resp. 2002) in the frame of the IAGOS Research Infrastructure and its predecessor MOZAIC (Marenco et al., 1998) program, based on the same instrument technologies. Good consistency in the measurements between the two programs (hereafter referred to as IAGOS) (Nédélec et al., 2015; Blot et al., 2021) leads to IAGOS temporal coverage of 26 (resp. almost 20) years for $O_3$ (resp. CO). IAGOS data provides robust $O_3$ and CO climatologies, allowing studies of long-term trends (e.g. Cohen et al., 2018) along with validation of chemistry transport

models (e.g. Sauvage et al., 2007b; Gressent et al., 2016) and satellite data retrievals (e.g. De Wachter et al., 2012) on a global scale. To complement the IAGOS observations, we use the potential vorticity (PV) field, which is part of the ancillary data (https://doi.org/10.25326/3) from the IAGOS database. The PV is calculated from the European Centre for Medium-Range Weather Forecast's (ECMWF) operational fields (horizontal resolution 1°, time resolution 3 hours), interpolated along IAGOS trajectories.

### 2.1.1 Data treatment

The tropical zone can be defined in several ways, such as by meteorological characteristics (e.g. location of the subtropical jets), climatic elements (e.g. precipitation rates) or by the geographical extent. Following the latter way, in the Tropospheric Ozone Assessment Report, Phase II (TOAR-II; https://igacproject.org/activities/TOAR/TOAR-II) Ozone and Precursors in the Tropics working group, the tropics are defined between 20° S and 20° N. In our study, we consider the extended area between

25° S and 25° N, in order to investigate interactions of pollution and the transport of air masses between the tropics and the subtropics. IAGOS observations are used to document $O_3$/CO vertical profiles and the (lower part of) UT . Only tropospheric measurements are taken into account, by applying a PV filter of 2 PV units (pvu) for each measurement during cruise phase, and for the measurements between 20–25° N/S during ascend/descend. The UT climatologies are derived by averaging the cruise data (300–185 hPa) on a 2 x 2.5° grid, for the period 1994–2020 (resp. 2002–2020) for $O_3$ (resp. CO). For the same

time periods, the climatologies over the vertical are derived by averaging the data into 10 hPa pressure bins from the surface





up to the upper limit of the profile. The upper limit is based on a distance criteria of a 300-km radius around the IAGOS observational site, similar to Petetin et al. (2016). This way we reduce uncertainties due to possible horizontal heterogeneity in the measurements, as the aircraft keeps moving in the horizontal plane during ascent and descent. The cut-off radius is set in alignment to a coarser global model resolution, in order to limit the potential heterogeneity inside a single grid box.

To determine a reliable climatological profile, we need to assess the statistical significance of the data. Similar to Logan (1999) and Sauvage et al. (2005), we compute the relative standard error (RSE) of the $O_3$ (CO) monthly mean, versus the number of flights per month. The RSE is defined as the fraction between the standard error ($SE = \frac{\sigma}{\sqrt{N}}$, with $\sigma$ the square root of the sample variance and $N$ the number of flights) and the $O_3$ (CO) monthly mean. The minimum number of flights required for statistical significance corresponds to the number above which RSE $\leq$ 10 %. We choose RSE less than 10 %,

because RSE depends not only on the number of measurements, but also on the $O_3$ (CO) variability which is high over the tropics (Thompson et al., 2003b). For each site with an adequate number of flights per month, we consider an individual profile of $O_3$ (CO). Otherwise, we combine sites in clusters (see Table 1), in order to increase the amount of data and get significant climatological profiles. Besides, the clusters can be useful for validation of models with a coarse horizontal resolution, because they represent a wider area as resolved by the models, which are not expected to capture small-scale variations in the ozone

field (e.g. Emmons et al., 2010). For clustering, the sites should be: i) in relatively close distance from each other, ii) governed by similar meteorological conditions, and iii) display similar characteristics in the vertical distribution of $O_3$ (CO) (see Sect. 3.2).

The meteorological conditions in the tropics are peculiar, with different seasonal patterns depending on the region. For instance, in Africa the main seasons are two (dry and wet) with two intermediate seasons passing from wet to dry and vice

versa (Sauvage et al., 2005; Lannuque et al., 2021). On the other hand, in Asia the seasons are defined by the Asian monsoon phases: Asian summer monsoon (wet season); Asian winter monsoon (dry season) and post monsoon. Thus, we considered it more appropriate for our analysis to deviate from the classical definition of the seasons, which fits better to studies concerning higher latitudes. Instead, we analyse the $O_3$/CO profiles and horizontal distributions over intermediate months of the tropical seasons (January, April, July and October), to highlight seasonal patterns.

## 2.2 SOFT-IO model


The SOFT-IO (SOft attribution using FlexparT and carbon monoxide emission inventories for In-situ Observation database) tool (Sauvage et al., 2017; http://dx.doi.org/10.25326/2) has been developed to investigate the origin of the observed IAGOS-CO, by coupling FLEXPART 20-days backward transport simulations with emission inventories. For each point of IAGOS trajectory, SOFT-IO estimates the CO contribution coming from 14 different geographical regions (see Fig. 1), for AN and BB

origin separately. We use Community Emissions Data System (CEDS2) AN emissions (McDuffie et al., 2020) and the Global Fire Assimilation System (GFAS) BB emissions (Kaiser et al., 2012) which include fire injection heights, to discriminate sources of CO anomalies over different regions of interest. For the calculations, the AN (resp. BB) emissions are updated on a monthly (resp. daily) basis.



**Table 1.** Description of individual sites and clusters used in this study. The location of the sites is displayed in Fig. 1

|  | Individual sites/Clusters | IAGOS sites |
|---|---|---|
| South America | South Brazil | Rio de Janeiro (Brazil), São Paulo (Brazil) |
|  | Caracas | Caracas (Venezuela) |
|  | Bogota | Bogota (Colombia) |
| NH Africa | Lagos | Lagos (Nigeria) |
|  | Sahel | Abuja (Nigeria), Ouagadougou (Uganda), Niamey (Niger) |
|  | Gulf of Guinea | Lome (Togo), Yaounde (Cameroon), Douala (Cameroon), |
|  |  | Libreville (Gabon), Accra (Ghana), Cotonou (Benin), Port Harcourt (Nigeria), |
|  |  | Abidjan (Ivory Coast), Malabo (Equatorial Guinea) |
| SH Africa | Central Africa | Luanda (Angola), Kinshasa (Democratic Republic of Congo), Brazzaville (Congo) |
|  | Windhoek | Windhoek (Namibia) |
| Arabia and Eastern Africa (AEA) | AbuDhabi | Abu Dhabi (United Arab Emirates), Muscat (Oman) |
|  | Khartoum | Khartoum (Sudan) |
|  | Addis Ababa | Addis Ababa (Ethiopia) |
|  | Jeddah | Jeddah (Saudi Arabia) |
| Asia | South China | Hong Kong (China), Guangzhou (China), Xiamen (China) |
|  | Gulf of Thailand | Kuala Lumpur (Malaysia), Singapore ( Singapore) |
|  | Madras | Madras (India) |
|  | Hyderabad | Hyderabad (India) |
|  | Mumbai | Mumbai (India) |
|  | Manila | Manila (Philippines) |
|  | Bangkok | Bangkok (Thailand) |
|  | Ho Chi Minh City | Ho Chi Minh City (Vietnam) |

SOFT-IO estimates the contribution to CO anomalies emitted by primary sources during the last 20 days, while it does not calculate the background CO. The background CO can be emitted by primary sources older than 20 days, and by secondary sources such as oxidation of methane and non-methane volatile organic compounds. The meteorological fields are based on 1° x 1° ECMWF analysis and forecast with a time resolution of 6 and 3h respectively.

Several studies (e.g. Cussac et al., 2020; Lannuque et al., 2021; Petetin et al., 2018) have used SOFT-IO to track back the sources of CO measured by IAGOS. Sauvage et al. (2017) validated SOFT-IO performance against IAGOS CO observations 160  for several regions and tropospheric levels. Results show SOFT-IO's capability to detect more than 95 % of the observed CO anomalies over most of the regions, without any strong dependence on altitude or region of the CO plume. SOFT-IO captures the intensity of CO anomalies with bias lower than 10–15 ppb for most of the regions and tropospheric levels. The bias is higher in extreme pollution events and might be related to uncertainties in the emissions inventories. In our study, CO anomalies are defined as the positive difference between the observed and the background CO mixing ratio. Background CO mixing ratio





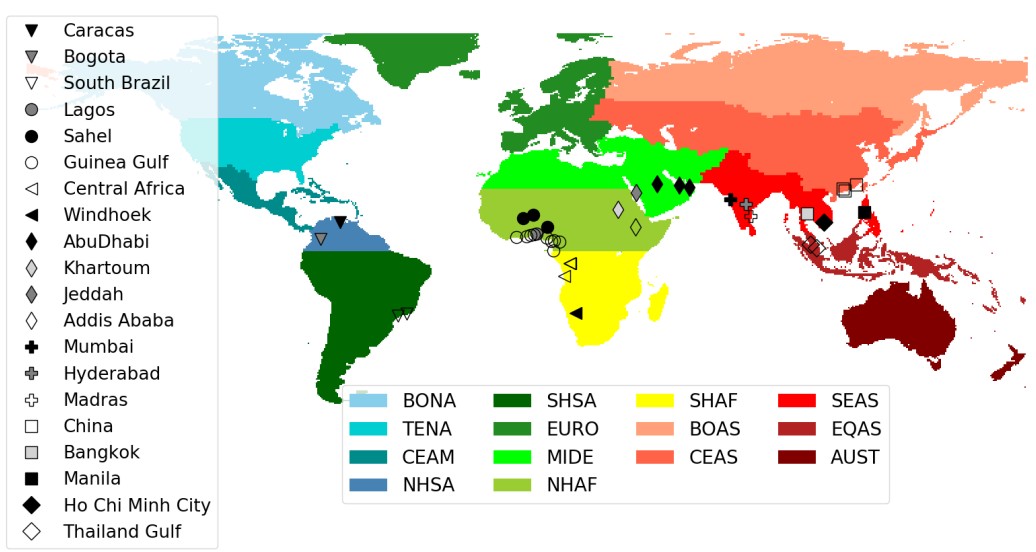

**Figure 1.** Locations of tropical sites served by IAGOS, and geographical source regions used in SOFT-IO model. BONA: Boreal North America; TENA: Temperate North America; CEAM: Central America; NHSA: Northern Hemisphere South America; SHSA: Southern Hemisphere South America; EURO: Europe; MIDE: Middle East; NHAF: Northern Hemisphere Africa; SHAF: Southern Hemisphere Africa; BOAS: Boreal Asia; CEAS: Central Asia; SEAS: Southeast Asia; EQAS: Equatorial Asia; AUST: Australia and New Zealand.

represents a reference value, not affected by surface emission or pollution events. For this reason, it is computed as the monthly climatological median CO of a remote area away from polluted regions, in the upper troposphere (300–185 hPa, during the whole study period 2002–2020).

### 2.3   IASI-SOFRID observations

The IASI sensor onboard MetOp-A (launched in 2006) has a 12 km footprint at nadir and a 2200 km swath allowing an overpass
twice daily at 9:30 and 21:30 local solar time. IASI provides information for the atmospheric composition e.g. content of trace gases such as $O_3$ (Eremenko et al., 2008; Barret et al., 2011; Boynard et al., 2016), CO (George et al., 2009; De Wachter et al., 2012) and $N_2O$ (Barret et al., 2021). We use $O_3$ (v3.5) and CO (v2.1 up to 2014, and v2.2 up to 2019) IASI retrievals performed with SOFRID (Barret et al., 2020; De Wachter et al., 2012).

    SOFRID-$O_3$ v3.5 retrievals use a dynamical a priori profile based on latitude, season and the tropopause height (Barret
et al., 2020). In the tropics, where the surface temperature, thermal contrast and tropopause height are the highest, SOFRID-$O_3$ retrievals allow two independent pieces of information, one in the troposphere and one in the UTLS (Barret et al., 2020). Comparisons with ozonesonde measurements for the period 2007–2017, showed that SOFRID-$O_3$ is biased low in the tropical troposphere and UTLS, by $3 \pm 16$ % and $12 \pm 33$ % respectively in the Northern Tropics (0–30°N), and by $8 \pm 14$ % and





21 ± 30 % in the Southern Tropics (0–30°S) (Barret et al., 2020). Comparisons between SOFRID-$O_3$ retrievals using a single

a priori profile and a dynamical a priori profile showed improvements (e.g a general increase in the correlation coefficients

and the amplitude of the retrieved variability) mostly in the troposphere. The change of the a priori profile leads to minor

differences in the UTLS, indicating the highest sensitivity of IASI to this layer (Barret et al., 2020).

For SOFRID-CO v(2.1 and 2.2 after 2014), two independent pieces of information are provided in the lower (surface–

480 hPa) and upper (480–225 hPa) troposphere (De Wachter et al., 2012). IASI correctly captures the seasonal variability of

CO over southern Africa (Windhoek) and European mid-latitudes (Frankfurt) in lower (resp. upper) troposphere relative to

IAGOS data (resp. correlation coefficients of 0.85 (0.70)). At Windhoek, SOFRID-CO is biased low in the lower (resp. upper)

troposphere by 13 ± 20 % (resp. 4 ± 12 %).

We use monthly averaged SOFRID-CO and $O_3$ retrievals on a 1° x 1° grid from 2008–2019. We focus on pressure levels

corresponding approximately to the independent pieces of information, and on daytime measurements when larger thermal

contrast between the surface and the atmosphere results in increased sensitivity of the instrument (Clerbaux et al., 2009).

# 3 Results

## 3.1 $O_3$ and CO over the Northern and Southern Tropics

Figure 2 displays the tropical lower troposphere (LT) IASI CO (a) and middle troposphere (MT) $O_3$ (b) annual distributions

averaged over the 2008–2019 period. The pressure ranges differ for CO and $O_3$ because they are adjusted to the sensitivity

of the instrument for each compound (see Sect. 2.3). For $O_3$, the stripes along the 10° latitude bands are due to the use of a

dynamical a priori profile (Sect. 2.3), resulting in discontinuities between adjacent latitude bands with different a priori profiles.

Nevertheless, the stripes are a minor issue, as the use of a dynamical a priori profile largely improves the retrieved $O_3$ profiles

in terms of variability and correlation in most latitude bands, relative to the previous version which uses a single a priori profile

(Barret et al., 2020).

Northern Tropical CO and $O_3$ mixing ratios are generally higher than those in the SH (Fig. 2 and Table 2). Geographically,

the largest CO (165 ppb) and $O_3$ (60 ppb) maxima are found over Northern Africa (Table 2), followed by comparable CO and

$O_3$ mixing ratios over Southern Africa and East Asia (145–150 ppb of CO and 50 ppb of $O_3$) (Table 2). The smallest magnitude

of O3 and CO maxima are observed over South America.

**Table 2.** Annual mean LT CO and $O_3$ mixing ratio (in ppb) over the Northern Tropics (0–25° N), Southern Tropics (0–25° S), Northern
Africa (10° W–12.5° E; 0–12.5° N), Southern Africa (10–35° E; 2.5–20° S), East Asia (92.5–110° E; 10–27° N) and South America (35–
50° W; 0–20° S) based on IASI data (Fig. 2).

|  | NH | SH | Northern Africa | Southern Africa | East Asia | South America |
|---|---|---|---|---|---|---|
| CO | 115 | 100 | 165 | 145 | 150 | 110 |
| $O_3$ | 75 | 45 | 60 | 50 | 50 | 45 |





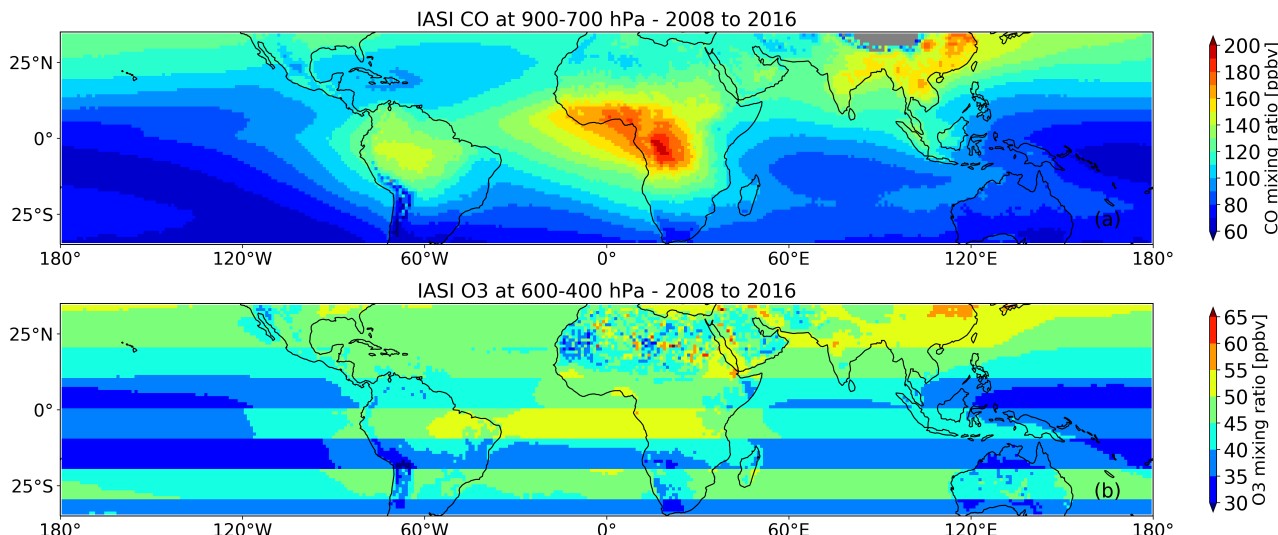

**Figure 2.** IASI LT CO (between 900–700 hPa) (a) and MT O$_3$ (between 600–400 hPa) (b) mixing ratios averaged from 2008 to 2019 on a 1x1 grid.

Figures S1 and S2 show the monthly mean CO flux due to BB and AN emissions respectively. The BB emissions in both hemispheres show strong seasonal variability, as a consequence of the annual shift of the Intertropical Convergence Zone (ITCZ), and the alternance of the rainy and dry seasons. In contrast, AN emissions are located mostly in the NH and have weaker seasonal variability. The differences in CO abundances between the two hemispheres could be related to the spatial distribution of the AN emissions, and the seasonality of the fires. The main emissions of SH CO are the annual dry-season fires, while NH CO is related to the larger population and AN activities (Fig. S2) (Edwards et al., 2004).

The CO, and consequently the O$_3$ maxima over Northern Africa could be related to the strong AN emissions over the Western African coast (Fig. S2a–d). During the dry season, local BB emissions are also active, and could contribute to the CO and O$_3$ maxima (Fig. S1a). High CO and O$_3$ over East Asia are likely related to anthropogenic emissions (Fig. S2a–d), in contrast to the SH regions (Southern Africa and South America) where the local dry-season BB emissions are significantly stronger than AN (Fig. S1c–d).

The persistent African CO and O$_3$ maxima are exported over the South Atlantic (Fig. 2a–b), contributing to the wave-one O$_3$ pattern. The wave-one is associated with systematic high O$_3$ (60–65 ppb) above the South Atlantic and low O$_3$ (30–40 ppb) above western Pacific. Similar asymmetry in the O$_3$ pattern is observed in the NH, but with smaller magnitude because both sides of equatorial Pacific show similarly low O$_3$ abundance, while O$_3$ above the North Atlantic is lower than above the South Atlantic (10–15 ppb difference on average) (Fig. 2b).

In this Section we have shown that the CO and O$_3$ maxima over the Northern and Southern Tropics are related to AN and BB emissions, as well as transport in the tropical troposphere. We will therefore focus on the following questions: what are the contributions of the AN and BB emissions to the O$_3$ and CO abundances in the tropics? Are regional or local emissions





responsible for the $O_3$ and CO observed enhancements? To answer these questions, in the next section we analyse the $O_3$ and CO tropical profiles to quantify their persistent and repetitive characteristics. Using the SOFT-IO tool, we aim to establish

connections between these characteristics and local or regional AN and BB emissions.

## 3.2 Regional characteristics of tropical $O_3$ and CO

In this Section, we focus on the tropical CO and $O_3$ distributions based on IAGOS and IASI data. The combination of the high vertical resolution of IAGOS (Figs. 4, 6, 7, 8 panels 1 and 2), and the high spatial and temporal coverage of IASI 2-D global distributions (Fig. 3 a–l), allows us to investigate in detail data-sparse regions like the tropics, and monitor their atmospheric

composition.

The results shown in Fig. 3 (a–d) motivated our choice in combining IAGOS sites in clusters when it is necessary to increase the number of measurements. The LT CO maxima, like over the Gulf of Guinea, cover a wide area. Thus, cities located close to each other are likely to experience similar air masses. According to the wind maps they are also affected by similar meteorological conditions (Fig. S3).

Figures 4, 6, 7, 8 (panels 1 and 2) display the monthly average vertical distributions of $O_3$ and CO based on IAGOS data, since 1994 and 2002 respectively, for the African, Asian, South American, Arabian and Eastern African clusters. Panels 3 to 5 represent the mean contribution to these CO mixing ratios from AN and BB emissions as estimated by SOFT-IO, with information about their geographical origin (see Sect. 2.2 and Fig. 1). To better understand $O_3$ and CO anomalies, Fig. 5 displays the CO contributions in three tropospheric layers related to different dynamical regimes: lower troposphere below 750

hPa corresponding roughly to planetary boundary layer (PBL); mid troposphere above up to 300 hPa, and upper troposphere above up to 200 hPa corresponding to the beginning of maximum convective detrainment.

### 3.2.1 Africa (NH Africa: Lagos -Nigeria-, Sahel and Gulf of Guinea; SH Africa: Central Africa, Windhoek-Namibia-)

The striking feature of CO and $O_3$ over the NH African clusters (Sahel, Gulf of Guinea and Lagos) is the LT maxima during the dry season (January) (Fig. 4 panels 1–2). CO maximises close to the surface, with larger mixing ratios over Lagos (850

ppb) than Sahel (500 ppb) and Gulf of Guinea (400 ppb), mainly due to local AN emissions (58 % over Lagos and Sahel)(Figs. 4 panel 3a; A1 panel 1a; 5a; A2a), despite the active local fires (Fig. S1a). This is consistent with the increasing AN emissions (Liousse et al., 2014) and decreasing BB area extend (Hickman et al., 2021) over northern Africa. The $O_3$ gradient close to the surface (Fig. 4 panels 1a–1c) is likely related to surface deposition and titration by highly concentrated nitrogen oxide (NO) (Monks, 2005) which is expected along with high CO emissions. The $O_3$ maximum and the elevated CO levels (exceeding

300–500 ppb) at 2.5 km over the three clusters, likely indicate chemically processed air masses where $O_3$ has been produced by precursors. The air masses above the Gulf of Guinea and Sahel are transported from the continent (Fig. S3a) by the north-easterly Harmattan flow (Sauvage et al., 2005). AN emissions are the dominant source of CO at 2.5 km over Lagos and Sahel (Figs. 4 panel 3a and A1 panel 3a). The enhanced $O_3$ and CO are confined in the LT due to the stability of the Harmattan flow and Saharan anticyclone which prevent vertical mixing (Sauvage et al., 2005). The stability is due to a temperature inversion





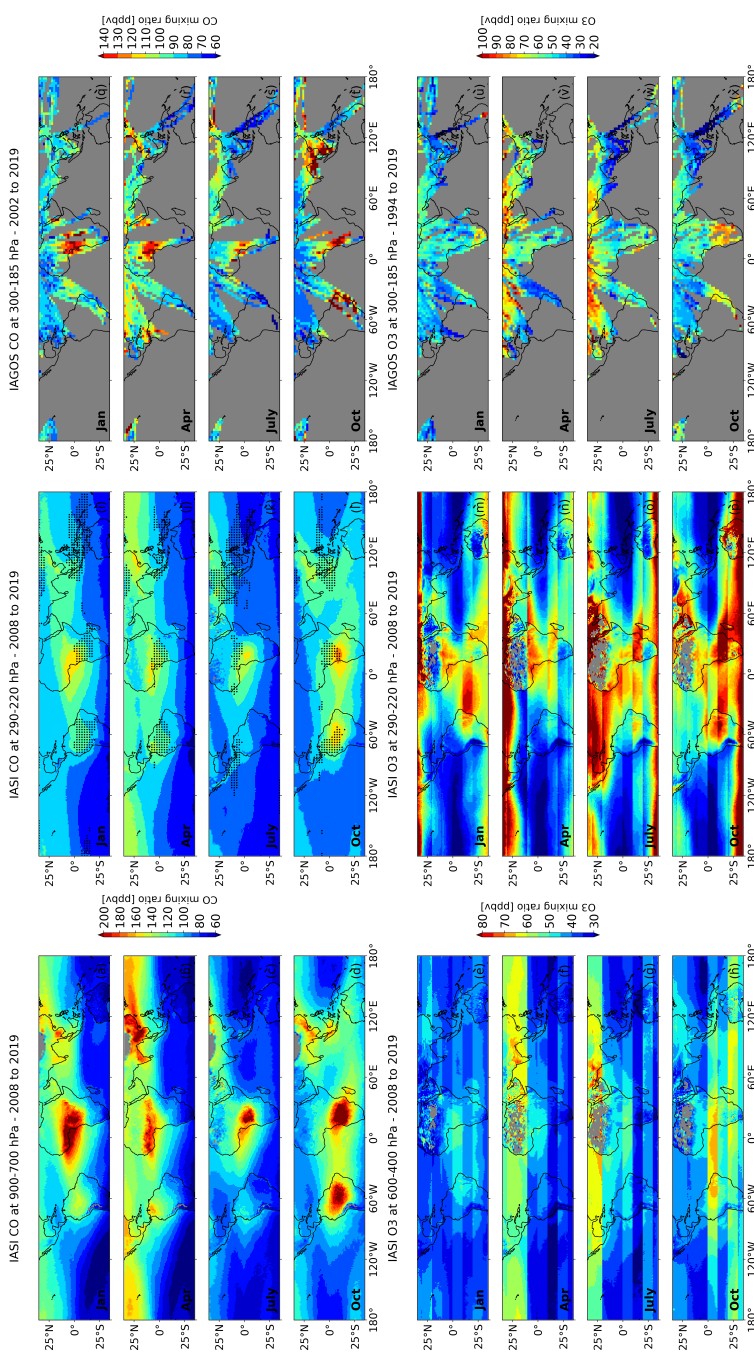

**Figure 3.** Monthly mean LT CO distributions (a–d), MT O$_3$ distributions (e–h), UT CO distributions based on IASI (i–l) and IAGOS (q–t), UT O$_3$ distributions based on IASI (m–p) and IAGOS (u–x).

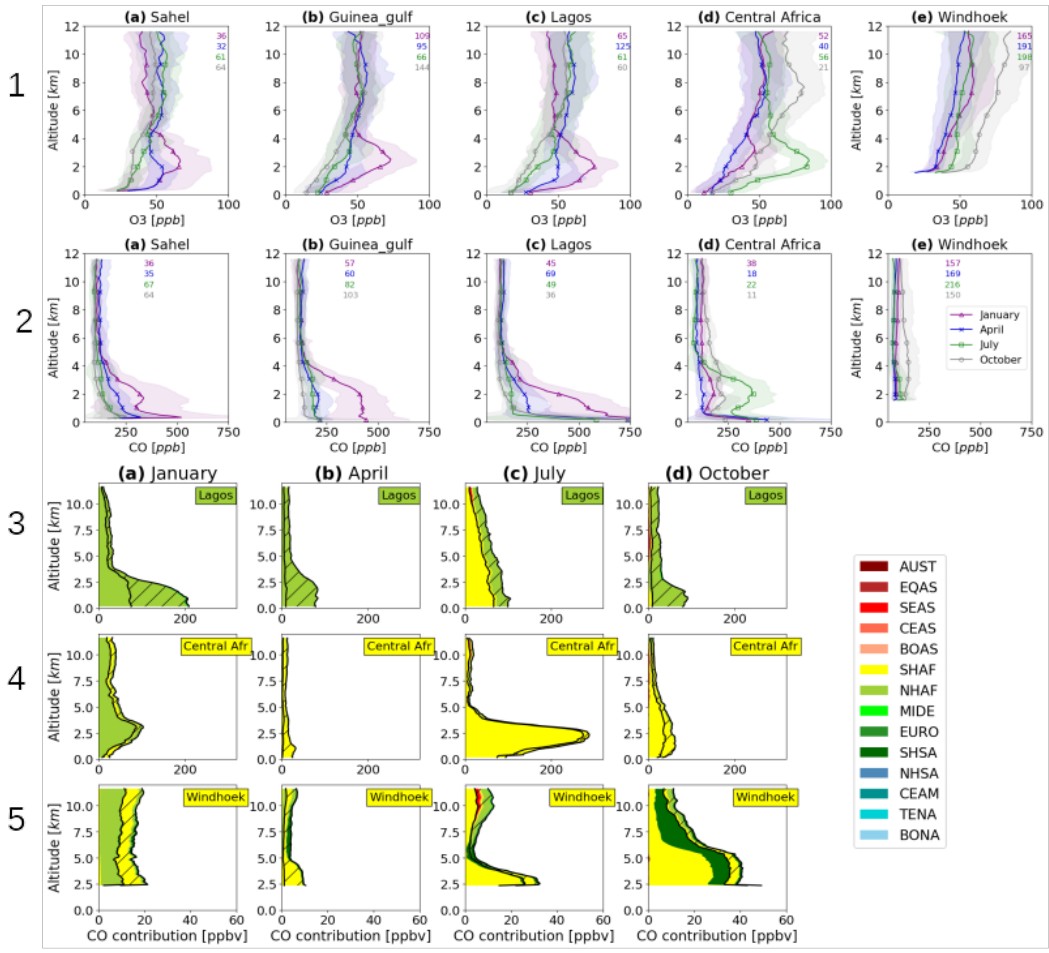

**Figure 4.** IAGOS monthly mean $O_3$ (panel 1) and CO (panel 2) vertical distributions for the African clusters and sites. The annotated numbers correspond to the number of flights per month, given in the same colour as in the legend. The shadowed part corresponds to ±1 one standard deviation. The location of the clusters and sites is displayed in Fig. 1. Vertical distribution of CO contributions (in ppb), averaged over all the positive CO anomalies observed in the IAGOS vertical profile (panels 3 to 5). The geographic origin of CO emissions is indicated by the different colours, with the hatched part showing AN contribution, and the non-hatched part BB contribution. Note that the source region where the site belongs to is indicated by the colour of the box.

which characterises the trade current, as moist cooler air above the surface is capped by dry warmer air above, resulting from advection by the Harmattan flow or subsidence in the anticyclone.

Elevated LT CO levels (below 4 km) are observed all year long over the NH African clusters (Fig. 4 panels 2a–2c) except October. During the transition from the NH dry to wet season (April) when the fires are suppressed (Fig. S1b), CO mainly comes from local AN emissions (Figs. 4 panel 3b; A1 panels 1b and 2b), located over Ethiopia and Nigeria (Fig. S2b). The fact that SOFT-IO attributes approximately 80 ppbv of CO to local AN emissions (Figs. 4 panel 3b; A1 panels 1b and 2b),





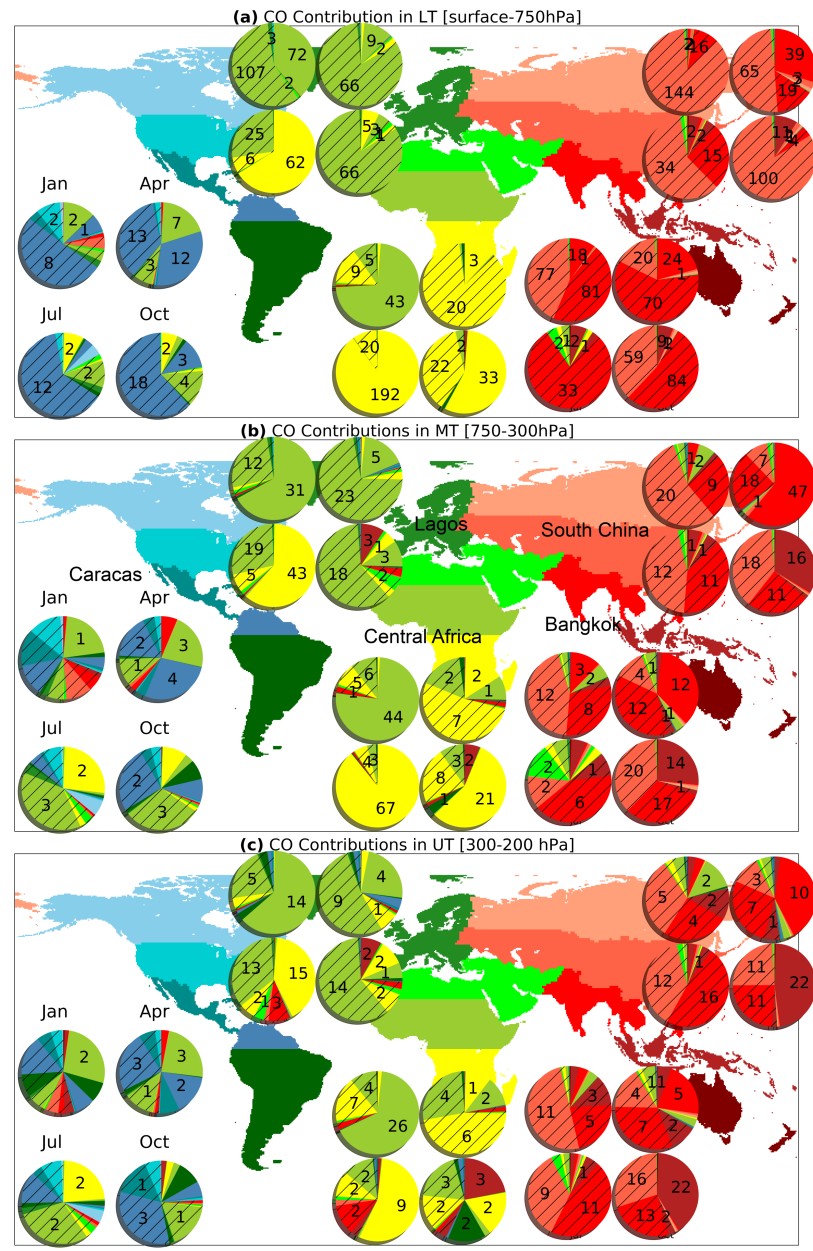

**Figure 5.** Mean SOFT-IO contributions (in ppb), averaged over all the positive CO anomalies for the tropical sites (Caracas, Lagos, Central Africa, South China and Bangkok) for low (a), mid (b) and upper (c) troposphere. The geographic CO origin of CO is indicated by the colours, for the AN (hatched) and BB (unhatched) contributions. Each pie corresponds to a different month and each group of four pies refer to a different site (see panel b).





while the observed anomaly reaches 200–250 ppbv, indicates underestimation of the NHAF AN emissions. Nevertheless, these high CO concentrations in April are detected by IASI in the LT (Fig. 3b) over the whole of West Africa indicating the large-scale extent of the impact of these emissions. The enhanced LT $O_3$ over the three clusters in April (Figs. 4a–c) indicates possible $O_3$ formation during the transport of the aforementioned emissions towards Sahel and the Gulf of Guinea. A small $O_3$ enhancement is also detected by IASI in the mid-troposphere (400–600 hPa) over west Africa (Fig. 3f). During the dry season, nitrogen is accumulated in soils (Jaeglé et al., 2004). The beginning of rains activates the bacterial nitrification leading to significant release of NO which is rapidly converted into $NO_2$ via oxidative processes. Thus, enhanced $NO_x$ concentrations also contribute to the $O_3$ increase over NH Africa in April (Saunois et al., 2009). The $O_3$ enhancement is larger over Sahel relative to Lagos and the Gulf of Guinea (Fig. 4 panel 1a–c) because of higher $NO_2$ concentrations above dry savannas (Sahel) compared to wet savannas and forests (Southern Western Africa) (Adon et al., 2010). After excess nitrogen is consumed, the wet-season NO emissions decrease, contributing less to the local $O_3$ (Adon et al., 2010).

During the NH wet season (July), CO close to the surface over Lagos is mostly attributed to southern African (SHAF) emissions (70 ppb) (Fig. 5a). This result is clearly consistent with IASI LT CO July distribution (Fig. 3c) which suggests transport from the fire region where the highest concentrations are detected towards the Gulf of Guinea and southern West Africa. The contribution of SHAF emissions is relatively constant from the surface to about 5 km (Fig. 4 panel 3c). This suggests transport of chemical mature air masses from the SH African fires towards Lagos, via the south easterly trade winds turning to the monsoon westerly flow (Sauvage et al., 2007a; Barret et al., 2010). The lower $O_3$ mixing ratio close to the surface is likely attributed to rapid deposition to forested areas and photochemical destruction in the moist monsoon air (Reeves et al., 2010). Similar to Lagos, CO over the Gulf of Guinea and Sahel originate from local AN and SHAF BB emissions (Fig. A1 panels 1c and 2c).

Below about 4km, the annual $O_3$ minimum occurs in October over the three clusters (Fig. 4 panel 1a–c). The CO maximum mixing ratio below 1 km is due to local AN emissions (Figs. 4 panel 3d; A1 panels 1d and 2d). In contrast with the other months, the CO mixing ratio above the surface maximum decreases sharply with altitude showing low CO concentrations from 2 km to 12 km. Indeed, in October, the monsoon flow has disappeared and West Africa is impacted by the north easterly trade winds which block the transport of air masses impacted by BB from SHAF as is clearly visible on the LT CO distribution from IASI (Fig. 3d). This is confirmed by the predominant local (NHAF) origin of CO over Lagos which is almost not impacted by SHAF BB in October (Fig. 4 panel 3d). The similar $O_3$ profile close to the surface with the one in July (Fig. 4 panel 1c), indicates significant ozone chemical and depositional sinks as in July. The influence of the moist air over Lagos and Guinea Gulf responsible for the photochemical destruction of $O_3$ below 2.5 km is confirmed by high levels of RH (Fig. not shown).

The classical increase of $O_3$ from the surface to the MT in October (and July) is because the role of photochemistry changes from a net sink to a net source of ozone above 6km, depending on the $NO_x$ concentration (Jacob et al., 1996). In the tropics, photochemical $O_3$ destruction dominates the lower troposphere (Archibald et al., 2020), where water vapour concentrations are high, and in highly polluted regions where there is direct removal by titration with NO (Monks et al., 2015). The vegetation can also act as a rapid sink for $O_3$ via dry deposition (Cros et al., 2000). The lack of these sinks in the free troposphere, coupled





with lower water vapour concentrations leads to an increase of $O_3$ with altitude (Archibald et al., 2020). Lightning can also increase $O_3$ mixing ratios in the MT and UT (Barret et al., 2010).

$O_3$ and CO distribution over Central Africa in the SH dry season (July) is very similar to NH Africa in the respective dry season (Fig. 4 panels 1d and 2d). CO is characterised by two distinct maxima close to 400 ppb close to the surface and between 2 and 4 km, exclusively due to local emissions (SHAF) (Fig. 4 panel 4c). This is visible on the IASI LT CO distribution with

a strong isolated maximum over the whole central African region (Fig. 3c). The contribution of local fires is lower close to the surface (80 ppb) than in the upper layer (280 ppb). The $O_3$ gradient close to the surface is due to influence by the southern monsoon flow (Sauvage et al., 2005) (high RH below 1.5 km; Fig. not shown). CO emitted above the fires (over Angola, Zambia and Dem.Rep. of Congo Fig. S1c) is transported towards the IASI CO maximum over Central Africa via the south easterly winds (Fig. 3c). During the transport $O_3$ is formed (Sauvage et al., 2005, 2007a). As in NH Africa, $O_3$ and CO enhancements

over Central Africa in the dry season are confined below 4 km, because of the stability of the layers below 5 km due to strong temperature inversions.

In October, 3 maxima at the surface, 2 and 4 km visible over Central Africa are due to local emissions (SHAF AN and BB). The lower CO concentrations between the surface and 2 km is probably resulting from the enhancement of the contribution of these emissions with altitude (Fig. 4 panel 4d). In January the vertical distribution of CO is characterised by two maxima, at

the surface and at 2km. Below 1 km, the main contributions are local AN, and BB NHAF emissions. The contribution from the NHAF fires intensifies and becomes the only important one between 2 and 4 km. During transport from northern Africa, the air masses impacted by BB emissions are chemically processed resulting in the formation of an $O_3$ secondary maximum of 50 ppb coincident with the CO maximum (Fig. 4 panel 1d and 2d) as described in Sauvage et al. (2005).

Interestingly, the annual CO surface maximum in Central Africa Occurs in April, before the beginning of the SH fires.

It is due to local AN emissions (SHAF) (Fig. 4 panel 4b). The measured CO maxima reaches 350 ppb, while SOFT-IO attributes 40 ppb above the background levels to the aforementioned sources. This means that SHAF AN emissions are likely underestimated. Above 1km, in the absence of fire contributions, CO remains constant with 100 ppb which is the annual minimum, and the $O_3$ profile is characterised by a steep gradient and the lowest annual concentrations. IASI LT CO distribution (Fig. 3b) indicates that the CO minimum measured by IAGOS above 1km over Central Africa in April extends over the whole

central and southern Africa. The maximum close to the surface is indeed not detectable by IASI.

At Windhoek, $O_3$ and CO maximise in October after SH dry season (July) (Fig. 4 panels 1e and 2e). This CO peak has the smallest magnitude among the African clusters (150 ppb at 4 km), while $O_3$ peak is among the largest, reaching 80 ppb in the UT (11.5 km).

The LT CO anomalies over Windhoek in October are mainly caused by local BB emissions (68 % BB versus 12 % AN)

(Fig. A2a). Interestingly, the contribution from the local fires is larger than in July (by 7 ppb in the LT; Fig. 4 panel 5c), when the peak of the fire emissions occurs (Fig. S1c–d). These high CO concentrations in October are detected by IASI in the LT (Fig. 3d) over the whole of South Africa reflecting the large extent of the impact of these emissions. Using MOPPIT CO and MODIS fire count data, Edwards et al. (2006) also noticed the time lag between the peak of the fires and the CO concentration over South Africa. They attributed the lag to smoldering fires at the end of the burning season. The CO emissions,





and thus the concentrations, are larger over the savanna fires during October, because of the low combustion efficiency of the smoldering fires resulting in increased CO emissions factors (Zheng et al., 2018b). In addition, there is non-negligible influence from Southern Hemisphere South America (SHSA) emissions (20 % mostly BB)(Figs. 4 panel 5d and A2a). The SHSA BB contribution increases with height (30 % contribution in MT and 50 % in UT) contributing to the $O_3$ maximum observed in the UT (Fig. A2b and c) (Sauvage et al., 2006). The absence of a strong CO maximum at the height of the $O_3$ maximum suggests

an additional source of $O_3$ over Windhoek's UT. As already mentioned (Sect. 3.1), the high $O_3$ over South Africa is associated with the South Atlantic $O_3$ maximum which intensifies in October due to the strong $LiNO_x$ production over southern Africa and south America (Sauvage et al., 2007b, c).

It is noteworthy that an $O_3$ enhancement of 80 ppb is also observed over Central Africa at 8 km in October (Fig. 4 panel 1d) highlighting that the stronger seasonal variability of $O_3$ in the mid and upper troposphere in SH than NH Africa is due to the

intense lightning activity in the SH. The IASI UT $O_3$ distribution clearly shows that the $O_3$ maximum covers the entire region from South America to Africa south of the Equator (Fig. 3l).

The annual minimum of $O_3$ and CO over Windhoek occurs in April (Fig. 4 panels 1e and 2e), the transition period from SH wet to dry season, when the local fires are suppressed (Fig. S1b). The CO mixing ratio is less than 100 ppb over the whole tropospheric column and comes from local AN emissions. In comparison to Central Africa, Windhoek is less influenced by

SHAF emissions due to its remote location away from sources (Petetin et al., 2018). The $O_3$ minimum of less than 30 ppb probably results from titration by NO above the surface.

### 3.2.2   Asia: Mumbai, Hyderabad, Madras-India-, Ho Chi Minh City-Vietnam-, Manila-Philippines-, Bangkok-Thailand-, Gulf of Thailand, South China

Over the Asian clusters, the CO profiles display the highest mixing ratios in the surface layer (below 1km) all year long (Fig.

6 panel 2). The annual maximum occurs in January (except over Manila and the Gulf of Thailand) due to the lowest boundary layer height during winter. The winter surface CO maximum ranges from 300 ppb over oceanic sites (Madras) to 700 ppb over megacities (Ho Chi Minh and Mumbai), and is mainly attributed to local or regional AN emissions (Figs. 6 panels 3a to 5a; A1 panels 3a to 5a). Over the Indian and South China clusters in January, local emissions (SEAS and CEAS resp.) are dominant, with contributions in the range of 85 to 95 % (Figs. 6 panel 3a and 4a; A1 panel 3a and 4a and 5a). The impact of these Chinese

emissions (CEAS) is not limited to a local scale, as they dominate the LT CO anomalies over the rest of the Asian clusters (except Bangkok) with contributions in the range of 52 % (over the Gulf of Thailand) to 75 % (over Manila) (Figs. 5a, A2a and A3a). Their advection is favoured due to the northeasterly trade winds. Bangkok is also impacted by CEAS emissions (42 %), but the local AN (45 %) and BB (10 %) contributions are more important (Fig. 5a).

In winter, elevated CO mixing ratios below 2.5 km are related to $O_3$ enhancements (Figs. 6 panel 1). During winter the

chemical ageing of the air masses in the LT is favoured by: i) the confinement of the CO-rich air masses due to the large-scale subsidence preventing upward vertical motions (Lelieveld et al., 2001) and ii) the cloud free conditions promoting $O_3$ formation. This $O_3$ enhancement has been described in Barret et al. (2011) for South Asia during the post monsoon based on IASI $O_3$ data for 2008. It is also visible on the climatological IASI MT $O_3$ map (Fig. 3f). The accumulation of CO and $O_3$ in





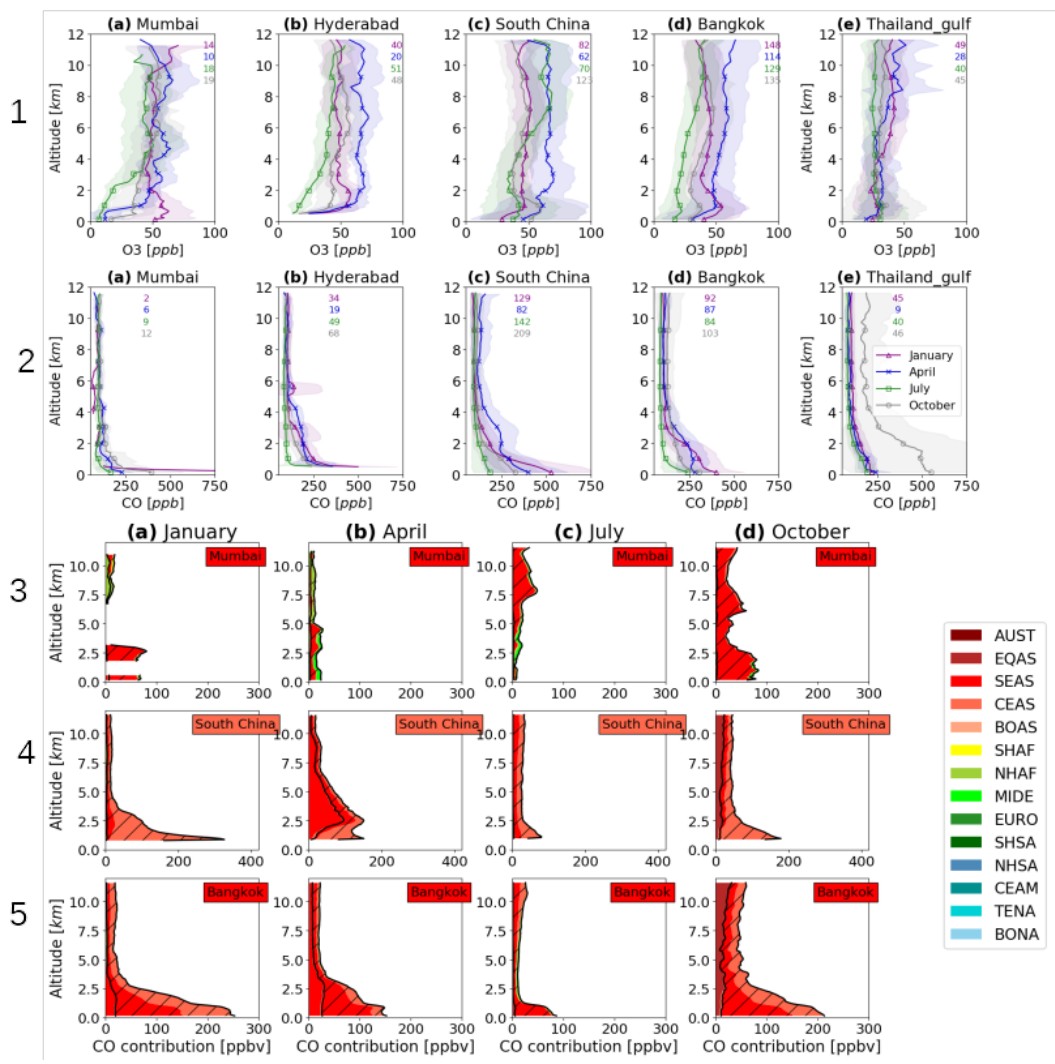

**Figure 6.** Same as Fig. 4 for the Asian clusters.

the LT over the Asian clusters, is observed in lower altitudes than in the NH African ones (see Sect. 3.2.1). This is due to the

lower PBL height in DJF over tropical Asia than Africa as suggested by Kalmus et al. (2022).

During the pre monsoon season (April) CO and $O_3$ are both enhanced above the PBL and below 4 to 6 km over most sites of the Asian clusters. Local AN emissions control the CO anomalies over the majority of the Asian sites (Figs. 6 panels 3b–5b; A1 panels 3b–7b), while spring SEAS fires significantly impact South China and Bangkok, but also Ho Chi Minh City and Manila. The contributions of the SEAS fires in this cluster are in a range of 20 and 30 % in the LT and MT (Figs. 5a and b;

A3a and b). In spring, the fires are mostly located above East Asia and especially the region of Myanmar, Northern Thailand and Laos (Fig. S1b) and the corresponding large CO concentrations are captured by IASI (Fig. 3b). The westward LT and





MT winds above the fires explain the BB outflow towards South China and the Pacific ocean (Figs. 3b and S3b and S3f). The enhanced MT $O_3$ is attributable to the intense solar radiation associated with the important amounts of precursors from AN and BB emissions which were previously evidenced. This is in agreement with the observed $O_3$ maximum in spring over South

China (e.g. Dufour et al. (2010) using IASI data) and Bangkok (Sahu et al. (2013) using IAGOS data). Using observational (IAGOS and IASI) and model (Model for OZone And Related chemical Tracers-version-4 model) data, Yarragunta et al. (2019) found that local AN emissions are responsible for the CO and $O_3$ abundances over South India during the pre-monsoon season. This is in accordance with the SOFT-IO contributions over the Indian clusters (Figs. 6 panels 3b; A1 panels 3b and 4b; A3a). However, it is worth noticing that CO anomalies over Mumbai are also caused by transport of AN emissions from MIDE (36

%) in the LT, and NHAF (30 %) in the MT (Fig. A3a and b). In the UT, where the transport of the air masses is favoured, the impact of NHAF emissions dominates over Mumbai (54 %) and Hyderabad (50 %). The UT CO and $O_3$ transport from NHAF towards the Arabian sea and South India is also captured by the IASI maps (Figs. 3j and 3n), indicating $O_3$ photochemical production during the transport.

The BB contribution is also important during the post monsoon season (October) because of active fires over Indonesia (Fig.

S1). The Gulf of Thailand cluster is the most affected, from the surface (600 ppb), to the UT (Fig. 6 panel 1e). Interestingly, the CO mixing ratio at 6–12 km is approximately 200 ppb over the Gulf of Thailand cluster, the highest CO abundance in the MT and UT among the Asian clusters. IASI CO data (Fig. 3d) and wind fields (Fig. S3d) show that the LT CO-rich air masses impacted by the fires (Fig. S1d) are advected towards the SE Asian coast (South China, Gulf of Thailand, Bangkok and Manila), as confirmed by the SOFT-IO contributions (10 % EQAS contribution on average) in addition to the local AN

influence (Figs. 6 panels 4d and 5d; A1 panels 6d and 7d; 5a; A2a and A3a). The collocated $O_3$ enhancement (below 2 km) over the SEAsian coastal clusters (Figs. 6 pane 1c–1e) indicates $O_3$ production by BB and AN precursors.

The contribution of the EQAS BB intensifies in the UT in October, reaching 40–57 % over the SE Asian coastal clusters, and 33 % (resp. 50%) over Hyderabad (resp. Madras) (Figs. 5c; A2c and A3c). The UT CO maximum above the fires is also captured by IASI and IAGOS UT data (Figs. 3l; 3p and S1d). Based on MLS CO data, Livesey et al. (2013) also found an UT

CO maximum over Indonesia and attributed it to episodically strong convection, in agreement with the low outgoing longwave radiation at Fig. 3p. In contrast, the UT and MT $O_3$ distribution show a SE-NW gradient (Figs. 3l and 3h) with lower $O_3$-levels over the Maritime continent and the southern Indian Ocean and higher ones over India and the Arabian sea. This was reported by Barret et al. (2011) as a result of convection over the first region and subsidence of precursor enriched air masses over the second one.

The LT $O_3$ and CO mixing ratios over the Asian clusters minimise during the summer monsoon (July) (Fig. 6 panels 1–2). The reversal of the north-easterly trades to the monsoon flow (Fig. S3c) results in advection of $O_3$- and CO-poor air masses from the Indian ocean towards Asia. The lowest $O_3$ levels close to the surface are observed over Mumbai due to the stronger oceanic influence (high relative humidity close to the surface compared to the other clusters, Fig. not shown). Furthermore, convective clouds result in cloudy conditions, and rain scavenges $O_3$ precursors resulting in lower $O_3$ production than in clear

sky conditions (Mari et al., 2000; Safieddine et al., 2016). The steep CO gradient close to the surface (below 1 km) clearly indicates the convective uplift of polluted PBL air masses towards the UT. The resulting enhancement of CO in the UT within





the AMA analysed in Park et al. (2008) and Barret et al. (2016) is clear from IASI (Fig. 3k). In contrast, the positive south-north O₃ gradient between the Maritime continent and north SAsia and Middle East (Figs. 3o and w) is associated with: i) the photochemical ageing of air masses while they are recirculating towards Middle East, allowing sufficient O₃ production during
transport (Lawrence and Lelieveld, 2010) and ii) the high insolation over Middle East favouring O₃ photochemical production ((Barret et al., 2016).

### 3.2.3 South America: Caracas -Venezuela-, Bogota -Colombia- and SBrazil

Over Caracas and Bogota, the concentrations of CO in the troposphere are maxima in April and minima in January, while the highest concentrations occur in October over SBrazil (Fig. 7 panel 2). In April, CO concentrations exceed 400 ppb over Bogota
below 1 km above the surface, and 200 ppb up to 2 km over Caracas. The CO concentrations detected by IASI over northern Venezuela and Colombia are also maxima in April (Fig. 3b). This is clearly related to the large vertical extent of the high concentrations which improves the detection by IASI. In terms of CO, Bogota is the most polluted cluster over South America throughout the year.

In January, CO concentrations are below 300 ppb (resp. 180 ppb) over Bogota (resp. Caracas and SBrazil) below 1km
a.s.s.(Fig. 7 panel 2). The CO mixing ratios decrease below 100 ppb above the polluted layers all year long, with exception of South Brazil, where a first maximum (150–200 ppb) occurs in the MT (2–4 km) and a second one (200–250 ppb) is observed above 8 km in October. IASI clearly detects the MT (Fig. 3d) and UT (Fig. 3p) maxima over most of tropical south America in October.

The O₃ profiles over South America generally display the classical increase from the surface to the MT (Fig. 7 panel 1). The
increase of O₃ with altitude is attributed to the lack of depositional and chemical sinks in the FT, in combination with lower water vapour concentrations, and lightning emissions, as discussed in Sec. 3.2.1.

Over Caracas, tropospheric CO is maximum in April (250 ppbv) and the profile displays one maximum of O₃ (40 ppb) in the LT and one in the MT (50 ppb). Over Bogota and SBrazil O₃ is maximum in October over the whole troposphere with mixing ratios reaching 45 and 60 ppb in the UT. For Bogota (resp. SBrazil) the tropospheric O₃ annual minimum occurs in July (resp.
April). As IAGOS over SBrazil below 6 km, the IASI distributions over tropical South America (Fig. 3 e to h) display lower tropospheric O₃ in January and April than in July and October.

From SOFT-IO we can see that, over SBrazil in October, CO enhancement below 1 km is caused by local AN (52 %) and BB (44 %) emissions (Fig. 7 panel 5d). In the MT and the UT, the BB contribution exceeds 80 % because of the strong convection moving over the BB regions (Liu et al., 2010). This is also indicated by low outgoing longwave radiation (Fig. 3l) which is used
as a convection proxy (Park et al., 2007). The uplifted BB products are trapped in an anticyclonic circulation developed over Central South America (Fig. S3l). CO from SHSA fires are transported over Bogota, at the edge of the anticyclone, but does not reach Caracas which lies outside of the anticyclone (Fig. 7 panels 3d and 4d). Their photochemical processing contributes to the seasonal O₃ enhancement over South America which is the western part of the wave-one pattern (Thompson et al., 2003b; Sauvage et al., 2006). This is highlighted by the collocation of IASI UT CO (Fig. 3l) and O₃ (Fig. 3p) maxima within





**Figure 7.** Same as Fig. 4 for the South American clusters.





the anticyclone. Nevertheless, as described by Sauvage et al. (2007c) the lightning activity over South America and Africa in October is the most important cause of the $O_3$ wave-one pattern.

Below 1km, the annual CO maximum over SBrazil in April is due to local AN emissions (Fig. 7 panel 5b) located over the southern part of Brazil (Fig. S2b). The observed CO enhancement reaches approximately 350 ppb (Fig. 7c), while SOFT-IO attributes 65 ppb above the background levels to the aforementioned emissions. This indicates that SHSA AN emissions are

underestimated by the SOFT-IO calculations. The observed CO enhancement at 1.5 km is a new CO feature compared to Yamasoe et al. (2015) that studied CO climatology over SBrazil for the period 1994–2013. The CO enhancement is due to additional data that were collected for the year 2014. SOFT-IO shows increased local AN contributions for March–April–May 2014 relative to the previous years. This is in agreement with the CEDS inventory, which shows a peak in AN emissions over South Brazil (18–29° S and 35–52° W) for the year 2014, mostly coming from the transportation sector (Fig. not shown).

Over Caracas, the annual CO maximum below 2 km in April mostly comes from local AN (35 %) and BB (32 %) emissions (NHSA) (Figs. A2, and 7 panel 5b). This local origin of emissions is corroborated by the elevated IASI LT CO mixing ratios (Fig. 3b) collocated with the strong AN emissions above Colombia and Venezuela, and active fires above the latter (Figs. S1b and S2b). Transport also plays an important role with 20 % of the anomalies caused by BB NHAF emissions (Fig. 5a). The $O_3$ maximum collocated with the CO one at 2 km (Fig. 7 panels 1a and 2a), indicates $O_3$ production during transport of NHAF air

masses impacted by BB. The second $O_3$ maximum above 5 km is also noticed by Yamasoe et al. (2015). Using the GEOS-chem model, they identified local anthropogenic sources followed by lightning, as the main sources of $O_3$ precursors over Caracas in April. In addition, the $O_3$ maximum at around 6 km in October was attributed to local anthropogenic sources and lighting from Africa. This transport pathway from Africa to Caracas in October is confirmed by SOFT-IO (Fig. 7 panel 3d).

It is interesting to note that the CO mixing ratio between 2 and 4 km over Caracas is elevated ($\geq$ 150 ppb) compared

to the free tropospheric background ($\approx$ 100 ppb) all year long. The January–July and October maxima are lower than the April one with dominant CO sources being local AN and African (SHAF or NHAF depending on the season) emissions with contributions of 30 % each. In January there is also a small (15 %) influence from AN central and equatorial Asian emissions (CEAS and EQAS). In Bogota, local (71 %) and NHAF (26 %) emissions control the CO annual maxima close to the surface in April, (Fig. 7 panel 4b). During the rest of the year, local AN emissions ($\geq$ 60 %) control the LT CO anomalies. Emissions

from Africa are also contributing to LT CO in Bogota with 15 % in July (SHAF and NHAF) and 27 % in January (NHAF). In contrast to Caracas, emissions from SHSA also contribute to LT CO anomalies over Bogota, with 31% in July and 10 % in October.

### 3.2.4  Arabia and Eastern Africa (AEA): Khartoum -Sudan-, Addis Ababa -Ethiopia-, Jeddah -Saudi Arabia-, Abu Dhabi

The striking feature of the AEA clusters is the elevated $O_3$ in the free troposphere (FT) (70 ppb on average centered at around 8 km) for all the clusters during April and for the northern clusters of Jeddah and Abu Dhabi during July (Fig. 8 panel 1 a–d). The particularly low CO mixing ratio accompanying the $O_3$ enhancements around 8 km over the AEA clusters (Fig. 8 panel





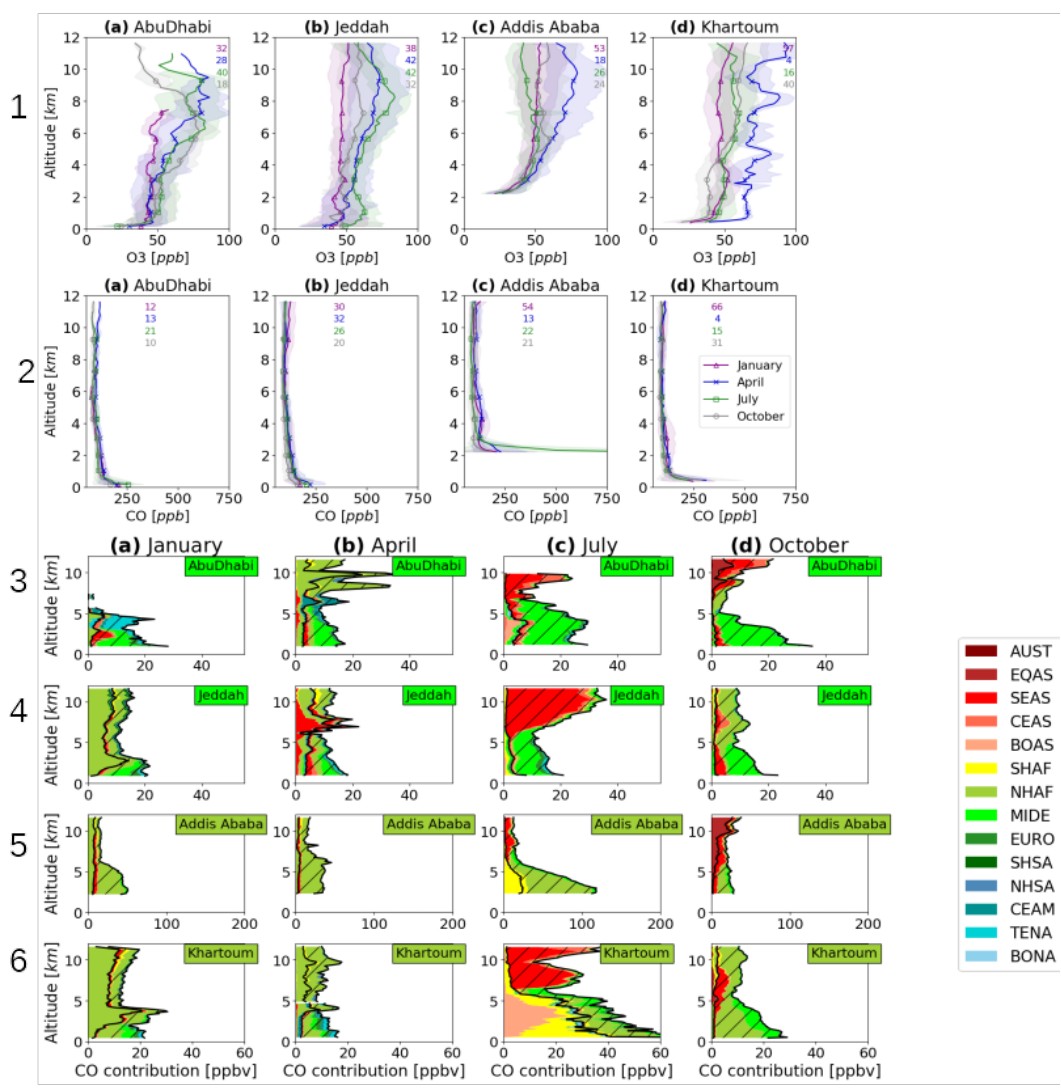

**Figure 8.** Same as Fig. 4 for the Arabian and Eastern African clusters.

2a–d) points to a dynamical origin of $O_3$. The $O_3$ enhancements over the 4 sites of AEA and the anticorellation with CO, are also detected by IASI in the UT and MT (Fig. 3j, k, n and o).

Tropopause foldings in the vicinity of the subtropical jet stream are associated with downward transport of stratospheric ozone (Stohl et al., 2003; Lelieveld et al., 2009; Safieddine et al., 2014) resulting in a tropospheric $O_3$ enhancement. The $O_3$ flux from the stratosphere to the troposphere in the vicinity of the NH subtropical jet peaks during spring and summer (Tang et al., 2011). This is in agreement with Cohen et al. (2018) that found the maximum $O_3$ to CO ratio over the Arabian peninsula during spring and summer (their Fig. A1), using IAGOS data for the period 1994 to 2013. This indicates higher occurrence of 480   $O_3$-rich and CO-poor air masses, reflecting a stronger stratospheric influence. Nevertheless, large $O_3$ regional enhancements





are detected by IASI over the Arabian sea similarly to Jia et al. (2017) based on TOC from OMI/MLS. Jia et al. (2017) attributed these $O_3$ enhancements to emissions from India (50 %), with smaller contributions from the Middle East and Africa (30 %). This is in agreement with SOFT-IO, which shows a significant contribution from SEAS over Jeddah (29 %) and a lower one over Addis Ababa (7%) (Figs. 8 panels 4b and 5b, and A2a). SOFT-IO also attributes large contributions from NHAF AN and

BB emissions over Abu Dhabi and to a lesser extent over Khartoum (Fig. 8 panels 3b and 6b). The contribution of American sources over Abu Dhabi indicates eastward transport, which is not present in the rest of the AEA clusters. Due to its northern position, Abu Dhabi is affected by the subtropical westerly jet in the UT. In contrast the rest of the AEA clusters are affected by the tropical easterly jet which brings CO from Asian regions.

In July, the Middle East summer $O_3$ maximum is also partly related to subsidence of AMA air masses which brings $O_3$

produced from South Asian AN and $LiNO_x$ emissions (Barret et al., 2016). The polluted air masses from South and SouthEast Asia uplifted by monsoon deep convection are trapped in the AMA which extends westward to Northeast Africa and the Middle East (Barret et al., 2016; Park et al., 2007). Over Khartoum and Jeddah (resp. Addis Ababa and AbuDhabi) 20 ppb (resp. 10 ppb) of CO originates from SEAS at 6–12 km. The impact of the SEAS emissions is stronger over Jeddah (78 %) than over Khartoum (60 %) and Addis Ababa (46 %) (Fig. A2c). Addis Ababa and Khartoum, further to the south, are outside of the

AMA and therefore characterised by lower levels of $O_3$ in the UT (Fig. 8 panel 1). Furthermore, the $O_3$ minimum over Addis Ababa (45–50 ppb) is related to the ITCZ located between 5° N and 10° N during the NH wet season (Lannuque et al., 2021). The UT $O_3$ enhancement over Arabia and the Arabian sea, and the transition to lower concentrations south of the tip of Arabia are also clear with the IASI map (Fig. 3c). The $O_3$ minimum over Africa is caused by uplift of local African $O_3$-poor air masses from the surface in the ITCZ (Fig. 3c). The increase of $O_3$ northwards (such as over Khartoum with 60 ppb) is due to

the $O_3$ production within uplifted CO-rich air masses, transported away from the ITCZ by the upper branches of the Hadley cell (Lannuque et al., 2021).

One common characteristic among the AEA clusters is the elevated CO mixing ratio in the surface layer (below 1km) all year long (Fig. 8 panel 2). The surface maximum is larger over Addis Ababa (700 ppb in July) and Khartoum (350 ppb in April), than in Jeddah and AbuDhabi (<250 ppb). Over the East African sites (Khartoum and Addis Ababa), a layer of enhanced CO

is observed in the FT, in January and April. This winter to spring high CO layer in the FT over Eastern Africa is detected by IASI which clearly shows that it does not reach Arabia (Fig. 3a–b). IASI data (Fig. 3i–l) also displays little annual UT CO variability over this region.

In January, the surface CO maximum is mostly controlled by local AN emissions over the AEA clusters (Figs. 8 panels 3a to 6a). Above the surface layer, strong Ethiopian AN emissions (Fig. S2) control the CO anomalies over Addis Ababa with

contribution of 71 % in the LT and 58 % in the MT (Fig. A2a). Influence from the NHAF fires is also evident (12 % in the LT and 20 % in the MT) (Fig. A2a). The impact from the NHAF fires intensifies over Khartoum and Jeddah between 2 and 4 km with contributions of 58 % and 53 % respectively (Fig. A2a). The effect of the NHAF emissions towards eastern Africa (Khartoum and Addis Ababa) and Jeddah is also detected by IASI (Fig. 3a), which shows a negative eastward CO gradient. As expected, the fire contribution is stronger in the western African clusters such as Lagos, due to the prevailing north easterly

winds (Fig. S3a) (see Sect. 3.2.1). The co-occurring $O_3$ enhancement over Khartoum and Jeddah below 4 km reflects $O_3$





formation during transport from the fires (Fig. 8 panels 1b and 1d). The small enhancement of $O_3$ is also captured by IASI in the MT (Fig. 3e).

In July, the CO surface maximum is again caused by local AN emissions (Fig. 8 panels 3c–5c), except over Khartoum where air masses from SHAF fires are the dominant source of CO (Fig. 8 panel 6c). The combination of local AN (70 %) and SHAF
BB (23 %) emissions is responsible for the annual CO maximum at the surface over Addis Ababa in July (Figs. 8 panel 5c, and A2a). Interestingly, the impact of the SHAF fires below 4km over Khartoum and Addis Ababa is stronger than the impact of local fires during the respective dry season (Figs. 8 panels 5ac and 6ac). The $O_3$ enhancement below 4 km over the Jeddah, Khartoum and Addis Ababa indicates $O_3$ production during the transport of CO-rich air masses impacted by the SHAF fires (Fig. 8 panels 1 j to l). In contrast, over Abu Dhabi the $O_3$ enhancement in the FT (Fig. 8 panel 1a) is likely related to transport
of CO-rich air masses from the MIDE and BOAS regions (Fig. 8 panel 3c).

In October, long range transport from Asia (SEAS AN and EQAS BB) plays a significant role in CO anomalies over the AEA sites (Figs. 8 panels 3d–6d), especially over Addis Ababa and Jeddah. In the LT, the northeasterlies (Fig. S3d), transport CO-rich air masses from Asia towards eastern Africa. This transport of CO from Asia over the Arabian sea is well captured by IASI (Fig. 3d).
Above 4km in October, $O_3$ enhancements are observed over the AEA sites especially over Abu Dhabi which is the east-ernmost site of the AEA region (Fig. 8 panel 1). IASI detects a MT $O_3$ increase above the Arabian sea and Northern India (Fig. 3h). The $O_3$ enhancement in the MT over the Arabian sea detected with ozone soundings during the INDOEX campaign (1999–2000) has been attributed to Indian sources uplifted over the marine boundary layer by the sea breeze circulation in Lawrence and Lelieveld (2010). It was further analysed and documented with IASI $O_3$ data by Barret et al. (2011) who already
highlighted the MT $O_3$ enhancement over northern India and the northern part of the Arabian sea during the post-monsoon season. The $O_3$-rich air masses are further transported towards Eastern Africa by the prevailing northeasterlies (Fig. S3h) as documented by the predominant SEAS origin of FT CO over the AEA sites (Figs. 8 panels 3d and 6d).

### 3.3 Control factors of tropical $O_3$ and CO

In this Section we present the main features of the tropical $O_3$ and CO distributions. Figure 9 displays the annual maxi-
mum/minimum of $O_3$ (a) and CO (b) mixing ratios and their corresponding mean height. The annual maxima/minima are calculated based on monthly averaged mixing ratios over vertical layers with 40 hPa thickness. Figure 10 displays the transport pathways of CO emissions from the African, South American and Asian source regions, towards the 20 tropical sites in the LT (a), MT (b) and UT (c). We show the source regions and the months corresponding to the largest amounts of transported CO (in ppb). Figure 11 displays the AN and BB contribution to CO anomalies (in ppb) over the tropical UT (300–185 hPa).
Overall, the CO profiles above all tropical clusters display an annual maximum above the surface layer (approximately at 0.5 km) (Fig. 9b). This is also valid for Caracas, Bogota, Windhoek and Addis Ababa which are located at high altitude above the sea surface (with a mean elevation of 0.9 km, 2.6 km, 1.6 km and 2.3 km respectively). For all the clusters located in the NH tropics (African, Asian and South American), the CO-polluted boundary layer, is mainly attributed to local AN emissions, even for clusters such as over West Africa, where BB is expected to be of great importance (Reeves et al., 2010; Mari et al.,





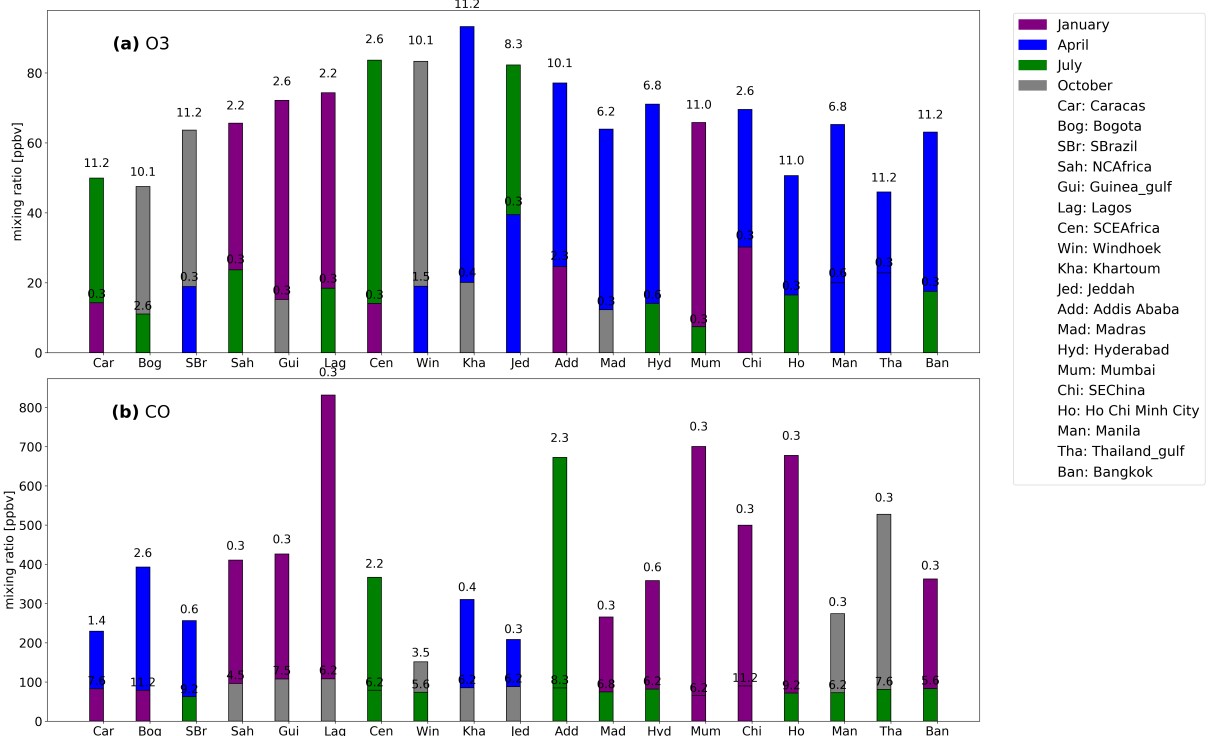

**Figure 9.** O$_3$ (a) and CO (b) annual maximum (higher bar) and minimum (lower bar) mixing ratio observed over the tropical clusters. The annotated number on top of each bar indicates the altitude (in km) of the observed annual maximum/minimum mixing ratio. The colour in the bar indicates the month of the maximum/minimum.

2008; Sauvage et al., 2005) (Figs. 5a; A2a; A3a). This finding confirms the key role of the AN emissions in the NH, related to larger population compared to the SH, and enhanced AN urban and industrial activity, as discussed in Edwards et al. (2004). Concerning the SH, the surface-layer pollution is predominantly caused by BB over SH Africa (Central Africa and Windhoek) during the dry season (Figs. 5, A2), and by AN over the SBrazil (Fig. A2). Interestingly, the CO maxima over the latter occurs before the burning season. This is in accordance with previous studies suggesting fossil fuels as the main CO source over Sao Paulo and Rio de Janeiro (Alonso et al., 2010), and decreasing BB over South America (Andela et al., 2017; Deeter et al., 2018). The decrease of BB CO emissions is due to the long-term declining deforestation rates, especially over forested areas (≈ 54 %) and over savanna and shrublands (≈ 39 %) (Naus et al., 2022). The importance of the AN emissions is also evident over Central Africa with non negligible contributions during dry season (10 %) (Fig. 5). Also, the polluted surface layer over Central Africa is present all year long, with large AN contributions of 40 % and 86 % during the transition seasons, when the fires are suppressed (Fig. 5a). Thus, the impact of the AN emissions is also important in the SH.

The CO maxima show strong variations in terms of magnitude and season among the tropical clusters. This is because they are mostly caused by local emissions with varying intensity and seasonal pattern, depending on the region. In contrast, the CO





**Table 3.** Total (AN + BB) CO emission rates (in $10^{-10}\,kg\,m^{-2}\,s^{-1}$) based on CEDS and GFAS emission inventories over West Africa (10°
W–12.5° E; 0–12.5° N), Central Africa (10–35° W; 2.5–20° S), East Asia (92.5–110° E; 10–27° N), Maritime Continent (93–121° E; 10°
S–10° N), South Brazil (35–50° W; 0–20° S) and Arabia and Eastern Africa (30–60° E; 5–25° N).

|  | West Africa | Central Africa | East Asia | India | Maritime Continent | South Brazil | AEA |
|---|---|---|---|---|---|---|---|
| January | 6 | 1 | 4 | 3 | 1.5 | 1 | 1.5 |
| April | 3 | 1 | 11 | 3.5 | 1.5 | 0.5 | 1 |
| July | 2.5 | 10 | 3 | 3 | 2 | 1 | 0.5 |
| October | 2.5 | 3 | 3 | 3 | 6 | 4 | 1 |
| Annual | 3.5 | 3.7 | 5.5 | 3.1 | 3 | 1.5 | 1 |

minima are uniform in terms of intensity levels of mixing ratios, close to the CO background levels of 100 ppb, due to mixing
and transport over the lifetime of CO. As expected, they occur in the FT, in the absence of the emissions and where CO is
chemically destroyed. As for the CO maxima, their strong seasonality is related to the seasonality of the surface emissions and
the meteorological conditions, which differ over each region. Further discussions on the magnitude and the seasonality of the
CO maxima and minima will follow later.

Because of its complex chemistry, the situation for $O_3$ is more complicated. Africa is the only region where the annual
$O_3$ maximum occurs in the LT (2.5 km) during the dry season (Fig. 9a Sahel, Guinea Gulf, Lagos and Central Africa). The
co-occurrence of maximum $O_3$ with the maximum in CO over Africa during the local fires indicates stronger dependency of
$O_3$ on the surface BB CO emissions for these regions, in agreement with Sauvage et al. (2007b). South China is the only Asian
cluster where the annual $O_3$ maximum is observed in the LT during the active local fires (April), but it is not accompanied
by the annual CO maximum (Fig. 9), suggesting that the ozone maximum has been formed differently. In contrast, over the
other regions, the annual $O_3$ maximum is observed in the FT above 6 km (Fig. 9a). This likely indicates that $O_3$ is formed by
photochemical processes, and is associated with larger ozone production efficiency (Sauvage et al., 2007c). In regions such as
Arabia (Jeddah and Abu Dhabi), the lack of CO enhancement in the UT indicates dynamical origin of $O_3$ (e.g. stratospheric
influence and transport of $O_3$ and precursors from Asia; see Sec. 3.2.4). In contrast, in regions such SBrazil and Windhoek in
October, the co-occurrence of $O_3$ and CO enhancement in the MT and UT indicates tropospheric origin for $O_3$ (e.g. fires and
$LiNO_x$ emissions; Secs. 3.2.1 and 3.2.3). The annual $O_3$ minima for all the tropical clusters are observed close to the surface
(below 0.5 km on average) (Fig. 9a). This is related to the chemical and deposition sinks of $O_3$ located in the LT (see Sect.
3.2.1 for more details).

The highest CO and $O_3$ maxima among all the tropical clusters occur over NH Africa in the LT during the dry season
(January) mostly due to local AN emissions. According to IASI (Fig. 3), the CO-rich and $O_3$-rich air masses due to the
African emissions show a large extend along the tropical Equatorial Africa, and accumulate in the LT due to the stability of
the Harmattan winds (Sauvage et al., 2005). Table 3 displays the total (AN and BB) CO emissions rates over several regions of
interests based on the sum of CEDS and GFAS emission inventories. Indeed, the $O_3$ and CO maxima co-occur with the highest





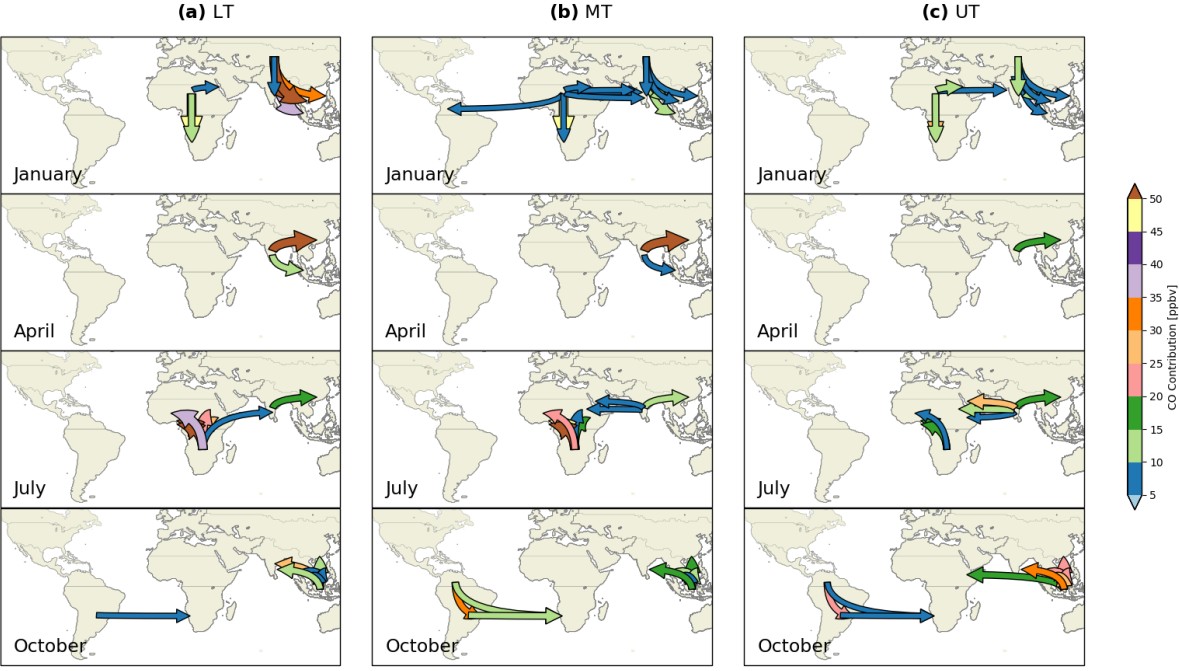

**Figure 10.** Transport of CO (AN+BB) emissions from the African, South American and Asian source regions towards the 20 tropical sites taken into account for this study. The colorbar shows the amount of CO transported in ppb.

emissions over Western Africa in January, confirming their strong dependency on the surface emissions. The $NO_x$-limited $O_3$ production regime over Western Africa (Saunois et al., 2009; Zhang et al., 2016) likely explains the $O_3$ maxima when the local emissions, and thus the $NO_2$ concentration (Jaeglé et al., 2004), intensify in the region. The largest $O_3$ and CO mixing ratio

over Lagos (Fig. 9) is due to its proximity to the strong Nigerian AN emissions, as confirmed by SOFT-IO (see Sect. 3.2.1). As expected, the impact on CO is higher in the proximity of the emissions, while the CO mixing ratio decreases downwind (towards Sahel and Gulf of Guinea) (Fig. 9) because of physical processes, such as dilution by mixing and entrainment (Martin et al., 2017), and CO consumption in $O_3$ build-up in fires (Chatfield et al., 1996). The $O_3$ maxima show smaller variations (of approximately 10 ppb) among the NH African clusters. In contrast to CO, the $O_3$ enhancement does not strongly depend on the

proximity to emissions, as it is produced during the transport and chemical ageing of air masses rich in precursors (Sauvage et al., 2007b).

The second highest CO and $O_3$ maxima over the tropical regions are observed over Asia (Fig. 9). As in NH Africa, the CO maximum occurs in January, when the stability of the trade wind results in accumulation of CO-rich air masses in the LT. The surface-layer CO maximum is attributed to local AN emissions over Indian and South China clusters (85–95 %). In the rest

of the Asian cluster, there are non-negligible contributions from regional AN sources of the Asian cluster in addition to the dominant local ones. According to Table 3, the CO emissions over East Asia and India are lower than the ones over Western





Africa in January. This explains the lower CO mixing ratio over Asia than over Lagos. SOFT-IO seems to represent better the Asian contributions relative to the Africa ones. As mentioned in Sect. 3.2.1, the AN emissions over NH and SH Africa are likely underestimated by the SOFT-IO computations (Fig. 4 panels 3b and 4b). This is confirmed by that fact that CO mixing

ratio is higher over Africa than over Asia, in contrast to the CO contributions estimated by SOFT-IO. Thus, the impact of the African AN emissions is likely underestimated.

     Previous studies have already found concentrations of pollutants in West Africa (e.g. Lagos, Abidjan, Cotonou) comparable to those observed over Asian megacities (Assamoi and Liousse, 2010; Adon et al., 2016; Sauvage et al., 2007b). Indeed, the rapid growth over African megacities is responsible for increasing emissions from diffuse and inefficient combustion sources

(Marais and Wiedinmyer, 2016). This increase is mostly attributed to the growing residential source mainly for cooking and heating (Zheng et al., 2019), but also to traffic emissions (related to large number of two-stroke vehicles, poor fuel quality and poorly-maintained engines) (Assamoi and Liousse, 2010). In contrast, Eastern China has had one of the largest decreases in CO emissions (Hedelius et al., 2021) due to technological changes with improved combustion efficiency and emission control measures (Zheng et al., 2018a). Using MOPPIT data for the period 2002–2018, Buchholz et al. (2021) found the largest

reduction in CO concentrations over China. This reduction is attributed to declines in local CO emissions since 2002, related to replacing residential coal use with electricity and natural gas, and to the implementation of Clean Air Policies (van der A et al., 2017) around 2010. In India, on the other hand, there are no regulation in the emissions, and this explains the highest CO mixing ratios among the Asian clusters (Fig. 9). Previous studies have already reported increasing CO emissions over India from 1996–2015, due to several factors such as increases in residential and agricultural sources (Pandey et al., 2014) and to

power production and transport activities (Sadavarte and Venkataraman, 2014).

     As in NH Africa, the CO-rich air masses accumulated in the LT over the Asian clusters in January result in a secondary LT $O_3$ maximum. However, these maxima are significantly lower (40–60 ppb) (Sect. 3.2.2) than the NH African ones (65–75 ppb) (Fig. 9a; Sect. 3.2.1), even for clusters with similar LT CO mixing ratios (e.g. Sahel and South China) (Fig. 9a). This is because: i) the CO emissions are less strong over the Asian clusters, as mentioned before, and ii) the $O_3$ enhancement over Asia

is caused by AN-polluted air, while in NH Africa by mixed (AN and BB) polluted air. During the Atom campaign, Bourgeois et al. (2021) found that $O_3$ levels are more enhanced in mixed air pollution, because they are associated with greater $NO_x$ and peroxy acyl nitrates (a $NO_x$ reservoir compound), and thus increased $O_3$ production, in comparison to BB- or AN-polluted air alone. This is in agreement with the $O_3$ annual maximum in April over East Asia (Fig. 9a), over clusters such as South China and Bangkok, which are affected by the local fires.

Unlike Africa, the highest emission rates over East Asia and India are observed in April (Table 3). In the absence of the stability of the north easterlies, the air masses are not confined close to the surface like in January, and thus the secondary CO maxima above the surface is also captured by IASI (Fig. 3f). Over East Asia, the contribution of the local fires is also present in addition to the local AN emissions. The impact of the fires dominates in clusters such as South China and Bangkok, and is evident over Manila and Ho Chi Minh City (see Sect. 3.2.2). Interestingly, the NH African fires in January correspond

to 72 % of the global burned area, whereas the NH Asian fires only to the 2.5 % (Van der Werf et al., 2010). However, both regions contribute significantly to the global CO concentrations (44 % for Africa and 22 % for Asia) because of more





complete oxidation, and thus reduced CO production, over grass fires (Africa savannas), relative to fires in forests and peatlands (deforestation and peatland fires over Asia) (Van der Werf et al., 2010). The large extend of the impact of the NH Asian fires is displayed in IASI map, with an outflow towards SE Asian coast and the Pacific (Fig. 3f). The stronger winds in April than

in January, which does not favour the accumulation of the pollution, and the eastwards transport pathway (Fig. 10 panel 1 a–b) leads to lower CO mixing ratio in April than January, despite the higher emission rates (Table 3).

Concerning India, local AN emissions are responsible for the CO enhancement in April, with negligible BB contribution (Figs. 6 panel 3a and A1 panels 3a and 4a). The CO emissions rates over India are high during the whole year, showing weak seasonal variability (Table 3). The LT CO distribution over India shows strong seasonal variability which is not explained by

the seasonality of the emissions. The LT CO is rather linked with seasonal changes in the meteorological circulation. Similarly to East Asia, during January the air masses are transported southward due to the north easterlies, while the reversal of the winds to southwesterlies in July results in northward transport (Figs. 3 and S3) (Lawrence and Lelieveld, 2010). Because of this circulation pattern in July, the oceanic influence brings clean air masses over the Asian clusters resulting in an annual CO minimum during the Asian summer monsoon (Fig. 9). The CO-rich air masses for the surface are uplifted in the upper

troposphere due to deep convection over the area (Sect. 3.2.2).

As for CO, $O_3$ seasonality is also linked with the seasonality of the meteorological conditions and dynamics over Asia. The $O_3$ maximum in April is attributed to the intense solar radiation associated with important amount of precursors from mostly AN emissions, except for South China where BB emissions dominate. The $O_3$ minimum occurs during the asian summer monsoon (July), because of lower $O_3$ production in the presence of convective clouds relative to clear sky conditions (Sect.

655     3.2.2).

Despite the CO emissions reductions over South China, the $O_3$ levels remain relatively high (Fig. 9a). This is because the $O_3$ production regime over South China is VOCs-limited (Li et al., 2013), and the total NMVOCs emissions increased in China by a factor of 3.5 (1997–2017) because of activity increases in the solvent, energy, and industry sectors (Zheng et al., 2018a). As discussed in Wang et al. (2017), despite the successful controls of $NO_x$ emissions from coal fired power plants since 2010 over

Eastern China, it is recommended to apply controls over VOCs emissions as they control the local $O_3$ distribution. In contrast, over India the $O_3$ production regime is $NO_x$-limited (Kumar et al., 2012), as the local emissions are mostly associated with incomplete combustion proccesses by biofuel burning, and thus higher NMHC to $NO_x$ emission ratio as compared to other regions of the Northern Hemisphere (Lawrence and Lelieveld, 2010).

Concerning Central Africa, the $O_3$ and CO maximum in the LT during the dry season, indicates the strong dependence of the

CO and $O_3$ distribution on the surface emissions, as over NH Africa. The CO magnitude over Central Africa is similar to the one over Sahel and Guinea Gulf during the respective dry season, even though the emissions rates are higher over the former (Table 3). This is because higher amount of CO impacted by the SH African fires is transported towards the NH Africa due to the trade winds, relative to the respective southward transport during the NH dry season (Fig. 10). In addition, the $O_3$ mixing ratio is slightly higher over Central Africa (85 ppb) likely indicating rapid photochemical $O_3$ production by BB precursors

(Singh et al., 1996) during the SH fires. Concerning Windhoek, the $O_3$ maximum in the FT (85 ppb at 10 km) (Fig. 9) indicates





that $O_3$ production is controlled by $LiNO_x$ emissions at higher altitude (Sauvage et al., 2007b) during the peak of the lightning activity over South Africa (Fig. not shown).

The smallest LT CO maximum over the NH are observed over Arabia and East Africa clusters and South America (Fig. 9) because of the smallest emissions rates among the tropics (Table 3). The CO emissions over Middle East are mainly related

to electricity generation, water desalination, and industry supplied by oil and gas deposits with cheap but relatively clean fuels (Krotkov et al., 2016). In addition, because of its location between the two highest emittors (Asia and Africa), transport plays a significant role in CO enhancements over AEA, especially in the MT and UT where long range transport of emissions is favoured (Figs. 10 panels 1 b–c; 2 a–c and 2b–c). This transport from Asia and Africa over AEA clusters determines the $O_3$ maxima over the AEA clusters (Sect. 3.2.4). Similarly, over NH South America, the local AN contributions are much smaller

than the respective local Asian of African ones, indicating lower pollution levels over South America than Asia and Africa. The $O_3$ maximum is controlled by $LiNO_x$ emissions at higher altitudes.

From the previous analysis, all the tropical clusters and the associated CO source regions exhibit primarily local influence, in the proximity of the region where they are emitted. However, CO transport plays also an important role in the CO distribution over the tropics. CO sources located over Africa show the maximum influence on the regional tropical CO. The highest impact

of the African emissions is found at an inter-hemispheric scale, where CO from the dry-season African regions is transported towards the wet-season African (Fig. 10 panels 1 and 2). As a result, CO contributions of 45–50 ppb (resp. over 50 ppb) from NHAF (resp. SHAF) is found over SHAF (resp. NHAF) during the respective dry season in the LT and MT. This transport of precursors results in a secondary $O_3$ maximum, as can be seen by IASI maps (Fig. 3e and g).

Impact of the SHAF BB in July is also found in the LT over India with contributions of 5–10 ppb (Fig. 10 panel 2a).

Similarly, NHAF AN and BB contributions in January are found over South America (5–10 ppb in the MT) and India (5–10 ppb in the MT and UT) (Fig. 10 panel 1a). The impact of the NHAF emissions towards South America (10–15 ppb in MT; 5–10 ppb in the rest) is increased in April (Fig. not shown) and significantly contributes to the local South American annual maximum (30 % and 50 % of CO anomalies over Caracas (LT and MT resp.) (Fig. 5a–b).

During the transport of the SHAF (resp. NHAF) emissions towards the ITCZ location in the North (resp. South) Africa, the

air masses reach convective regions and are injected in the North African (resp. South African) upper troposphere (Fig. 11). This explains why the SHAF (resp. NHAF) emissions are dominant in the wet-season hemisphere during July (resp. January) (Figs. 11 and resp. A6 and A4). Nevertheless, the NHAF contribution in the UT CO anomalies is present on a local scale all year long, above NH Africa and South Atlantic. During the dry season, the impact of the NHAF emissions is stronger and extends to a wider area over South America, Middle East, South Asia (Figs. A4 NHAF).

The impact of CO emissions from South America is extended over South Africa during October. This is because of the anticyclone over Central South America which traps the CO emitted locally, and transports it towards the east by westerlies (see Sect. 3.2.3). The highest transport in terms of CO amount from NHSA and SHSA occurs in the MT (10–15 ppb each). IASI maps show an $O_3$ and CO enhancement over the tropical South Atlantic (Figs. 3 d and h, 3l and p). Thus, the South American emissions mostly coming from BB (Figs. 11, A7 SHSA and NHSA) contributes to the wave-one pattern. Nevertheless, the





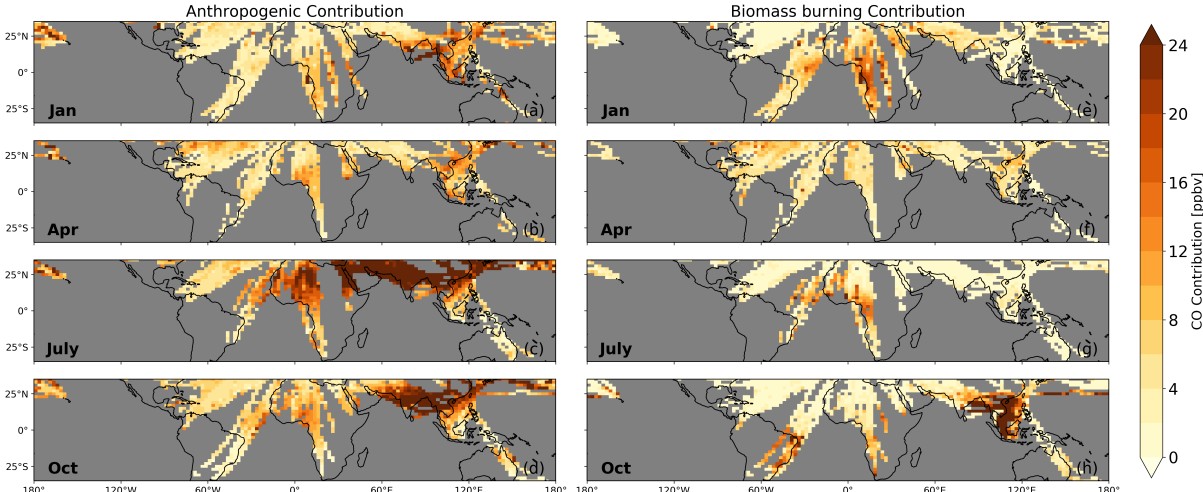

**Figure 11.** Mean AN (a–d) and BB (e–h) contributions in ppb over the tropical upper troposphere (300–185 hPa) from 2002–2019.

most important source of $O_3$ over the tropical South Atlantic is $LiNO_x$ emissions from South America and South Africa, as highlighted by previous studies (Sauvage et al., 2007b, c).

The contribution of Asian emissions in the tropical LT is limited to a regional or local scale, as they are mostly impact neighbour Asian regions (Fig. 10a)(see Sect. 3.2.2 for more details). CO export from Asia is favoured during the Asian summer monsoon and post monsoon (July and October) in the UT, where the transport is favoured due to stronger winds relative to the 710 surface (Fig. S3).

During the Asian summer monsoon, the CO-rich (and $O_3$-poor) air masses from the PBL (Fig. 3c and g) are convectively uplifted in the UT (Figs. 11c and A6), and trapped in the AMA circulation (see Sect. 3.2.2). The impact of the Asian emissions on the UT CO anomalies is extended over Arabia (25–30 ppb) and Eastern Africa (25–30 ppb) (Figs. 11c; 10 panel 3c and A6 SEAS and CEAS). Interestingly, subsidence of air masses from AMA above Arabia are responsible for an $O_3$ maximum above 715 AEA (Fig. 9a) (Sect. 3.2.4). During the post monsoon season (October), sporadic convection uplifts CO emitted by EQAS fires (Figs. 11g, A7 EQAS) (see Sect. 3.2.2). At the same time, convection over continental Asia uplifts SEAS AN emissions (Figs. 11d and A7 SEAS) in the UT. As a result, CO from the Asian emissions impacts CO anomalies in the UT over Eastern Africa with EQAS contribution of 15–20 ppb, and SEAS contribution of 5–10 ppb (Fig. not shown).

## 4 Summary and conclusions

IAGOS $O_3$ and CO observations since 1994 and 2002 respectively, were used in order to analyse vertical profiles over 20 tropical sites, along with the (lower part) of the upper tropical troposphere. IAGOS data combined with global 2D distributions based on IASI-SOFRID $O_3$ and CO retrievals since 2008, were used in order to study the characteristics and seasonal variability of the tropical tropospheric $O_3$ and CO distributions.





In the LT, the CO anomalies over the tropics are caused by a combination of AN and BB emissions. In the majority of the
clusters, local AN contributions are dominant all year long. The BB contribution increases or dominates over some clusters,
when the regional or local fires are active. Local AN emissions have greatest impact over Asia where they account for more
than 80 % of the CO. The BB impact increases over South China (35 % in April), and dominates over the Gulf of Thailand (90
% in October) during the local fires (SEAS and EQAS resp.). Over NH Africa, local AN contributions are in the range of 60–
85 % all year, except July. During the SH dry season, CO impacted by the SHAF fires is transported northwards contributing
significantly to LT CO anomalies over NH Africa (53–66 % over Lagos and Guinea Gulf). Similar impact of the SHAF fires
is found over Khartoum in July. In contrast, the rest of the Arabian and Eastern Africa clusters are impacted by local AN
emissions all year long (70–95 %). Over South America, stronger AN contribution are found over the SH (81–94 % over
SBrazil) than in the NH (75–80 % Caracas and Bogota), while the contributions from the local fires are similar (51 % over
Caracas in April and 53 % over SBrazil in October). The highest BB impact is found over SH Africa during the NH and SH dry
season with contributions of 60 and 90 % respectively. As expected, the local BB dominate the LT CO anomalies during the
local fires, however there is important transport from the NHAF. Despite the fact that BB dominates over SH Africa during the
dry seasons, the AN emissions are important during the transition periods (46 and 80 %). Our results highlight the importance
of the AN emissions over the tropics, even in the SH. This is in accordance with the global decreasing trends of BB (Andela
et al., 2017) and the increasing AN emissions (Granier et al., 2011).

In the MT and UT, the BB contributions are increased compared to the LT, and their effect dominates over more clusters.
Also, the contribution of the transport is more important than in the LT, where mostly local emissions dominate. Over NH
Africa, the BB dominates twice a year, during the NH and SH dry seasons, because of local and SHAF fires respectively. In
SH Africa, as in the LT, BB dominates all year long except April. In addition to the African BB, AN SEAS and BB SHSA
contributions are found in the MT and UT. Over Asia, BB from SEAS in April, exceeds the AN contribution over the SE Asian
coast (South China, Manila, Ho Chi Minh City) in the MT. In contrast, the EQAS BB effect is stronger in the UT, extending
over SE Asian coast (China, Ho Chi Minh City, Manila) and India (Madras), but also Eastern Africa (Addis Ababa).

Over Africa, the O$_3$ and CO maxima are observed in the low troposphere during the respective dry season. The role of the
local AN emissions are more important than previously noted (Reeves et al., 2010; Mari et al., 2008; Sauvage et al., 2005) as:
i) local AN emissions define the O$_3$ and CO anomalies over NH Africa, and ii) the persistent CO-rich surface layer is caused
by local AN emissions (40 and 86 %) in the absence of the local fires. Africa is also the most important tropical region in
terms of export of emissions in the tropical troposphere. According to IASI horizontal distributions, the main export pathway
is the inter-hemispheric transport of O$_3$ and precursors from the dry-season African regions to the wet-season ones ($\approx$ 50 ppb),
confirmed by SOFT-IO contributions. During the dry season, the NHAF (resp. SHAF) fires are the dominant source of CO
over AEA (resp. Khartoum and Jeddah) in the MT and UT, and they also reach India accounting for 5–10 ppb in the MT and
UT. Transport of mostly BB emissions from NHAF and SHAF occurs all year round towards northern South America in all
tropospheric layers. The highest NHAF regional impact is found over Caracas in the MT and UT (30 % on average). In contrast,
the impact of Asian emissions, is mostly limited on a regional or local scale, especially in the LT and MT. The transport of the
Asian emissions is important only during the Asian summer monsoon in the UT towards Arabia and NH Africa.





The highest abundances of the $O_3$ (75 ppb) and CO mixing ratio (800 ppb) among the tropical clusters are found over
Northern Hemisphere Africa at about 2.5 km altitude. This is largely a result of the local AN emissions as suggested by the
co-occurrence of the peaks of O3 and CO in the LT. In contrast over Asia, the second most polluted region, the distributions
are mostly controlled by meteorological conditions associated with the Asian monsoon phase. The CO maximum occurs in the
LT during January, due to the stability of the northeasterlies which confine the CO-rich air masses to the LT. In contrast, annual
maximum of O3 occurs during the pre monsoon season (April) when the increased solar radiation favours O3 production.
During the Asian summer monsoon, $O_3$ and CO mixing ratio minimize in the low troposphere because of : i) transport of clean
oceanic air above continental Asia, ii) reduced photochemical $O_3$ production due to cloudy conditions, and iii) convective uplift
of CO-rich air masses from the surface towards the Asian upper troposphere.

Over Asia, the LT and MT CO and $O_3$ anomalies are mostly impacted by regional or local Asian emissions of AN origin. The
BB contribution is important during April and significantly contributes to $O_3$ and CO anomalies over South China. According
to IASI, the BB impact extends over the tropical Pacific. The impact of the AN Asian emissions is important only in the UT
during the Asian monsoon and post monsoon season (July and October). According to IASI, the polluted air masses from
the surface are uplifted in the UT in July and are trapped in the AMA. These air masses are transported over Arabia and
Northern Africa (CO contributions of 25–30 ppb) causing the annual $O_3$ maxima due to subsidence and high isolation over
the regions. This highlights the importance of long range transport for the air quality in the UT over Arabia, which shows the
lowest CO local contribution and the highest $O_3$ levels among the tropical clusters. The CO transport towards Eastern Africa
in the UT by the Tropical Easterly Jet, is found in October when the air masses impacted by the Indonesian fires, and the AN
continental source are uplifted in the UT, and transported towards Eastern Africa (CO contributions of 15–20 ppb and 5–10
ppb respectively).

Last, over South America the local CO contributions at the surface level are as low as over Arabia and Eastern Asia. During
the dry season (October), when the convection moves over the South American fires, CO and precursors are trapped in an
anticyclonic circulation developed over Central South America, resulting in the annual local maxima of $O_3$ and CO. The
transatlantic transport of $O_3$ and precursors over the Atlantic can be seen by IASI and this contributes to the $O_3$ wave-one
pattern (Sauvage et al., 2007c). This is confirmed by SOFT-IO which calculates contribution of 10–15 ppb from SHSA and
10–15 ppb NHSA, in the altitude of the anticyclone (MT).
Overall, the importance of anthropogenic emissions is highlighted over the tropics, not only in the NH but also in SH.
The interconnections among the tropical regions, especially transport of $O_3$ and precursors originating from Africa, makes it
necessary to assess the pollution on a local scale in order to improve air quality on a local and region scale over the tropics.



# Appendix A: SOFT-IO CO contributions

## A1 Vertical profiles

**Figure A1.** Same as Fig. 4 (panel 3) for CO contributions over Sahel (1), Gulf of Guinea (2), Madras (3), Hyderabad (4), Ho Chi Minh City (5), Gulf of Thailand (6) and Manila (7).





**A1   Low, mid and upper troposphere**

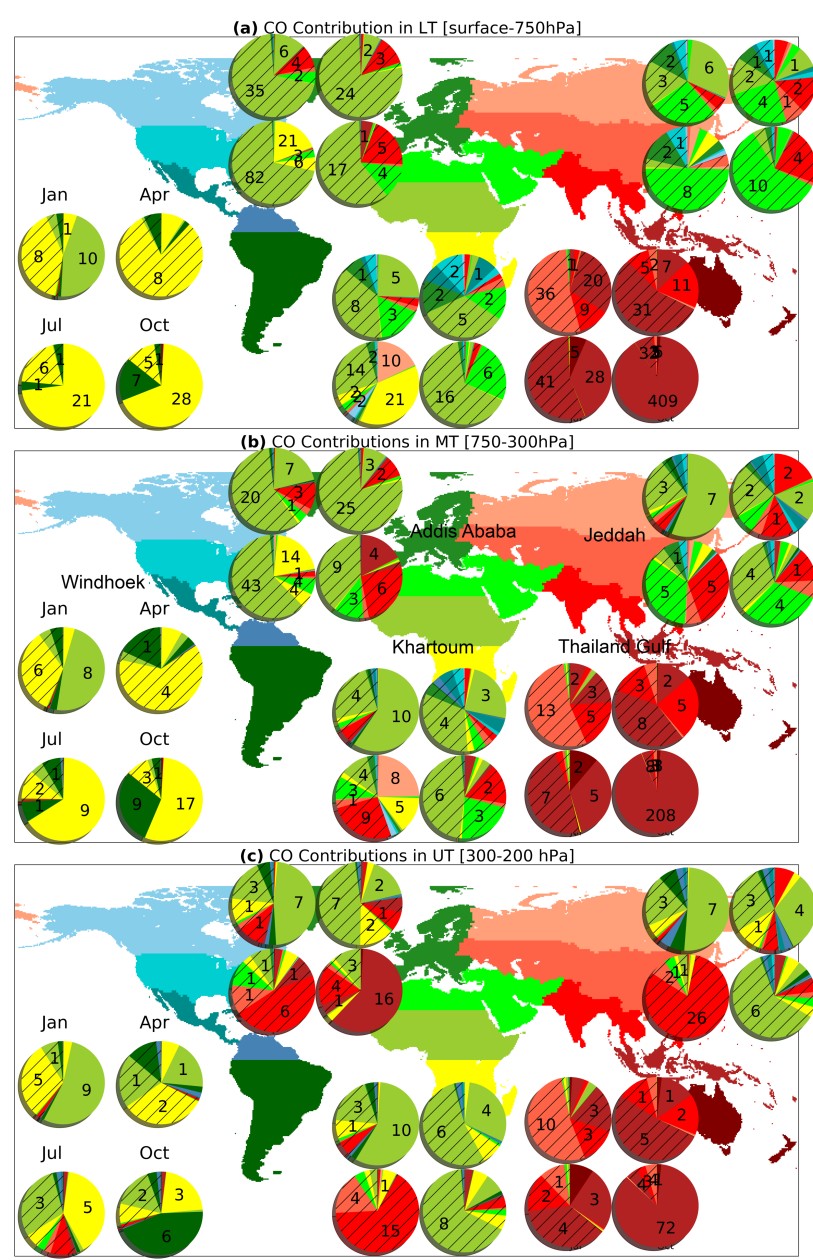

**Figure A2.** Same as Fig. 5 for CO contributions over Windhoek, Addis Ababa, Khartoum, Jeddah and Gulf of Thailand.





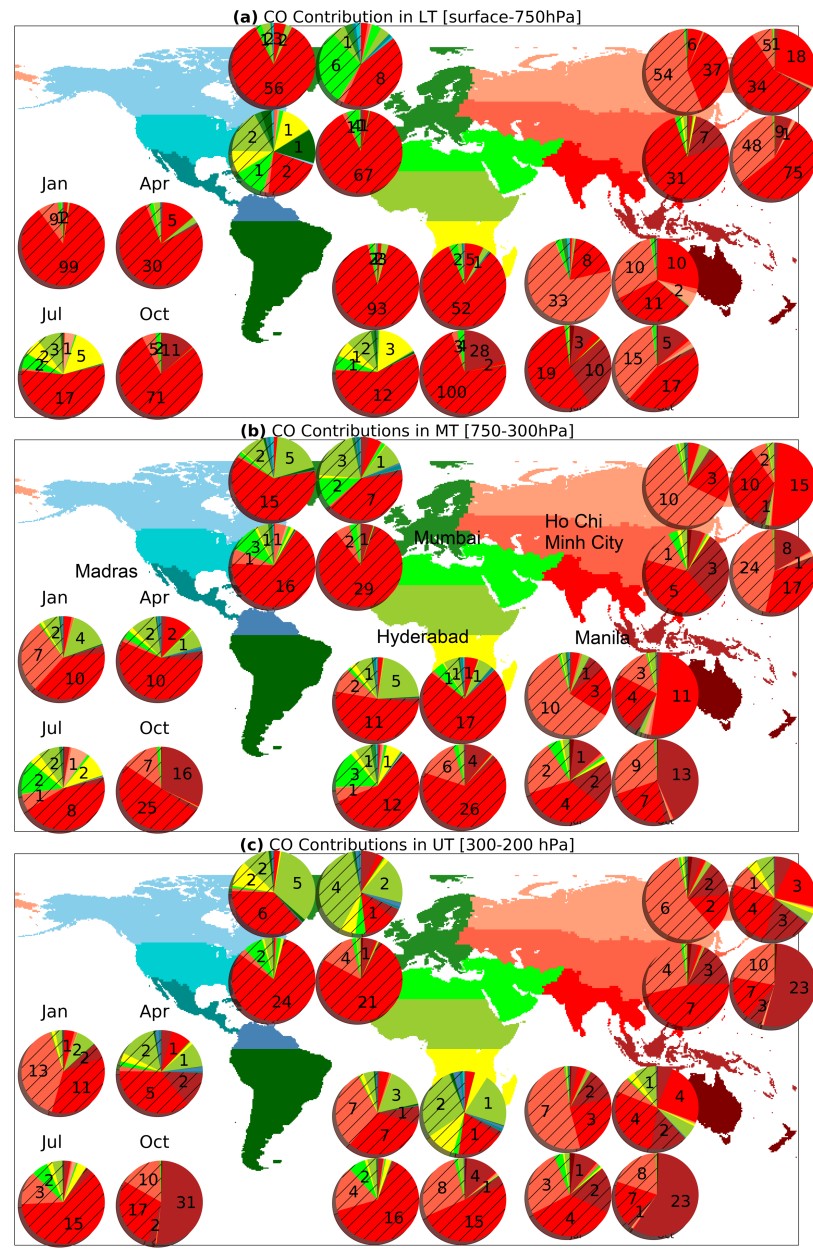

**Figure A3.** Same as Fig. 5 for CO contributions over Madras, Mumbai, Hyderabad, Ho Chi Minh City and Manila.


## A2 Upper troposphere

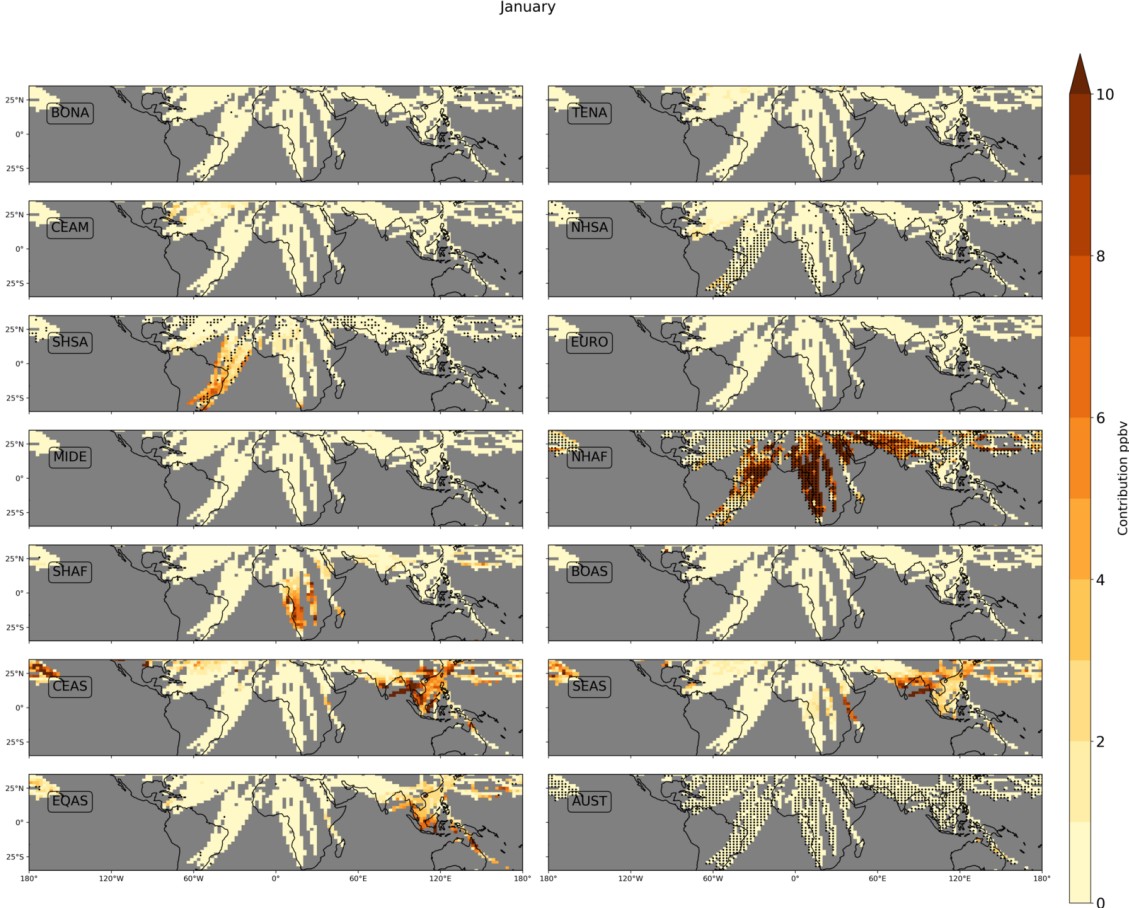

**Figure A4.** Mean CO contribution (in ppb) per source region in the tropical UT (300–185 hPa) averaged from 2002–2019 for January. The hatched part indicates BB as the dominant source of CO.



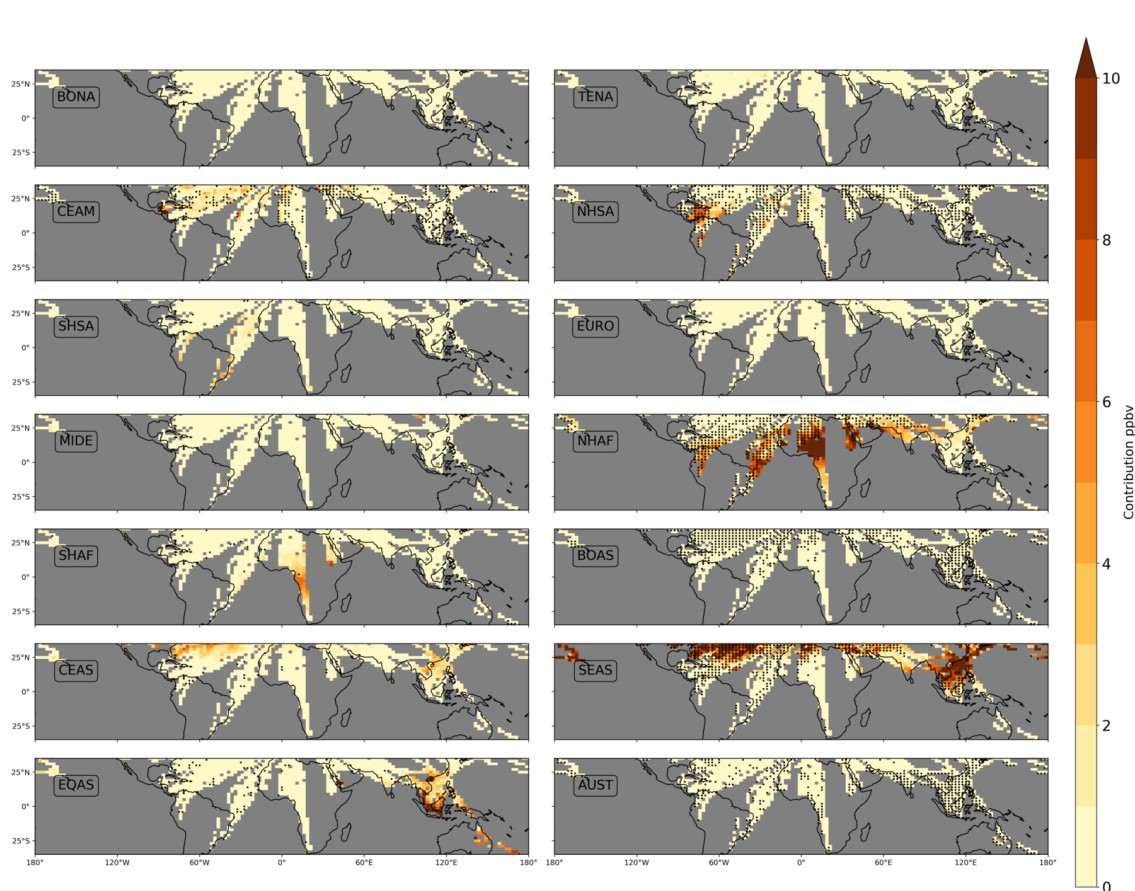

**Figure A5.** Same as Fig. A4 for April.





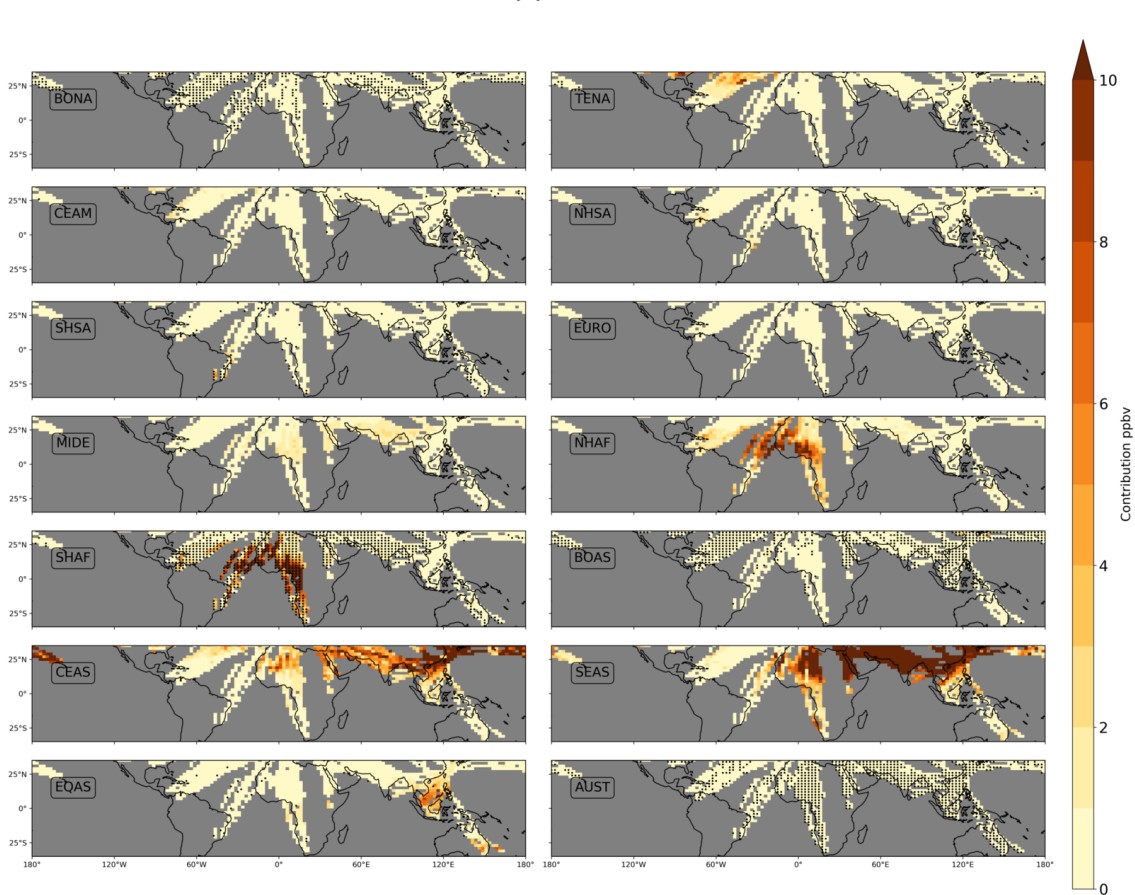

**Figure A6.** Same as Fig. A4 for July.





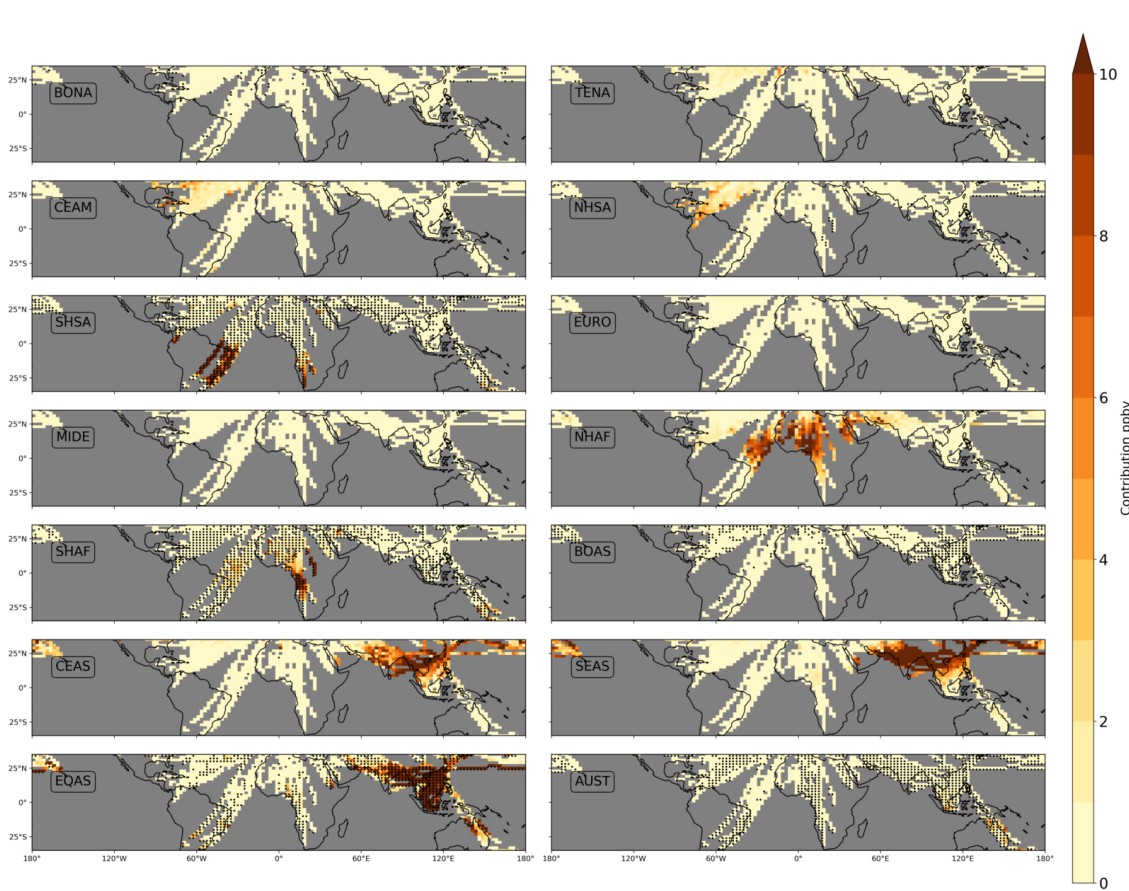

**Figure A7.** Same as Fig. A4 for October.



*Data availability.* The IAGOS data are available on the IAGOS data portal (https://doi.org/10.25326/20). The SOFT-IO v1.0 products are part of the ancillary products of IAGOS central database (https://doi.org/10.25326/2; https://doi.org/10.25326/3). The SOFRID-O$_3$ data are freely available on the IASI-SOFRID website (http://thredds.sedoo.fr/iasi-sofrid-o3-co/, last access: 8 June 2022; SEDOO, 2014).

*Author contributions.* MT, BS and BB designed the research. All the co-authors contributed to acquisition of data. MT analysed the data. MT, BS and BB interpreted the data. MT drafted the article. MT, BS and BB revised the article. VT and HC commented the article.

*Competing interests.* The authors declare that they have no conflict of interest.

*Acknowledgements.* We acknowledge the strong support of the European Commission, Airbus and the airlines (Deutsche Lufthansa, Air France, Austrian, Air Namimbia, Cathay Pacific, Iberia, China Airlines and Hawaiian Airlines) that have carried the MOZAIC or IAGOS
equipment and performed the maintenance since 1994. IAGOS has been funded by the European Union projects IAGOS–DS and IA-GOS–ERI. Additionally, IAGOS has been funded by INSU-CNRS (France), Météo-France, Université Paul Sabatier (Toulouse, France) and Research Center Jülich (FZJ, Jülich, Germany). The IAGOS database is supported in France by AERIS (https://www.aeris-data.fr). IASI is a joint mission of EUMETSAT and the Centre National d'Etudes Spatiales (CNES, France). The authors acknowledge the CNES for financial support for the IASI activities.
*Financial support.* This research has been supported by Bonus Stratégique programme at Université Paul Sabatier Toulouse III who funded the first author's doctoral position.





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
