# Peer review of "Tropical tropospheric ozone and carbon monoxide distributions: characteristics, origins and control factors, as seen by IAGOS and IASI"

_Atmospheric Chemistry and Physics, 2022_

## Author Comment (AC1)

Atmos. Chem. Phys. Discuss., referee comment RC1

https://doi.org/10.5194/acp-2022-686-RC1, 2022

[Figure]

**Response to Anonymous Referees on acp-2022-686**

**Tropical tropospheric ozone and carbon monoxide distributions: characteristics, origins and control factors, as seen by IAGOS and IASI**

Maria Tsivlidou et al.

We thank the Reviewers for their comments and suggestions. Below we provide our answers to their specific comments and the details of the changes made to the revised manuscript.

**Response to Anonymous Referee 1**

*> Comment 1: Paragraph 2.1. The term "accuracy" should not be used in a quantitative way. Do the author refer to the systematic error, instead (since the precision is quantified)?*

> The measurement uncertainties are considered to be caused by random error (Nedelec et al., 2015).
>
> We rephrased the sentence in lines 95-96, page 4 of the revised manuscript as follows:
>
> "with  an overall uncertainty of ±2 ppbv ±2 % (±5 ppbv ±5 %) and a time resolution of 4 (resp. 30) seconds  (Nedelec et al., 2015)."

*> Comment 2: Pag 5, line 127. "The RSE is defined as the \*fraction\*"...This is unclear. Do you mean \*ratio\*?*

> Yes. Corrected.

*> Comment 3: Figure 2. The stripes visible for ozone need more discussion. These discontinuities do not represent geophysical features but uncertainty in the data products. For some regions discontinuities of 5-10 ppb were visible which could be considered a quantification of the*

*uncertainty related to the use of the different a priori profiles. Are these spatial discontinuities visible also in other vertical layers?*

The stripes are visible in other vertical layers. However, Barret et al. 2020 validated SOFRID-O3 v1.6 (considering a single a priori profile) and v3.5 (considering a dynamical a priori profile) using ozonesondes for the period 2008 to 2017. Despite the presence of the stripes, their study clearly documents the improvements in O3 retrievals for v3.5, in terms of variability and correlation in most latitudinal bands. The biases between SOFRID and sondes are also lower for v3.5 than for v1.6. This indicates that SOFRID v1.6 provides a smooth O3 distribution with larger biases, while in SOFRID v3.5 the effect of the a priori profile is visible but the biases are lower. Therefore, despite the presence of stripes the use of a dynamical a priori profile significantly improves the O3 retrievals.

Figure 2 in section 3.1 "O3 and CO over the Northern and Southern Tropics" of the original manuscript has been removed from the revised manuscript in order to take into consideration Reviewers #2's comments 1 and 2. However, in order to take into account Reviewers #1 comment we added the lines 190-193 in page 8 in the revised manuscript to discuss the stripes visible on the ozone distributions:

"The stripes along the 10° latitude bands in IASI O3 maps (Fig. 2 e–h and q–t) are due to the use of a dynamical a priori profile, resulting in discontinuities between adjacent latitude bands with different a priori profiles. Nevertheless, the use of a dynamical a priori profile largely improves the retrieved O3 profiles in terms of biases, variability and correlation relative to the previous version based on a single a priori profile (Barret et al., 2020). "

*> Comment 4: Moreover a kind of noise that was not discussed in the paper was visible for ozone over the desertic regions of the northern Africa and Arabian peninsula. Since especially the first feature affects the regional average values, I would like to see a discussion about potential impact to the obtained results.*

As mentioned just above, Figure 2 of the original manuscript is removed in accordance to Reviewers #2 comment 1 and 2. However, Reviewer #1 is right, the noise over the desertic regions is not discussed in the original manuscript. Our results related to northern tropical Africa are focused on latitudes southern than the desertic african region. Since IASI distributions are analyzed in a complementary way to IAGOS, we are mostly focused on Western Africa where most IAGOS sites are located. Despite some missing values over the Arabian peninsula, the IASI O3 distributions and seasonal characteristics over Arabia agree qualitatively with IAGOS.

For instance, IASI captures the O3 maximum in July also seen by IAGOS, and documented by previous studies (e.g. Barret et al., 2016; Park et al., 2007).

In order to make this issue clearer we added the following statement in lines 188-189 in page 8 of the revised manuscript:

"The measurements above Northern Africa are erroneous, due to retrieval problems in the presence of desert ground with sand emissivity interfering with the O3 signature (Boynard et al., 2018)."

> *Comment 5: Page 23, line 489: please specify AMA*

The abbreviation is now presented where the Asian monsoon anticyclone (AMA) is first mentioned in the text in line 335, page 16 of the revised manuscript. The new sentence is:

"The resulting enhancement of CO in the UT within the Asian monsoon anticyclone (AMA) analyzed in Park et al. (2008) and Barret et al. (2016) is clear from IASI (Fig. 3k)."

> *Comment 6: The analysis was mostly based on the IAGOS profiles at selected locations. Along the manuscript, the authors nicely discuss also the intra-regional differences observed at different sites in the same regions and attributed observed differences. Potential limitations for the results upscaling should be also highlighted in the Conclusion section.*

The fact that IAGOS  measurements are representative of the urban background a few hundreds meters above the ground, and of the regional scale at higher altitudes in the lower troposphere has been addressed in comment 12 of Reviewer #2.

In addition throughout the paper, we take advantage of  IASI CO and O3 global distributions in order to complement the limited spatial coverage of IAGOS (e.g. line 273 page 14 and line 299 page 15 of the original manuscript).

Text has been added in the conclusions, in lines 589-593, page 28 of the revised manuscript:

"Furthermore, IASI, which provides global daily O3 and CO distributions with a coarse vertical resolution, allows us to complement IAGOS observations on the global scale over the data sparse tropical band. Throughout the paper we have shown that the anomalies detected by IAGOS are often also detected by IASI at the regional scale."

References

Barret, B., Emili, E., and Le Flochmoen, E.: A tropopause-related climatological a priori profile for IASI-SOFRID ozone retrievals: improvements and validation, Atmos. Meas. Tech., 13, 5237–5257, https://doi.org/10.5194/amt-13-5237-2020, 2020.

Barret, B., Sauvage, B., Bennouna, Y., and Le Flochmoen, E.: Upper-tropospheric CO and O3 budget during the Asian summer monsoon, Atmos. Chem. Phys., 16, 9129–9147, https://doi.org/10.5194/acp-16-9129-2016, 2016.

Boynard, A., Hurtmans, D., Garane, K., Goutail, F., Hadji-Lazaro, J., Koukouli, M. E., Wespes, C., Vigouroux, C., Keppens, A., Pommereau, J.-P., Pazmino, A., Balis, D., Loyola, D., Valks, P., Sussmann, R., Smale, D., Coheur, P.-F., and Clerbaux, C.: Validation of the IASI FORLI/EUMETSAT ozone products using satellite (GOME-2), ground-based (Brewer–Dobson, SAOZ, FTIR) and ozonesonde measurements, Atmos. Meas. Tech., 11, 5125–5152, https://doi.org/10.5194/amt-11-5125-2018, 2018.

Park, M., Randel, W. J., Gettelman, A., Massie, S. T., and Jiang, J. H.: Transport above the Asian summer monsoon anticyclone inferred from 1015 Aura Microwave Limb Sounder tracers, Journal of Geophysical Research: Atmospheres, 112, https://doi.org/10.1029/2006JD008294, 2007.

**Response to Anonymous Referee 2**

*General comments:*

*> Comment 1: I find the paper extremely hard to follow. It is extremely long (41 pages excluding references!), and would really benefit from a clear structure in the way the results are discussed. I would recommend using subsections for each cluster so that the reader can know straight on what exactly is discussed (e.g., which cluster is discussed? LT, MT or UT? Which month? O3 or CO? Measured mixing ratios or SOFTIO contribution? etc.).*

In order to consider this comment we decided to remove section 3.1 (see comment below) and to rewrite parts of sections 3.2 and 3.3 of the original manuscript. In the revised version of the manuscript, we also improved the structure of the discussion by re organizing the information for the regions (e.g. over Africa the maxima of O3 and CO in the respective dry season and minima of O3 and CO in the transition period from the wet to the dry season are discussed together) in order to make the paper easier to read. In addition, we improved sections 3.1 and 3.2 removing redundant statements. All these modifications improve the clarity of the paper. At the end, the paper has been shortened by 3 pages.

*> Comment 2: In my opinion, section 3.1 does not bring anything to the paper. Discussing IAGOS data as done in the following sections is already plenty enough for one paper. In addition, the section 3.1 reads as purely speculative, and raises questions that are actually answered later on in the manuscript. What is the point of that?*

In order to take Reviewer #2's comment into account and to reduce paper length, we have decided to remove section 3.1 "O3 and CO over the Northern and Southern Tropics". The initial idea was actually to raise question and answer them later in the manuscript, but Reviewer #2 is right this section was not bringing new results. In the revised version of the manuscript, section 3.2 becomes section 3.1.

*> Comment 3: There are a lot of acronyms throughout the paper, especially with regard to the source regions of CO. It makes sense to use those in the figures, but the authors should consider righting the name of contributing regions in full to avoid having to refer to Figure 1 all the time. In addition, NH and SH are often used in the paper as referring to the northern tropics and southern topics. This is really confusing as NH and SH are classically used for the whole northern and southern hemispheres. I would recommend using different abbreviations (NT and*

Initially all these acronyms were present to avoid an even longer document. However, as asked by Reviewer#2, the acronyms for the source regions have been replaced by the name of the regions, and NH and SH referring to the tropics has been changed to NT and ST throughout the revised manuscript. The acronym PBL has been replaced to planetary boundary layer in the revised manuscript.

*> Comment 4: The authors separate the troposphere in 4 different layers (surface, LT, MT and UT) but, unless I missed something, never define those layers in the manuscript. It complicates the discussion, as these definitions drive the interpretation of the data. This should be clarified at the beginning of the manuscript.*

We added a more detailed description of the low, mid and upper tropospheric layers used in the study in the revised manuscript.

We had to introduce some new acronyms as the three tropospheric layers are defined differently for IASI and IAGOS/SOFT-IO due to their different vertical resolution. For IASI, the three tropospheric layers correspond approximately to its independent pieces of information: low troposphere defined between 900-700 hPa ($LT_{iasi}$), mid troposphere between 600-400 hPa ($MT_{iasi}$) and upper troposphere between 290-220 hPa ($UT_{iasi}$).

For IAGOS and SOFT-IO vertical profiles, we define the tropospheric layers related to different dynamical regimes: the low troposphere between the surface and 750 hPa (LT) corresponding roughly to the planetary boundary layer, the mid troposphere above up to 300 hPa (MT) and upper troposphere above up to 200 hPa corresponding to the beginning of convective detrainment. Last, the upper tropospheric layer corresponding to the IAGOS cruise phase between 300-185 hPa is defined as $UT_{cruise}$.

The discussion in the results is mostly based on the IAGOS and SOFT-IO profiles. For this reason, we gave the acronyms LT, MT and UT. We believe that annotating an index (e.g. iasi, cruise) in the 'LT', 'MT' and 'UT' is the most straightforward way for the reader to understand the tropospheric level we are referring to (low, mid or upper) and to distinguish between the concrete pressure ranges depending on the instrument used. In addition, the pressure range of IAGOS during cruise is mentioned in different parts of the original (and revised) manuscript. As a result, we define a separate acronym to make clear the definition of the pressure range and avoid repeating and clarifying it every time it is mentioned.

The definitions of the layers are specified in the revised manuscript:

Lines 71-72, page 3 of the revised manuscript for IAGOS cruise phase:

"Using equipped commercial aircraft, IAGOS measures vertical profiles, along with the (lower part) of the tropical upper troposphere between 300–185 hPa ($UT_{cruise}$))."

Lines 198-202, page 8 of the revised manuscript for IASI:

"We focus on daytime measurements when larger thermal contrast between the surface and the atmosphere results in increased sensitivity of the instrument (Clerbaux et al., 2009) and on pressure levels corresponding approximately to the independent pieces of information: low troposphere defined between 900–700 hPa ($LT_{iasi}$), mid troposphere between 600–400 hPa ($MT_{iasi}$) and upper troposphere between 290–220 hPa ($UT_{iasi}$))."

Lines 169-170 in page 8 of the revised version, for SOFT-IO:

"We performed an evaluation of SOFT-IO for the lower troposphere (LT, surface-750 hPa), the mid troposphere (MT, 750-300 hPa), and upper troposphere (UT, 300-200 hPa). "

> *Comment 5: A crucial piece of information is missing from the paper in my opinion, that bears on how the data should be interpreted. The IAGOS flights used by the authors date back from 1994 and 2002 for O3 and CO, respectively. However, there is no information on how the IAGOS flights used in the current analysis are spread throughout this time period. For a given cluster, are there always the same number of flights per month throughout the time period? Do all clusters span the same time range, or are some timeseries shorter than others? I assume that all clusters are different in the time range with data availability. If so, how would this impact the comparison of clusters, and the source attribution?*

This information was missing from the original manuscript. The availability of the measurements spans different time periods for each site/cluster, as shown in Figs. 1 and 2 below for CO and O3 respectively. To ensure the representativeness of our O3/CO climatology, we applied a statistical method based on other studies treating sparse spatiotemporal measurements such as ozonesondes (e.g. Logan, 1999) and IAGOS data (e.g. Sauvage et al., 2005). Logan (1999) determined that in the tropics, 20 ozone soundings were required below 500 hPa for reliable climatologies. Sauvage et al. 2005 found that the ozone monthly means over each site should be representative with 15 measurements over Gulf of Guinea, 13 over East Africa and 8 over Central Africa. In our case, as detailed in the manuscript, we use a 10% threshold for the relative standard error to determine the minimum number of flights required for representative climatologies. In order to make clearer the data

availability of IAGOS, we modified line 103 in page 4 in the revised manuscript:

"Good consistency in the measurements between the two programs (hereafter referred to as IAGOS) (Nédélec et al., 2015; Blot et al., 2021) leads to IAGOS temporal coverage of 26 (resp. almost 20) years for O3 (resp. CO), depending on the availability of the flights."

Figures 1 and 2 are added in the Supplementary material (Figures S4 and S5) in pages 7 and 8 and we added lines 123-125 in page 5 of the revised manuscript:

"The temporal availability of the measurements differ for each site and cluster, as it depends on the flight schedule of the aircraft (see Figs. S4 and S5). For this reason, to determine a reliable climatological profile, we need to assess the statistical significance of the data."

We also added the line 588 in page 28 of the revised manuscript:

"IAGOS O3 and CO observations since 1994 and 2002 respectively, were used (when available) in order to analyse vertical profiles over 20 tropical sites, along with the (lower part) of the upper tropical troposphere. "

[Figure]

Figure 1: Availability of IAGOS CO measurements for each site/cluster from 2002 to 2020.

[Figure]

Figure 2: Same as Fig.1 for IAGOS O3 measurements from 1994 to 2020.

> Comment 6: There is a number of flights provided for each cluster in Figures 4, 6,7 and 8, but it is really unclear from the captions of these figures what this number refers to. Is that the average number of flights

*per month for the entire time period considered here? Or is that the total number of flights for all months in the given time period?*

This was not clear in the original version. The numbers in Figures 4, 6, 7, 8 refer to the total number of flights for each month in the given time period. We adjusted the caption of Figure 3 page 11 in the revised manuscript as follows:

"The annotated numbers correspond to the **total** number of flights per month for the IAGOS period, given in the same colour as in the legend."

*> Comment 7: If so, how do you quantify the uncertainty due to the low number of flights in some places (e.g., Mumbai or AbuDhabi)?*

As stated by Reviewer #2, there are cases such as Abu Dhabi above 10 km and Mumbai with lower number of flights than required to be statistically significant (see comment 5). We decided to include these sites in the paper as the number of flights is often close to the threshold and they can provide valuable information. In addition, when number of observations, is much lower than the threshold we added caution statements as in lines 312-313, page 16 and 379-380, page 19 respectively, of the revised manuscript:

"CO anomalies over Mumbai are also caused by transport of AN emissions from the Middle East (36%) in the LT, and NH Africa (30%) in the MT (Fig. 5a and b). In the UT, the impact of NH African (AN and BB) emissions dominates over Mumbai (54 \%) and Hyderabad (50 \%). It has to be noted that the number of profiles over Mumbai (6) and Hyderabad (19) are lower than the threshold established for representativeness (see Section 2.1.1)."

"Note that only limited number of profiles are available over Abu Dhabi above 10 km and Khartoum in April."

*> Comment 8: There is no discussion at any point of the uncertainty in the contribution of sources to CO using the SOFTIO software. Throughout the paper, relative or absolute contributions of anthropogenic and biomass burning emissions to measured CO are provided, but with no associated uncertainties. I would imagine that this uncertainty is quite high (stemming from the uncertainty in emission inventories, uncertainty in back trajectory computation, etc.) and needs to be quantified.*

*At the very least, I would like to see on all figures (the ones that report CO contributions from AN and BB sources like figures 4 and 5) the proportion or absolute amount of CO anomalies that is unaccounted for. If and when that amount is larger than the contributions actually accounted*

*for by SOFTIO, the authors need to discuss how much confidence can be put in the interpretation of those results.*

As detailed in the SOFT-IO description and evaluation paper (Sauvage et al 2017, see their discussion in section 4) and in papers using it (e.g. Petetin et al., 2018b; Lannuque et al., 2021), SOFT-IO uncertainties are due to either an underestimation of emissions by the inventories, and/or to a misrepresentation of the plumes in the model (in horizontal or vertical space and time), inherent to the uncertainties on the meteorological fields or parameterization of the transport (turbulence, convection) in the model (at the same location and time as the in situ measurements). These are the same uncertainties in all models, even when chemistry is considered and simulated. Note that studies evaluating CO simulated by chemistry and transport models always underestimate CO compared to IAGOS measurements, especially in CO anomalies (e.g. Cussac et al., 2020 or the evaluation of the Monitoring Atmospheric Composition and Climate reanalysis performed with IFS model coupled with chemical model such as MOZART, in the frame of the Copernicus Atmosphere Monitoring Service, see http://www.iagos.fr/cams/). More generally, the overall performance of the global chemistry transport models in simulating CO is discussed in IPCC (2021). Even though the models capture the spatial distribution of the observed CO concentrations, they show regional biases up to 50 % (IPCC, 2021 and references therein). For instance, models are persistently biased in the Southern hemisphere and in the tropics, particularly over polluted regions such as India and East Asia. As a result SOFT-IO has to be seen as a tool to perform source attribution and to quantify the relative part of a source influence to another, but not as a tool perfectly able to simulate the exact CO concentrations, but this is a problem of most of the models in CO anomalies.

The overall uncertainty of SOFT-IO is given by the absolute difference between simulated and observed CO anomalies. As for all models, it is quite impossible to give exact quantification of each individual source of uncertainty as they are all related. However, an estimation of the most uncertain factor can be given. For this reason, Sauvage et al. (2017) have already extensively evaluated the performance of SOFT-IO relative to observed CO anomalies by IAGOS for: i) case studies of anthropogenic and biomass burning pollution events, and ii) for the entire IAGOS database available at the time of the study. As discussed in the original manuscript, Sauvage et al. showed that SOFT-IO detects 95% of the observed CO plumes on average, and that the main uncertainty comes from anthropogenic inventories rather than meteorological fields or biomass burning inventories, testing various state of the art anthropogenic and biomass burning emissions widely used in the community.

More precisely, Sauvage et al.(2017) conducted sensitivity analyses

to assess the dependency of SOFT-IO to input parameters (different meteorological field analysis and emissions inventories). Conclusions of their uncertainty estimation are the following:

SOFT-IO is not very sensitive to:

  i) the resolution of the meteorological input data,

  ii) the biomass burning global inventory (similar results between GFED4 and GFAS v1.2).

SOFT-IO is sensitive to:

  i) the altitude injection of biomass burning mostly for boreal fires,

  ii) regional biomass burning emission inventory but for high latitude regions, not relevant for our study,

  ii) AN emissions, with important regional differences depending on the emissions inventory.

All the SOFT-IO uncertainties have been clearly discussed in previous papers (e.g. Sauvage et al., 2017; Petetin et al.,2018b) and the evaluation of SOFT-IO is outside of the scope of this paper. However, in order to take into account Reviewer #2 concern, we performed a similar evaluation as in Sauvage et al., 2017 over the tropical sites and a longer period (2002-2020). We found very few sensitivity to meteorological parameters (analysis versus reanalysis) and back trajectories computation (time calculation for transport and sub grid processes), and to biomass burning emissions (GFAS vs GFED4) and altitude injection.

In our study, we use GFAS that gives similar results as GFED4 but provides altitude of fire injection for each fire event. We use the Community Emissions Data System (CEDS2) AN emissions (McDuffie et al., 2020) which is a state of the art anthropogenic inventory broadly used by the community, rather than MACCity which is based on projection after 2000. CEDS2 incorporates updated activity data for combustion and process-level emission sources, and updated scaling inventories. Therefore, CEDS2 inventory is expected to better represent the anthropogenic emissions. To investigate further the uncertainties related to AN emissions, we compared the CO emissions from available inventories (CEDS and MACCity) over tropical regions (South Brazil, West Africa, Southern Africa, India, East Asia and SouthEast Asia). There are significant differences in the emissions fluxes. These differences which display a seasonal pattern are larger over the most polluted regions and smaller over South America and Southern Africa. As shown in Sauvage et al. (2017), the differences are low for the BB inventories. In contrast, the large differences among the AN inventories brings larger uncertainties in

SOFT-IO calculations.

Comparing SOFT-IO using MACCity and CEDS2, we found absolute mean differences of

27% in the LT, 16% in the MT, 10%  in the UT.

This confirms that anthropogenic emissions are responsible for important uncertainties in SOFT-IO.

Our SOFT-IO simulations showed similar detection frequency of CO anomalies as Sauvage et al. 2017. SOFT-IO underestimates the anomalies by 10 ppb on average in the MT and UT and 45 ppbv in the LT.

In order to take into consideration the suggestion of Reviewer #2, we included the lines 160-162 in page 7 and lines 169-176, page 8 the revised manuscript:

"As detailed in their study, SOFT-IO uncertainties and biases are mostly due to uncertainties in emission inventories, and to a lesser extent to uncertainties concerning the meteorological fields and FLEXPART transport parameterizations (turbulence, convection)."

"For this reason, it is computed as the monthly climatological median CO of a remote area away from polluted regions, in the upper troposphere (300-185 hPa, during the whole study period 2002--2020) (Sect. S4 for more details). We performed an evaluation of SOFT-IO for the lower troposphere (LT,  surface-750 hPa), the mid troposphere (MT, 750-300 hPa), and upper troposphere (UT, 300-200 hPa). Our simulations detect CO anomalies at the same rates as Sauvage et al. 2017. On average, SOFT-IO underestimates the observed CO anomalies by 10 ppb in the MT and UT, and by 45 ppb in the LT. A sensitivity test has shown absolute differences of 27% in the LT, 16% in the MT and 10% in the UT between SOFT-IO simulations using AN emissions from MACCity and from CEDS2. This clearly highlights the large uncertainty stemming from uncertainties in AN emissions."

As stated by Referee#2, another source of uncertainty in comparing observed CO anomaly to SOFT-IO calculations can be related to background CO. This point is discussed in reply to comment 9 below.

*> Comment 9: In the same vein, it became really hard for me to trust the contributions calculated by SOFTIO given the significant gap between those numbers and the actual mixing rations actually measured by IAGOS. First, the authors should explain more explicitly how background CO is calculated. Is it one common value for all sites? Is there a vertical*

*resolution of the background?*

In the revised version we give more details on its calculations in lines 9-16, page 9 in the supplementary material of the revised manuscript.

The background (BG) was computed as the median of the UT (300-185 hPa) for each region and each month as shown in Fig. 3 below. For the NT African and AEA (resp. ST African) clusters, the BG area is located in the Northern (resp. Southern) part of Africa, far from fires injection through ITCZ. For the Asian clusters, the BG is over the tropical Pacific, and for the South American clusters over the North Atlantic. The computations of BG for each month (see Table below) allows to account for the CO seasonality.

However, the selection of the background mixing ratio is rather subjective, as discussed in Parrish et al. (2012). The UT median is generally used to assess baseline concentrations of atmospheric species that are not influenced by recent pollution as they have the characteristics of well-mixed air masses from different origins (Gressent et al., 2014).

In order to evaluate the impact of the definition of the BG, we followed an alternative definition using the median CO mixing ratio between 600-300 hPa but for each site. The comparison between this background and the UT background used in the paper showed 2.5 to 60 ppb differences, but most importantly no difference in the anomaly source attribution and in the relative contributions.

We added the lines 174-176 in page 8 of the revised manuscript:

"Another source of uncertainty comes from the definition of background CO. In order to assess this source of uncertainty, we used the 600-300 hPa median CO mixing ratio as background for each site. The differences between the two backgrounds are within 2.5-60 ppbv. Nevertheless, using the alternative background did not make any difference in the anomaly source attribution and in the relative contributions."

[Figure]

Figure 3: IAGOS monthly mean CO distribution averaged for the period 2002 to 2019. The red boxes indicate the location of the regions taken into account for the computation of the background CO mixing ratio.

|  | Northern Atlantic | Northern Africa | Southern Africa | Pacific |
|---|---|---|---|---|
| January | 78 | 83 | 82 | 85 |
| April | 91 | 98 | 71 | 77 |
| July | 77 | 87 | 70 | 66 |
| October | 75 | 81 | 74 | 77 |

Table 2: Background CO mixing ratio in ppb for each month and each region.

> Comment 10: Then, for most sites in this study SOFTIO only explains a

*(small) part of the calculated CO anomalies, leaving in some cases hundreds of ppb of CO unaccounted for. To me, this issue needs to be discussed in more detail, since the whole paper relies on the assumption that SOFTIO can explain "95% of the CO anomalies" as stated by the authors.*

The sentence "SOFT-IO can explain 95% of the CO anomalies" came from Sauvage et al. 2017 and refers to the number of the simulated anomalies, but not the intensity of the anomalies. As explained previously in comment 8, our calculations over the tropics show the same order of detection, and absolute differences of 10 ppb in the MT/UT, and up to 45 ppbv in the LT. This has been detailed in Sauvage et al 2017. As it seems unclear we better explain it in the revised version in line 162 and page 7 and rephrase "SOFTIO can explain "95% of the CO anomalies" by "SOFT-IO can simulate 95% of the observed number of anomalies".

*> Comment 11: In addition, the authors invoke throughout the paper that the shortcomings of SOFTIO are due to underestimated AN emission inventories. Why would that be the case, rather than underestimation of BB emission inventories for instance? Another thought would be that the missing CO (unaccounted for) could be within the uncertainties of the SOFTIO calculated contributions, but this is hard to conclude on in the present paper as uncertainties are not discussed. In any case, if there is so much uncertainty in the AN emission inventories in the first place, that would impact SOFTIO calculated contributions everywhere and throughout the tropospheric column, right? So again, uncertainties should be discussed.*

When CO anomalies are attributed to both AN and BB emissions (e.g. during the dry season of each hemisphere), it is of course not possible to distinguish which source is responsible from the SOFT-IO.

Nevertheless, as mentioned in the reply to comment 8 the sensitivity tests we performed over the Tropics using different inventories, additionally to the ones performed by Sauvage et al., 2017, highlight much larger SOFT-IO differences stemming from AN than from BB inventories (see reply to comment 8 for details and manuscript modifications page 8/line 174).

Furthermore, we comment the performance of SOFT-IO when the CO anomalies are attributed entirely to one source (AN) and to one source region. For instance, in the case of Africa (NH and SH) (line 261, page 14 and line 316, page 16 respectively in the original manuscript) and South America (line 445, page 21 of the original manuscript), we discuss the underestimation of the AN emissions during the transition periods, when the fires are suppressed. During this seasons, AN emissions cause the CO anomalies (as shown by

SOFT-IO attribution) because there are no or few fires, as can be seen by the fire counts based on MODIS for the period 2002 to 2012 in Yamasoe et al. (2015) (their Fig. 7) and discussed in Giglio et al. (2006). Text has been modified in lines 240-245 in page 13 of the revised manuscript:

"Over the NT African clusters, secondary CO and O3 maxima are observed below 4 km (Fig. 3 panels 1a–1c and 2a–2c) during the transition from the NH dry to wet season (April). LT CO mainly comes from local AN emissions (Figs. 3 panel 3b; A1 panels 1b and 2b). The fact that SOFT-IO attributes approximately 80 ppbv of CO to local AN emissions (Figs. 3 panel 3b; A1 panels 1b and 2b), while the observed anomaly reaches 200–250 ppbv and no or few fires are detected by MODIS (Yamasoe et al., 2015; their Fig.7), indicates an underestimation of the NH African AN emissions."

> *Comment 12: One of the main conclusions of the paper is that anthropogenic emissions contribute much more to CO anomalies measured by IAGOS compared to BB burning. Is that really surprising considering that the large majority of the sites studied here are megacities, with several million inhabitants, and therefore with overwhelming local AN emissions? Imagine IAGOS airports were located in the middle of fires, wouldn't you get the opposite results? It is not clear to me how you can generalize your findings to the whole tropical band.*

It is true that IAGOS sites are mostly located over megacities but this does not mean that IAGOS data are not representative of larger areas surrounding the cities. Indeed, Petetin et al. (Elementa 2018a) compared CO and O3 IAGOS measurements with the surrounding urban background stations. As explained in their study, urban background stations are located within the city or its suburbs but away from direct influence of traffic and urban emissions. Their results showed that O3 and CO mixing ratios measured by IAGOS in the first few hundred meters above the surface have similar characteristics as the ones at the surrounding urban background stations in terms of mixing ratio distribution, seasonal variations and trends. Furthermore, at higher altitudes in the lower troposphere, they showed that IAGOS data are representative of more distant regional surface stations from the Global Atmospheric Watch (GAW) network, outside of direct anthropogenic influence. This is related to the fact that between 1 to 5 km altitude, the aircrafts are 10 to 120 km away from the airports (see their Fig.1). In order to consider this discussion, the conclusion section has been modified as follows:

We added the lines 589-593, page 28 of the revised manuscript:

"One limitation of our study is the rather limited spatial coverage of

IAGOS profiles to limited locations. However, according to Petetin et al., 2018a, a few hundreds meters above the ground, these measurements are representative of the urban background and of the regional scale at higher altitudes in the lower troposphere."

We removed lines 785-787, page 33 of the original manuscript in order to avoid making general statements about the whole tropics.

*Detailed comments:*

*> l.35 Change to (e.g., Edwards et al. 2006) as this was not the first paper to use CO as a pollution tracer.*

Done

*> l.37-38 It would be good to include a quantification of the respective sources of CO here, and an adequate reference for this statement on CO being primarily emitted by anthropogenic emissions. I imagine you are referring to emission inventories? Please be explicit.*

Text has been modified accordingly in lines 36-39, page 2 of the revised manuscript:

"CO is primarily emitted by incomplete combustion, thus by anthropogenic (AN) and biomass burning (BB) sources (Galanter et al., 2000; Granier et al., 2011), with estimated contributions between 450-600 and 350-600 Tg CO yr−1 respectively (Lamarque et al., 2010; van der Werf et al., 2006). Its secondary sources include oxidation of VOCs and methane, with contributions in the range between 450–1200 and 600– 1000 Tg CO yr−1 (Stein et al., 2014)."

*> l.41-42 Bourgeois et al. (2020) recently presented a global-scale distribution of O3 in the remote troposphere based on aircraft observations. Probably a good idea to acknowledge that here.*

Text has been added in line 41-42, page 2 in the revised manuscript:

"Based on aircraft observations, Bourgeois et al. (2020) recently presented a global-scale distribution of O3 in the remote troposphere."

*> l.42 "inadequate" doesn't sound like the right word here. Maybe use "due to the paucity of observations". Also, please resolve the conflict: the troposphere above "developing countries in the tropics" is not "remote".*

*The remote troposphere applies to air far from emission sources, i.e., far from land.*

Text in lines 42-43 in page 2 of the revised manuscript has been modified accordingly:

"However,  uncertainties still remain in the global O3 distribution and sources of precursors due to  paucity of observations in the  free troposphere, especially over developing countries in the tropics (Gaudel et al., 2018; Tarasick et al., 2019)."

*> l.61-62 I disagree with this statement. Field campaigns in the tropics have provided invaluable insights on both the atmospheric chemistry and dynamics of this region, back from the 80's. The spatial coverage of these campaigns, especially airborne, is much larger than that of ozonesondes for instance, which map out O3 columns at very specific locations. I think that the least you can do is acknowledge the value of these campaigns and also name them (CAST, ATTREX, SAFARI, the NASA GTE campaigns (PEM-TROPICS), etc.).*

Text has been modified in lines 61-64 in page 3 of the revised manuscript:

"On short time scales, several international field campaigns have been carried out in the tropics, yielding measurements of various species over Africa (from TROPOZ 1987 to CAFE-Africa), Asia (from INDOEX to EMerge-Asia), South America (Cite-1/2/3, TROCCiNOX) and the tropical Pacific (from PEM-WEST-A/B to CAST/CONTRAST/ATTREX and Atom). The campaigns have provided invaluable insights on the atmospheric chemistry and dynamics of the tropical region. "

*> l.64 Why would ozonesondes under-represent the tropical upper troposphere? If anything, the altitude ceiling of ozonesondes is much higher than that of IAGOS or other airborne measurements.*

Indeed ozonesondes measure vertical profiles of O3 at higher altitudes than airborne measurements. We meant that ozonesondes provide UT measurements above the location of the site, which is limited compared to the IAGOS data during cruise phase. However, it is true that all these dataset have a complementary role in the monitoring of the atmospheric composition, especially over sparse-data regions like the tropics.

Text has been modified in lines 65-68 page 3 in the revised manuscript:

"On greater timescales, the Southern Hemisphere ADditional OZone Sounding (SHADOZ) program (Thompson et al., 2003a) provides long-term O3 observations over the tropics using ozonesondes since 1998. These measurements  offered a better understanding on vertical distribution and trends of tropical O3 (e.g. Thomspon et al., 2021)"

*> l.69 This is a bit misleading. Yes, you have a long time-series from IAGOS flights, but you have strong heterogeneity in how the flights are spread in time and space throughout these periods. If you are going to call out all the limitations of other research networks, then you should acknowledge that here.*

This comment is addressed in the comment just below.

*> l.59-74: These two paragraphs read as trying too hard to show that IAGOS is much better than the other research networks and field campaigns in the tropics. It may not be the intention of the authors, but I think that they should rather focus on showing that all infrastructures are complementary and the combination of all are necessary to fully understand the kind of research questions they are addressing.*

Text has been modified in line 69-72, page 3 of the revised manuscript:

"In a complementary way to these datasets, the IAGOS (In-service Aircraft for a Global Observing System; (Marenco et al., 1998; Petzold et al., 2015; Thouret et al., 2022) program has provided  O3 and CO measurements  over the tropics since 1994 and 2002 respectively. Using equipped commercial aircraft, IAGOS  samples vertical profiles at take off and landing, along with the lower part of the upper tropical troposphere at cruise altitude between 300 and 185 hPa (UTcruise). "

*> l.70 Madras is the fourth biggest city in India. Definitely not remote. Per*

*definition, IAGOS flies in and out of airports, so you expect most of their tropospheric profiles to be influenced by regional emissions. You cannot qualify IAGOS as a remote troposphere observation network.*

The fact that above a few hundred meters above the surface, IAGOS is representative of regional scale is discussed in detail in comment 12. However, it is true that Madras is not a remote location.

Remote has been removed.

*> l.83 All research should be novel. Please remove "for the first time"*

Done.

*> l.98-99 The horizontal resolution is the same during the ascend and descend phases and during the cruise phases?*

According to Petetin et al. (2018a), the vertical and horizontal speed vary depending on the ascent and descent time after/before take off/landing, and thus the vertical profile can be considered as semi-vertical with a maximum horizontal speed of 166 m/s. The horizontal resolution during ascent and descent ranges between 4-600 m for O3 (response time of the instrument 4s) and 30-5000 m for CO (response time 30 sec).

The cruising speed has a maximum value of 250 m/s (or 900 km/h). The horizontal resolution during cruise is 1 km (resp. 7.5 km) for O3 (resp. CO).

*> l.166 How do you define the lower part of the UT?*

We refer to the IAGOS cruise altitude at about 300–185 hPa or 9–12 km.

We modified line 72 in page 3 of the revised manuscript in order to make this definition clearer:

"It samples vertical profiles at take off and landing , along with the lower part of the upper tropical troposphere at cruise altitude between 300 and 185 hPa. "

*> l.117 How much of the data is discarded by applying this filter?*

In the tropics this filter does concern a negligible part (less than 2%)

of the data.

> *l.122 Please state the typical upper limit here*

To make this part clearer, we modified line 120 in page 4 of the revised manuscript as follows:

For the same time periods, the climatologies over the vertical are derived by averaging the data into 10 hPa pressure bins from the surface up to 200 hPa. We also applied  distance criteria of  300-km  around the IAGOS observational site, similar to Petetin et al., 2016. This way we reduce uncertainties due to possible horizontal heterogeneity in the measurements, as the aircraft keeps moving in the horizontal plane during ascent and descent.

> *l.125-137 An important missing information here is a table that shows the number of flights per month for each of the sites (or clusters) shown in Table 1, the time length of the time series, if there are flights for every month of every year, etc. Basically, the authors could expand Table 1 to include all the information that the readers need to assess the distribution of IAGOS flights in time and space.*

We have included two columns in Table 1 in page 6 of the revised manuscript with the total number of flights for O3 and CO for each site/cluster for the whole IAGOS period. Because of the length of the paper and for clarity reasons, we added the availability of IAGOS flights for O3 and CO measurements over the different clusters, in the supplementary material, in pages 7 and 8, in order to show the time length of the timeseries and the periods that are sampled for each site and cluster during the IAGOS period.

> *l.144 Does "intermediate" mean between the seasons? If so, why not take months during peak seasons?*

We mean the month during the peak season. Text has been modified in the revised manuscript:

"Instead, we analyse the O3/CO profiles and horizontal distributions  for months during the peak tropical seasons (January, April, July and October), to highlight seasonal patterns."

> *l.161 most regions, including the tropics?*

Yes, we confirmed the Sauvage et al. 2017 results with more tests over the tropics up to 2020, please see comment 8 for more detail.

> *l.162 most of the regions, including the tropics?*

Same as above.

> *l.166 Please specify where the remote area was located. You are only considering the UT to calculate the CO background. That does not seem right to me, since you will use this value to calculate CO anomalies throughout the tropospheric column and there is a vertical gradient in CO mixing ratios due to stratospheric air mixing. You should use the median CO value over a remote area averaged across the tropospheric column. Also, please provide a range of the CO background values thus obtained.*

The issues linked to the background definition have been discussed and addressed in reply to comment #9.

> *l.204-209 I don't understand these figures and the associated discussion. The discussion talks about NH and SH emissions of CO, but the figures do not show all of the NH and SH. The maps are truncated to a latitude band that seems randomly chosen and does not seem to correspond to the tropical band as defined by the authors early on in the manuscript.*

This comment refers to section 3.1 'O3 and CO over the Northern and Southern Tropics' of the original manuscript. In order to consider comments 1 and 2 , we decided to remove section 3.1 in the revised manuscript. Nevertheless, the acronyms NH and SH have been changed to Northern Tropics (NT) and Southern Tropics (ST) respectively when they refer to the tropics (as mentioned before in the general comment related to the acronyms).

> *l.210-214 Why speculate here on the sources of CO and O3 when you are actually answering this question later on in the manuscript using SOFTIO?*

Same as the comment just above.

> *l.220-225 This is not true. In this section, you only speculate, based on coincident maps of AN and BB emissions and CO and O3 concentrations, on the reasons for the anomalies. You have not shown any causality*

*(coincidence is not a sound scientific argument) nor have you excluded other potential sources such as long-range transport or stratospheric air mixing. In addition, the wave one pattern is already well described in the literature. You should cite appropriate literature here with regard to this effect. In any case, you should change "are related" to "could be related to".*

Same as just two comments above.

*> l.243 How do you define the LT?*

As mentioned in comment 4 about the definitions of the tropospheric layers, the LT is defined in the pressure range between the surface and 750 hPa.

*> l.245 What does this 58% value refer to? An average across the four months for the two sites? Why not include the Gulf of Guinea in that average? Is that an average for the LT only? How do you define the LT? Please be more precise.*

The 58% is the AN/(AN+BB) contribution averaged between the surface and 750 hPa. This value refers to the mean of January and is the same for Lagos and Sahel (in fact 57% which was rounded up to 58).

In order to make this result clearer we added the lines 224-226 page 9 in the revised manuscript:

"During the NH dry season (January), the AN contribution dominates over Lagos (58%) and Sahel (57%), while BB slightly dominates over the Gulf of Guinea (53%) (Figs. 3 panel 3a; A1 panels 1a and 2a; 4a; A2a). "

In order to be more precise about AN and BB contributions in NT Africa we have also modified the following sentences (line 598 in page 28 and line 616 page 29 of the revised manuscript) in the conclusion:

"Over  NT Africa, with contributions in the range of 57-85% local AN emissions largely dominate the CO anomalies all year long. There are a few exceptions of larger BB contributions in January over Guinea Gulf (57%) and in July over Lagos (53%) and Guinea Gulf (66%) during NH and SH African BB seasons. "

"The role of the local AN emissions are more important than previously  documented as: i) local AN emissions  control the CO anomalies over  Lagos and Sahel, and ii) the persistent CO-rich surface layer in Central Africa is caused by local

AN emissions (40 and 86 %) in the absence of local fires.

> *l.247 "extend" is not the right word here*

Corrected.

> *l.257 October is more polluted than January and April in Central Africa and similarly polluted as July in Sahel and Gulf of Guinea*

In order to respect comment 1 of Reviewer #2 related to the length of the paper, we removed this phrase from the revised manuscript as the CO enhancement in the lower tropospheric levels is not a unique characteristic of the African clusters.

> *l.260-261 But this is also true for the other months, right? Look at January for instance, where the mean CO ranges between 250 and 750 ppbv between 0-4km. For that month, SOFTIO only attributes 200 ppbv of CO anomaly, so it clearly misses a large fraction of the CO source for that month as well. On what do you base your statement that this is due to an "underestimation of NHAF AN emission"? The fact that IASI also sees elevated CO (as mentioned in the following sentence) is no proof that they are AN originated. Why not an underestimation of BB emissions? Could it not be an issue with how you defined your background CO in the first place? This clearly needs to be explained in more detail.*

This comment has been discussed in comment 11.

Furthermore, background CO differences between our different methods (see comment 9) account for a maximum of 60 ppbv which could not explain the differences between SOFT-IO and the observations pointed by Reviewer #2 that are about 250 ppbv in the LT for January.

> *l.273 69 ppb, not 70.*

Done

> *l.277-280 and l. 287-289 Here you discuss O3 sinks in the LT, which is redundant with the following paragraph (l.290-297). I suggest keeping the following paragraph and removing these sentences to improve the readability of the manuscript.*

Text has been modified according to Reviewers #2 suggestions.

> *l.314-317 Again, how can you conclude that SOFTIO underestimate AN emission, but not BB emissions for instance? It looks like SOFTIO struggles to match the CO anomalies near the surface for most African sites. Can that only be explained by incorrect AN emission inventory? If that is the reason, how can we trust the rest of the AN vs BB attribution by SOFTIO in the rest of the tropospheric column?*

The reason we comment on the underestimation of the AN emissions is discussed in comment 11. Briefly, we are discussing the underestimation of the AN emissions, as no fires were detected during April over Central Africa and AN emissions were the dominant source of the CO enhancement. The underestimation of AN emissions refers to this concrete month and this concrete region only and it is not a general conclusion about the AN emissions. The discrepancy between the modeled and observed anomaly cannot be explained only by incorrect AN emissions inventory, as there are other sources of uncertainty in SOFT-IO computations (see comment 8 for more details). However, it is not possible to quantify the bias of each individual source separately or the uncertainties coming from AN or BB emission inventories.

> *l.351 Madras is not an "oceanic site", it is a mega-city (fourth biggest city in India).*

Oceanic site has been removed.

> *l.442-445 This is true for all sites in South America, and most sites in Asia and in the Arabian Peninsula (on top of all sites in Africa). So, if all AN emission inventories are wrong in most regions, as suggested by the authors (and why is the reason for that?) How can the authors still quantify AN contribution throughout the tropospheric column with so much certainty?*

As mentioned in comment 11, the underestimation of the AN emissions is discussed over Africa and South America when there are no or few fires active, because AN emissions are causing the CO anomaly almost exclusively.

For AN emissions, we use CEDS2he LT CO maxima over the NT African clusters are due to a combination of AN and BB local emissions. T which is a state of the art inventory but can misrepresent reality. We document an underestimation of AN emissions in specific regions and seasons; this represents valuable information for the experts in charge of this inventory. The reason for

such an underestimation is out of the focus of our study and concerns emission experts.

> *l.561 So is 100 ppb the background level of CO? Shouldn't the background value be altitude dependent, with lower values in the UT?*

The background we use is season and region dependent. see comment 9 for more details.

> *l.571-573 How so? O3 could be formed by BB emissions with the CO maxima being still due to AN emissions. That O3 and CO maxima aren't collocated doesn't mean that O3 production isn't due to BB emissions.*

It is indeed complicated to draw accurate conclusions about O3 attribution without the use of a chemical transport model. The statement has been removed in the revised manuscript.

> *l.575 Isn't O3 always formed by photochemical processes?*

Text has been modified:

"This  indicates that O3  is likely associated with larger ozone production efficiency in the FT (Sauvage et al., 2007c)."

> *l.579 I was not aware that lightning also produces CO. How can LiNOx be responsible for elevated CO in the MT?*

Text has been modified in the revised manuscript:

"In contrast, in SBrazil and Windhoek in October, the co-occurrence of O3 and CO enhancement in the MT and UT indicates  O3 production from surface sources (e.g. fires  In addition, LiNOx emissions can contribute to the O3 production in the FT (Secs. 3.2.1 and 3.2.3)."

> *l.581 The large majority of your flights are over megacities. I think it is fair to assume that low O3 in the BL are due to titration by NOx*

Text has been modified in the revised manuscript:

"This is likely related to  deposition and titration by NO (see Sect. 3.1.1 for

more details)."

> l.582-583 This is not obvious to me. Looking at the various figures, it looks like BB emissions contribute a lot in January to most of the African clusters. In addition, the O3 maxima is closer to the MT than the LT (although it is not clear how the authors dined these two layers) whereas the CO maxima is at the surface. So, to me there is clearly a discrepancy with the text here.

As we discussed in comment 4, LT is defined from the surface below 750 hPa. Based on IAGOS auxiliary data, 750 hPa corresponds to 3 km. Indeed, as mentioned in lines 226 and 229 page 11 of the original manuscript, CO peaks close to the surface and O3 at around 2.5 km (accompanied by elevated CO levels) for the NT African clusters. Despite the fact that the O3 maximum is closer to the MT, and the CO maximum close to the surface, they both occur in the LT as defined by the authors. LT CO is attributed to local AN emissions over Lagos and Sahel, and local BB emissions over the Gulf of Guinea.

Lines 478-480 in page 24 is modified in the revised manuscript:

"The highest CO and O3 maxima among all the tropical clusters occur over NH Africa in the LT (at 0.3 km for CO and 2.5 km for O3) during the dry season (January) mostly due to local AN emissions (over Lagos and Sahel) and BB (over Guinea Gulf). "

> l.593 I have a hard time believing that CO would be diluted by BB impacted air masses. Please explain in more detail.

We removed this statement from lines 591-594, page 27 of the original manuscript.

> l.621-622 The authors should rephrase this sentence, as it seems to imply causality between enhanced CO and enhanced O3. Do they really mean that the elevated CO in the LT is responsible for the elevated O3?

The relation between the observed O3 and CO is useful to constrain our understanding of the factors controlling O3 (Voulgarakis et al., 2011). A correlation between O3 and CO was observed in aged pollution and BB plumes (e.g. Parrish et al., 1993), indicating that a region has experienced photochemical O3 production from its precursors (including CO) (Voulgarakis et al., 2011 and references therein). As we mention in the manuscript, during winter in Asia the chemical ageing of the air masses in the LT is favored by: i) the

confinement of the CO-rich air masses due to the large scale subsidence preventing upward vertical motions (Lelieveld et al., 2001) and ii) the cloud free conditions promoting O3 formation. Therefore, the elevated O3 is due to the combination of these reasons (accumulation of air masses in the surface, high solar radiation and large amount of precursors including CO).

Text has been modified:

"As in NH Africa, the CO-rich air masses accumulated in the LT over the Asian clusters in January  are accompanied by a secondary LT O3 maximum."

> *Conclusion There shouldn't be any references in the conclusion.*

References have been removed from the conclusion section. We only kept Petetin et al 2018a to discuss about IAGOS representativity of a regional scale (comment 12).

> *Table 2 Please indicate the pressure range for CO and O3 in the table, as they are not the same for the two species. Please change NH and SH to NT (northern tropics) and ST, otherwise it gets confusing.*

Table 2 was presented in section 3.1 of the original manuscript. As mentioned before, we decided to remove this section in the revised manuscript, in order to take into consideration from Reviewers #2 comments 1-2 related to this section and the length of the paper.

> *Figure 2 Please indicate that those are monthly mixing ratios averaged from 2008 to 2019.*

Same as the comment just above.

> *Figure 4 Please explain in the caption why Sahel and the Gulf of Guinea are not included in the figure (for BB vs AN contribution) but rather given in the appendix.*

We added in the caption of Figure 3 in page 11 of the revised manuscript:

"For clarity reasons the CO contribution for Sahel and Gulf of Guinea are displayed in Fig. A1."

*> Figure 7 Please expand the axes so that all of the data are shown in panels 3, 4 and 5.*

Done

*> Figure S1 The latitude band is not the same for all panels. Please homogenize.*

Done

References

Bourgeois, I., Peischl, J., Thompson, C. R., Aikin, K. C., Campos, T., Clark, H., Commane, R., Daube, B., Diskin, G. W., Elkins, J. W., Gao, R.-S., Gaudel, A., Hintsa, E. J., Johnson, B. J., Kivi, R., McKain, K., Moore, F. L., Parrish, D. D., Querel, R., Ray, E., Sánchez, R., Sweeney, C., Tarasick, D. W., Thompson, A. M., Thouret, V., Witte, J. C., Wofsy, S. C., and Ryerson, T. B.: Global-scale distribution of ozone in the remote troposphere from the ATom and HIPPO airborne field missions, Atmos. Chem. Phys., 20, 10611–10635, https://doi.org/10.5194/acp-20-10611-2020, 2020.

Cussac, M., Marécal, V., Thouret, V., Josse, B., and Sauvage, B.: The impact of biomass burning on upper tropospheric carbon monoxide: a study using MOCAGE global model and IAGOS airborne data, Atmos. Chem. Phys., 20, 9393–9417, https://doi.org/10.5194/acp-20-9393-2020, 2020.

Galanter, M., Levy, H., and Carmichael, G. R.: Impacts of biomass burning on tropospheric CO, NO x, and O3, Journal of Geophysical Research: Atmospheres, 105, 6633–6653, https://doi.org/10.1029/1999JD901113, 2000.

Giglio, L., I. Csiszar, and C. O. Justice (2006), Global distribution and seasonality of active fires as observed with the Terra and Aqua Moderate Resolution Imaging Spectroradiometer (MODIS) sensors,J. Geophys. Res.,111, G02016,doi:10.1029/2005JG000142.

Granier, C., Bessagnet, B., Bond, T., D'Angiola, A., Gon, H. Denier van der, Frost, G. J., Heil, A., Kaiser, J. W., Kinne, S., Klimont, Z., et al. (2011). "Evolution of anthropogenic and biomass burning emissions of air pollutants at global and regional scales during the 1980–2010 period". In: Climatic change 109.1, pp. 163–190. doi: 10.1007/s10584-011-0154-1.

Gressent, A., Sauvage, B., Defer, E., Pätz, H. W., Thomas, K., Holle, R., Cammas, J.-P., Nédélec, P., Boulanger, D., Thouret, V., et al. (2014). "Lightning NOx influence on large-scale NOy and O3 plumes observed over the northern mid-latitudes". In: Tellus B: Chemical and Physical Meteorology 66.1, p. 25544. doi: 10.3402/tellusb.v66.25544.

Lamarque, J.-F., Bond, T. C., Eyring, V., Granier, C., Heil, A., Klimont, Z.,

Lee, D., Liousse, C., Mieville, A., Owen, B., Schultz, M. G., Shindell, D., Smith, S. J., Stehfest, E., Van Aardenne, J., Cooper, O. R., Kainuma, M., Mahowald, N., McConnell, J. R., Naik, V., Riahi, K., and van Vuuren, D. P.: Historical (1850–2000) gridded anthropogenic and biomass burning emissions of reactive gases and aerosols: methodology and application, Atmos. Chem. Phys., 10, 7017–7039, https://doi.org/10.5194/acp-10-7017-2010, 2010.

Lelieveld, J. o., Crutzen, P., Ramanathan, V, Andreae, M., Brenninkmeijer, C., Campos, T, Cass, G., Dickerson, R., Fischer, H, De Gouw, J., et al. (2001). "The Indian Ocean experiment: widespread air pollution from South and Southeast Asia". In: Science 291.5506, pp. 1031–1036. doi: 10.1126/science.1057103.

Logan, J. A., Prather, M. J., Wofsy, S. C., and McElroy, M. B.: Tropospheric chemistry: A global perspective, Journal of Geophysical Research: Oceans, 86, 7210–7254, https://doi.org/10.1029/JC086iC08p07210, 1981.

Parrish, D. D., Holloway, J. S., Trainer, M., Murphy, P. C., Fehsenfeld, F. C., and Forbes, G. L. (1993). "Export of North American ozone pollution to the north Atlantic Ocean". In: Science 259.5100, pp. 1436–1439. doi: 10.1126/science.259.5100.1436.

Parrish, D. D., Law, K. S., Staehelin, J., Derwent, R., Cooper, O. R., Tanimoto, H., Volz-Thomas, A., Gilge, S., Scheel, H.-E., Steinbacher, M., and Chan, E.: Long-term changes in lower tropospheric baseline ozone concentrations at northern mid-latitudes, Atmos. Chem. Phys., 12, 11485–11504, https://doi.org/10.5194/acp-12-11485-2012, 2012.

McDuffie, E. E., Smith, S. J., O'Rourke, P., Tibrewal, K., Venkataraman, C., Marais, E. A., Zheng, B., Crippa, M., Brauer, M., and Martin, R. V.: A global anthropogenic emission inventory of atmospheric pollutants from sector-and fuel-specific sources (1970–2017): an 990 application of the Community Emissions Data System (CEDS), Earth System Science Data, 12, 3413–3442, https://doi.org/10.5194/essd12-3413-2020, 2020.

Petetin, H., Thouret, V., Fontaine, A., Sauvage, B., Athier, G., Blot, R., Boulanger, D., Cousin, J.-M., and Nédélec, P.: Characterising tropospheric O3 and CO around Frankfurt over the period 1994–2012 based on MOZAIC–IAGOS aircraft measurements, Atmos. Chem. Phys., 16, 15147–15163, https://doi.org/10.5194/acp-16-15147-2016, 2016.

Petetin, H, Jeoffrion, M, Sauvage, B, Athier, G, Blot, R, Boulanger, D, Clark, H, Cousin, J.-M., Gheusi, F, Nedelec, P, et al. (2018a). "Representativeness of the IAGOS airborne measurements in the lower troposphere". In: Elementa: Science of the Anthropocene 6. doi: 10.1525/elementa.280.

Petetin, H., Sauvage, B., Parrington, M., Clark, H., Fontaine, A., Athier, G., Blot, R., Boulanger, D., Cousin, J.-M., Nédélec, P., and Thouret, V.: The

role of biomass burning as derived from the tropospheric CO vertical profiles measured by IAGOS aircraft in 2002–2017, Atmos. Chem. Phys., 18, 17277–17306, https://doi.org/10.5194/acp-18-17277-2018, 2018b.

Sauvage, B., Fontaine, A., Eckhardt, S., Auby, A., Boulanger, D., Petetin, H., Paugam, R., Athier, G., Cousin, J.-M., Darras, S., Nédélec, P., Stohl, A., Turquety, S., Cammas, J.-P., and Thouret, V.: Source attribution using FLEXPART and carbon monoxide emission inventories: SOFT-IO version 1.0, Atmos. Chem. Phys., 17, 15271–15292, https://doi.org/10.5194/acp-17-15271-2017, 2017.

Stein, O., Schultz, M. G., Bouarar, I., Clark, H., Huijnen, V., Gaudel, A., George, M., and Clerbaux, C.: On the wintertime low bias of Northern Hemisphere carbon monoxide found in global model simulations, Atmos. Chem. Phys., 14, 9295–9316, https://doi.org/10.5194/acp-14-9295-2014, 2014.

van der Werf, G. R., Randerson, J. T., Giglio, L., Collatz, G. J., Kasibhatla, P. S., and Arellano Jr., A. F.: Interannual variability in global biomass burning emissions from 1997 to 2004, Atmos. Chem. Phys., 6, 3423–3441, https://doi.org/10.5194/acp-6-3423- 2006, 2006.

Voulgarakis, A, Telford, P., Aghedo, A., Braesicke, P, Faluvegi, G, Abraham, N., Bowman, K., Pyle, J., and Shindell, D. (2011). "Global multi-year O 3-CO correlation patterns from models and TES satellite observations". In: Atmospheric Chemistry and Physics 11.12, pp. 5819–5838. doi: 10.5194/acp-11-5819-2011.

Yamasoe, M. A., Sauvage, B., Thouret, V., Nédélec, P., Le Flochmoen, E., and Barret, B.: Analysis of tropospheric ozone and carbon monoxide profiles over South America based on MOZAIC/IAGOS database and model simulations, Tellus B: Chemical and Physical Meteorology, 67, 27 884, https://doi.org/10.3402/tellusb.v67.27884, 2015.

---

## Referee Report (RR1)

2nd review of Tsivlidou et al. "Tropical tropospheric ozone and carbon monoxide distributions: characteristics, origins and control factors, as seen by IAGOS and IASI".

I would like to commend the authors for incorporating most of the comments that were provided after the first round of review. I believe the manuscript in the present form is easier to read and provide some needed justifications for some of the methodological choices.

Unfortunately, I remain largely unconvinced by some of the responses:

i)      SOFTIO clearly struggles at reproducing the magnitude of the CO anomalies in all clusters of the tropical band. On some occasions, as little as 10% of the CO mixing ratios are accounted for by the model. If so little of the CO anomalies is explained by SOFTIO, how can the authors conclude on the sources of those anomalies? Let me try to be clearer here: if SOFTIO can only represent 10% of the CO anomaly at a given cluster, then the source attribution is only valid for those 10%. The remaining 90% are unaccounted for, and this should be clearly highlighted in the paper. This is the reason why in my original review, I had asked the authors to show on their figures how much of the CO anomalies are NOT accounted for by SOFTIO. Otherwise, and it may not be the intention of the authors, it borders on intellectual dishonesty.

ii)     In addition, I don't understand the authors response stating that "For instance, models are persistently biased in the Southern hemisphere and in the tropics, particularly over polluted regions such as India and East Asia. As a result SOFT-IO has to be seen as a tool to perform source attribution and to quantify the relative part of a source influence to another, but not as a tool perfectly able to simulate the exact CO concentrations, but this is a problem of most of the models in CO anomalies." To me, there are two serious problems with this response. If I was to rephrase it in a simpler way, it reads as "we know our model does a poor job at reproducing CO mixing ratios, but we are going to do it anyways because all models do equally poorly" and as "we can't reproduce CO mixing ratios, but we are still going to apportion sources contribution and disregard the remaining CO not reproduced by the model". I believe these statements speak for themselves.

iii)    The fact that SOFTIO struggles to reproduce CO mixing ratios indicate that at least one or both the AN and BB emission inventories used in this study severely underestimate CO emissions in the tropics. The authors keep claiming that this is mostly due to the AN emission inventory, but with no scientific evidence for it. Their first argument is that "Furthermore, we comment the performance of SOFT-IO when the CO anomalies are attributed entirely to one source (AN) and to one source region. For instance, in the case of Africa (NH and SH) (line 261, page 14 and line 316, page 16 respectively in the original manuscript) and South America (line 445, page 21 of the original manuscript), we discuss the underestimation of the AN emissions during the transition periods, when the fires are suppressed." Looking at their Figure S1 showing GFAS estimations of BB, clearly there is fire activity in Africa that would affect the clusters in April. Their

response is in direct contradiction with the data they show. Their second argument is that they performed a sensitivity analysis on emission inventories and found that AN emission inventories weighted more than BB emission inventories. Not surprising: the authors used GFED and GFAS as BB emission inventories, which are extremely similar to one another. On the other hand, they compared CEDS and MACCity for AN emission inventories, which comes down to comparing a Rolls Royce with a Toyota Corolla (and this is meant with no disrespect for Toyota). Of course, they would find a higher sensitivity of SOFTIO results to AN emission inventories. Now, what if the authors used the latest BB emission inventory, FINNv2.5? In their recent paper, Wiedinmyer et al. (2023) clearly show that both GFED and GFAS are lower by a factor of almost two compared to FINNv2.5 or FEER for CO emissions (see their Figure 4 pasted below).

[Figure]

05

**Figure 4.** Annually-averaged (2012-2019) emissions of $CO_2$, CO, formaldehyde ($CH_2O$), particulate black carbon (BC), ammonia ($NH_3$), ethane ($C_2H_6$), sulfur dioxide ($SO_2$), and nitrogen oxides ($NO_x$) from the Fire Inventory from NCAR version 2.5 (FINNv2.5), FINNv2.5 MODIS-only version, FINNv1.5, Global Fire Emissions Database (GFED), Fire Energetics and Emissions Research (FEER), Global Fire Assimilation System (GFAS), and Quick Fire Emissions Dataset (QFED). Bars show global totals broken up

10    into regional totals by color (Giglio et al., 2010), and shown in the global map here, namely Boreal North America (BONA), Temperate North America (TENA), Central America (CEAM), Northern Hemisphere South America (NHSA), Southern

For all those reasons, I still can't recommend publication for this manuscript, pending the above-mentioned issues are addressed.

---

## Referee Report (RR2)

**Review comments on Tsivlidou et al.: *Tropical tropospheric ozone and carbon monoxide distributions: characteristics, origins and control factors, as seen by IAGOS and IASI**

**General comments**

Tsivlidou et al. present a valuable study adding to the understanding of $CO/O_3$ distributions in the tropics. I very much appreciate the efforts the authors have gone through in their data analysis. I think the trajectory approach in the SOFT-IO model to disentangle the contribution of wildfire and anthropogenic emissions to the observed signals is sound and sufficiently well documented by Sauvage et al. (2017) to be used here.

However, I think that the manuscript is not yet suited for publication but needs to be significantly shortened and made more concise. A great part of the result section is dedicated to the lengthy and extremely detailed description of the observed and modelled profiles. Unfortunately the descriptions are difficult to follow, jumping back and forth between regions/locations, altitude regimes and the different figures, with the structure of the discussion often remaining unclear. Please comprise details and have a more structured discussion clearly presenting the similarities/differences between the regions and altitude regimes. The most relevant figures in the draft are Fig. 9 and Fig. 10. Given their importance and content they are not sufficiently discussed although they actually summarize the results of the preceding lengthy and detailed discussion.

**Specific comments**

– There are too many abbreviations used. I understand that this is an attempt to keep it short, but to a degree that the text gets close to unreadable. Please use full wording more often. In particular, the letter $T$ is used in the abbreviations for *tropics* and *troposphere* which makes it even more difficult to keep things sorted. The way abbreviations are embedded into the text sometimes seems strange with

regard to grammar.

– I was confused by the term 'observational site' being used for a moving platform. 'Locations' would be more appropriate in my opinion. The term 'site' is usually used for a (temporarily) fixed installation of measurements equipment in one place.

– I trust the ACP editorial team will eventually take care of this but the usage of italics in subscripts and units is inconsistent and wrong in many instances.

– it is confusing to have both an appendix and a supplementary document

– Line 522: Table 2 referenced here is not part of the draft.

**Abstract**

In my opinion the abstract is too long and not well organized. The main findings are unclear. I suggest to remove some details and make the abstract more concise. It should become clear what the main conclusions of the analysis are and why these are relevant.

– L 11: 'in above 6 km' does not make any sense

– L 13: What do you mean by 'The highest amount of transported CO'. Transport to Asia? Overall?

**Introduction**

– L 32: Why is stratospheric influence as the least important process mentioned first?

– L 37: I understand biomass burning throughout the manuscript refers to wildfires excluding usage of biogenic fuels which is attributed to anthropogenic emissions. This should be made clear here.

– L 59 constraint → constrain

– L 68: 'offered' – the choice of word reads strange here

**Data and Methods**

– Line 121f: what do you mean by 'a distance criteria of 300-km'?

– Line 165: what do you mean by 'with bias lower than 10-15 ppb'? Please specify what *bias* exactly refers to here.

– Line 173f: percentages are not absolute differences. The statement does not make any sense to me.

– Line 177: Why two? Which two backgrounds are referred here? The two mentioned pressure surfaces?

**Results**

All vertical profiles are discussed in terms of absolute altitude but in subsection 3.2. the different altitudes of the inbound/outbound airports are mentioned. In particular for the peaking altitudes presented in Fig. 8, I wonder what the results looked like if altitude differences relative to the local ground level were used.

– Line 234: my reading from the figure would be 62+6=68

– Line 240 and 248: Throughout the manuscript mixing ratios are discussed, not concentrations. Similar on several instances in the following.

– Line 246: the 'observed anomaly' to me is not evident in the figure.

– Line 255f: no need to cite Adon et al. twice within two lines. Skip first one.

– Line 258: there is no obvious peak in the CO profile in the figures

– L265: space missing between brackets

– L271: Reference to Fig 2l does not make sense, Fig. 2l shows CO.

– L445: If the vertical layers are defined on pressure as the vertical coordinate then why are km shown in the figures?

**Figures**

Overall, there are too many figures with too many panels and too small fonts.

The presentation of observations on an absolute mixing ratio scale and the modelled contributions as $\Delta CO$ is difficult to compare. Why are the vertical profiles shown not background corrected?

– Fig. 2: I suggest to have the panel labels in some lighter colour to make them visible.

– Figures 3,5,6, 7 are poor resolution and cannot be zoomed which is essential given the small panels and fonts.

– Figure 8: I was wondering about the order in which locations are presented on the x-axis. It would be logical to have the locations by longitude which does not seem to be the case.

**Summary**

The Summary largely rephrases the detailed discussion from above. Conclusions are presented alongside but are not worked out well. Please be more precise and separate the shortened descriptive discussion of the observations from the conclusions drawn.

– Line 611 should be 'NT' only

---

## Author Response (AR2)

**Letter to the Editor**

**Tropical tropospheric ozone and carbon monoxide distributions: characteristics, origins and control factors, as seen by IAGOS and IASI**

Maria Tsivlidou et al.

*In the second round of peer-review, I received strongly contrasting reviewer report and thus asked a third reviewer for their input. The third reviewer points out that, while they do not share the general concerns of reviewer 1 regarding the methodology, they conclude that major revisions are needed before the manuscript can be published with ACP. Thus, I encourage the authors to perform another major revision of the manuscript, considering all three new reviewer reports and focusing on presentation quality of the article. The paper will be accepted if the scientific results and conclusions are presented in a clear, concise, and well-structured way.*

Dear Editor,

we have carefully addressed in the revised version the comments from Referee#1 and the new Referee#3.

We have also responded to the 3 points raised in Referee#2's second assessment (Report#1), although we strongly disagree with his/her comments which are not at all unbiased. Referee first recognizes that we have considered his/her first comments but he/she objects to new ones to reject the publication. Furthermore, Referee focuses his/her new comments on the less concluding details of the paper disregarding most of the results which are concluding. Finally, his/her judgment is based on an incorrect assessment of the Figures with no explanations.

- First, Referee keeps on repeating that the model only reproduces 10% of the anomalies when this is absolutely not the case. The paper shows that the model reproduces the vast majority of anomalies. As explained in our answer, we do not understand why the Referee argues on supposed large underestimation by SOFT-IO. For the 20 clusters and in the 3 vertical layers and the 4 months, SOFT-IO reproduces on average 93,5% (relative difference) of the observed CO. The underestimation is slightly larger on average in the LT, but still SOFT-IO reproduces 87% of the anomalies. The largest underestimation is observed over the Sahel cluster in the LT in January, with 52,8% of the anomaly reproduced by the model, still far from the 10% claimed by Referee. The only occurrence of a 90% bias is over Mumbai in January in the 40hPa surface layer (see the answer to Referee#2 for details about biases in the high resolution profiles). Nevertheless, first our conclusions and main results are all based on the broad LT, MT and UT layers for which the model results are robust as explained above and second, there are only 2 profiles for Mumbai in January which makes this bias statistically irrelevant. This fact was already acknowledged with a warning in the first version of the paper. Therefore, the repeated mention to this accidental 10% by Referee#2 is far from fair. Then, Referee#2 suggests that we deliberately hide the worst results. This is not true, the

figures presented in the paper and in the Appendix are rather exhaustive, even too much according to Referee#3. The underestimations of the model were clearly explained in Section 2.2 lines 164-166 and lines 172-175 of the 1st round revised version. We give more precise numbers of SOFT-IO limitations in this 2nd round revised version. Imperfections and biases are a common feature of models such as SOFT-IO and we did not try to hide it on the contrary. We would like to emphasize that such a model evaluation is useful for a large community using emission inventories. Referee#2 then argues that we state that our model "does a poor job" but that we still want to use it. This appears contradictory to his/her allegation that we aimed at hiding our model limitations. Moreover, we never said that our model "does a poor job", we give an objective and fair scientific statement that all modelers know, models are imperfect by definition, because they use inputs (emission inventories, meteorological fields) and schemes (chemical schemes, convective parameterization, turbulence) that are all perfectible. Our model does a good job, contrary to what is claimed by Referee#2 without any scientific argumentation, and Figures and numbers provided by SOFT-IO speak by themselves. The version of SOFT-IO we use is based on a state of the art Lagrangian model (FLEXPART last version); a state of the art biomass burning inventory (GFAS last version) which is the only that provides biomass burning injection height and that is operational; a state of the art anthropogenic emissions (CEDS2); and state of the art meteorological analysis and forecast (ECMWF). All of them are separately or together largely used in hundreds of peer-reviewed publications of high level scientific journals. If we follow his/her sarcastic statement, no publications in atmospheric science based on a model should be published, as all models and input or subparts are impacted by uncertainties.

- Finally, Referee#2 argues that our conclusions are biased by the emission inventories we use based on a preprint (Wiedinmyer et al. in discussion, since Feb 2023) not yet accepted and which, as such, could absolutely not be known and used by the authors at the time of submission. This is not a fair and correct way of reviewing a paper.
Our study is based on the analysis of two datasets (IASI and IAGOS) and using the best SOFT-IO version at that time, and we never claimed that our conclusions were set in stone. The large majority of model studies in atmospheric science use one state of the art biomass burning inventory, and one state of the art anthropogenic emission inventory, and conclusions are obviously related to this particular choice of inventories, but also of meteorological fields, transport schemes, chemical schemes, version of the model etc. This is the same for all studies using models in a given configuration, so his argument is inadmissible. We have revised the conclusions when fire and anthropogenic contributions are of the same order, as the other inventories could change the results, but not when fire or anthropogenic very largely dominates.

In addition and most importantly, we would like to point out that this second review of Referee#2 does not comply at all with the reviewers' code of conduct in many respects. In particular, "The reviewer of a manuscript must judge objectively the quality of the manuscript and respect the intellectual independence of the authors. Under no circumstances is

personal                criticism                appropriate."                (see
[https://www.atmospheric-chemistry-and-physics.net/policies/obligations_for_referees.html](https://www.atmospheric-chemistry-and-physics.net/policies/obligations_for_referees.html))

Many comments of Referee#2 are indeed ironic, insulting and inappropriate in a scientific discussion such as "it borders on intellectual dishonesty" or "I believe these statements speak for themselves" or making fun of the comparison we did between CEDS2 and MACCITY (the last one was largely used by the community at the time of the study) without any scientific argumentation.

There seems to be an obvious conflict of interest with a strong will to prevent the publication of our work. Such behaviour should not be allowed in a publication of the rank of ACP.

Once again, we express our gratitude for the thorough review process and the comments of Referee#1, Referee#2 and Referee #3, which has helped strengthen our manuscript. We are confident that with the suggested modifications, our paper will be well--positioned for publication in the ACP journal.

Thank you for your time and consideration. We look forward to hearing from you regarding the final decision on our manuscript.

Sincerely,

Maria Tsivlidou

**Answer to Referee#2 (blue in the text)**

**2nd Review of Tsivlidou et al. "Tropical tropospheric ozone and carbon monoxide distributions: characteristics, origins and control factors, as seen by IAGOS and IASI".**

*I would like to commend the authors for incorporating most of the comments that were provided after the first round of review. I believe the manuscript in the present form is easier to read and provide some needed justifications for some of the methodological choices.*

We thank Referee for his/her comments, indeed we provided substantial answers to all his concerns expressed in the first review.

*Unfortunately, I remain largely unconvinced by some of the responses:*

*i) SOFTIO clearly struggles at reproducing the magnitude of the CO anomalies in all clusters of the tropical band. On some occasions, as little as 10% of the CO mixing ratios are accounted for by the model. If so little of the CO anomalies is explained by SOFTIO, how can the authors conclude on the sources of those anomalies? Let me try to be clearer here: if SOFTIO can only represent 10% of the*

*CO anomaly at a given cluster, then the source attribution is only valid for those 10%. The remaining 90% are unaccounted for, and this should be clearly highlighted in the paper. This is the reason why in my original review, I had asked the authors to show on their figures how much of the CO anomalies are NOT accounted for by SOFTIO. Otherwise, and it may not be the intention of the authors, it borders on intellectual dishonesty*

We do not agree with this comment. Referee focuses on "some occasions" where SOFT-IO would reproduce 10% of the anomalies but this does not happen.

We provide below quantitative arguments that demonstrate that Referee repeated statement about 10% is wrong and unfair:

- SOFT-IO reproduces remarkably well the shape of the CO vertical profiles and also their seasonal variability as shown in the profile Figs. 3, 5, 6 and 7.
- SOFT-IO performances on anomaly frequencies:
When focusing on the strongest CO anomalies (higher than the 75th percentile of IAGOS measurements), which are supposed to be less simulated by the model, we clearly see that SOFT-IO detects the strongest anomalies 85-90% (in frequency) of the time (Table 1 below). Lowest values are observed over Addis Ababa and Bogota in the LT because these sites are located above 2 km altitude, with few measurements in the LT layer (below 750 hPa).
- SOFT-IO performances on anomaly amplitudes:
Table 2 displays the absolute (left) and relative (right) differences between IAGOS and SOFT-IO anomalies. To be comparable, background has to be added to SOFT-IO (or subtracted to IAGOS measurements). The relative difference is then calculated as (IAGOS-(SOFT-IO+background)) / IAGOS, and allows to compare observed and simulated anomalies in the 3 different layers, for all the clusters, averaged for the 4 months.
On average for the 3 layers, SOFT-IO reproduces more than 93% of the amplitude of observed anomalies, and the largest underestimation occurs in the LT with 87%.
When looking at the LT layer where observed anomalies are the strongest and where model performances are supposed to be the lowest, January and April are the months with the lowest performances, with respectively 82,4% and 78,5% of the anomalies reproduced in average for the 20 clusters, and a standard deviation of respectively 20% and 13%.
Looking at individual clusters, months and layers, the 2 lowest performances occur in the LT over Sahel and Mumbai in January, with respectively 52.8% and 53,1% of the amplitude of the anomalies reproduced by SOFT-IO, which is far from 10% .

Of course, the profiles display larger differences especially close to the surface in polluted areas where large scale models have more difficulties to represent anomalies resulting from local contributions. That is why our conclusions are based upon the results concerning the three broad layers representing the LT, MT and UT. Furthermore, profile Figs. 3, 5, 6 and 7 with the x-scale extended to the largest CO mixing ratio for all months and all airports from a region probably induced the wrong idea that the model does a "poor job" reproducing the anomalies. Nevertheless, if we look at the same relative differences for the high resolution profiles than for the 3 broader layers (discussed above), above the surface level the model represents in majority the CO anomalies at 40 hPa vertical resolution with a bias lower than 40% (see Figure 1 below for Bangkok). Looking at the surface 40 hPa level, the bias reaches 80% only on 4 occasions: at Sahel in January, Central Africa in April or Lagos in April and October (see Figure 2 below for Lagos).

The ONLY case with 90% bias is at the SURFACE level over MUMBAI in JANUARY. But it is based on a low statistics (2 profiles in January) and a warning about this problem is already given in the paper: "It has to be noted that the number of profiles over Mumbai (6) and Hyderabad (19) are lower than the threshold established for representativeness (see Sect. 2.1.1)."

[Figure]

Figure 1: Vertical profile of relative difference between IAGOS and SOFT-IO anomalies (IAGOS- (SOFT-IO + BG) / IAGOS) averaged for January, April, July and October over **Bangkok**.

[Figure]

Figure 2: Same as Figure 1 for **Lagos**.

SOFT-IO's general underestimation (in ppb) is already discussed in the manuscript (Section 2.2, lines 161-163 and lines and 169-170), and the comparison between IAGOS and SOFT-IO (without the background that has to be added on SOFT-IO plots for correct comparisons, as explained in Section 2.2) is clear enough on Figs 3, 5, 6 and 7. The paper has already been commented as too dense with too many figures so we think it not appropriate to add some more.

We have added additional percentages of absolute and relative differences between IAGOS and SOFT-IO for the strongest anomalies in the text of Section 2.2, lines 170-174 of the 2nd round revised version.

|  | Accounted anomalies (%) | | |
| --- | --- | --- | --- |
|  | LT | MT | UT |
| Abu Dhabi | 93.72 | 84.67 | 75 |
| Thailand gulf | 98.57 | 92.94 | 89.62 |
| Central Africa | 96.72 | 93.97 | 94.5 |
| Sahel | 95.38 | 93.29 | 92.94 |
| South China | 99.04 | 94.49 | 94.29 |
| Gulf of Guinea | 99.45 | 97.87 | 96.64 |
| SBrazil | 94.93 | 85.05 | 85.28 |
| Addis Ababa | 54.65 | 90.05 | 85.49 |
| Bangkok | 98.71 | 94.99 | 94.52 |
| Bogota | 1.23 | 89.84 | 88.34 |
| Caracas | 97.1 | 83.53 | 78.16 |
| Ho Chi Minh City | 98.01 | 88.94 | 88.32 |
| Hyderabad | 94.69 | 94.17 | 91.69 |
| Jeddah | 95.54 | 88.14 | 86.82 |
| Khartoum | 90.63 | 82.27 | 84.81 |
| Lagos | 99.6 | 96.57 | 96.31 |
| Madras | 98.4 | 89.93 | 90.06 |
| Manila | 96.93 | 85.47 | 84.41 |
| Mumbai | 97.72 | 95.52 | 92.35 |
| Windhoek | 90.52 | 86.12 | 88.27 |

Table 1: Detection frequency of the CO anomalies higher than the 75th percentile of the IAGOS observations.

| | Absolute difference (ppbv) IAGOS - (SOFT-IO + BG) | | | Relative difference (%) (IAGOS - (SOFT-IO + BG))/ IAGOS | | |
|---|---|---|---|---|---|---|
| | LT | MT | UT | LT | MT | UT |
| Abu Dhabi | 5.75 | -0.89 | -4.18 | 3.18 | -1.59 | -4.04 |
| Thailand gulf | -7.71 | -20.74 | 7.43 | 0.97 | -5.63 | 2.59 |
| Central Africa | 58.79 | 16.29 | 20.37 | 25.32 | 9.92 | 15.46 |
| Sahel | 39.88 | 3.56 | 5.20 | 10.11 | -1.26 | 2.51 |
| South China | -10.00 | -1.73 | 0.77 | -7.23 | -1.97 | -1.78 |
| Guinea Gulf | 43.40 | 10.89 | 9.50 | 12.64 | 4.90 | 6.11 |
| SBrazil | 0.42 | 1.33 | 20.92 | -4.76 | -0.71 | 12.62 |
| Addis Ababa | 92.53 | -0.23 | 1.21 | 19.36 | -3.54 | -0.74 |
| Bangkok | 11.68 | 6.79 | 2.66 | 2.50 | 4.94 | 0.60 |
| Bogota | | 43.38 | 16.08 | | 27.14 | 13.60 |
| Caracas | 49.80 | 10.22 | 6.45 | 28.11 | 9.39 | 6.00 |
| Ho Chi Minh City | 94.07 | -0.71 | -2.64 | 29.75 | -1.28 | -2.42 |
| Hyderabad | 4.52 | 2.85 | 1.45 | 2.56 | 0.56 | -0.14 |
| Jeddah | 20.40 | 6.02 | 2.26 | 14.63 | 4.91 | 0.04 |
| Khartoum | 29.25 | 2.88 | 2.89 | 19.21 | 1.71 | 1.50 |
| Lagos | 127.52 | 10.24 | 13.09 | 28.78 | 3.61 | 8.15 |
| Madras | 18.05 | 2.58 | -1.67 | 9.95 | 0.61 | -1.99 |
| Manila | 41.85 | 0.03 | -0.08 | 22.13 | -0.83 | -2.19 |
| Mumbai | 43.95 | 4.08 | 2.65 | 24.23 | 3.60 | 1.56 |
| Windhoek | 10.43 | 10.67 | 14.17 | 6.63 | 8.10 | 11.38 |

Table 2: Absolute and relative difference between the observed (IAGOS) and simulated (SOFT-IO) anomalies averaged for 3 vertical layers over the 4 months (January, April, July, October).

*ii) In addition, I don't understand the authors response stating that "For instance, models are persistently biased in the Southern hemisphere and in the tropics, particularly over polluted regions such as India and East Asia. As a result SOFT-IO has to be seen as a tool to perform source attribution and to quantify the relative part of a source influence to another, but not as a tool perfectly able to simulate the exact CO concentrations, but this is a problem of most of the models in*

*CO anomalies." To me, there are two serious problems with this response. If I was to rephrase it in a simpler way, it reads as "we know our model does a poor job at reproducing CO mixing ratios, but we are going to do it anyways because all models do equally poorly" and as "we can't reproduce CO mixing ratios, but we are still going to apportion sources contribution and disregard the remaining CO not reproduced by the model". I believe these statements speak for themselves.*

Like any atmospheric model, SOFT-IO is not perfect as by definition, a model is a simplification of the true atmospheric processes (dynamic, chemistry), based on perfectible inputs (emission inventories, meteorological analysis) or subparts (chemical schemes, transport schemes). However our model does a very good job, as demonstrated in the reference paper of SOFT-IO (Sauvage et al., 2017), and also in this current study where more than 78% of the strongest anomalies are simulated (see Table 1). SOFT-IO uses a state of the art Lagrangian model (FLEXPART), state of the art meteorological ECMWF analysis and state of the art emissions inventories (GFAS and CEDS2 last versions), all largely used by the community and published in high standard peer reviewed journals. GFAS is for instance used in the Copernicus Atmosphere Monitoring Service (CAMS) model simulations (https://atmosphere.copernicus.eu/) (e.g. Flemming et al., 2017; Inness et al., 2019). It is operational and provides fire injection altitude. CEDS2 is used by some of the models included in the IPCC reports (e.g. Emmons et al., 2020; Horowitz et al., 2020; Griffiths et al. 2020).

Emmons, L. K., Schwantes, R. H.,Orlando, J. J., Tyndall, G., Kinnison, D.,Lamarque, J.-F., et al. (2020). The Chemistry Mechanism in the Community Earth System Model Version 2 (CESM2). Journal of Advances In Modeling Earth Systems,12,e2019MS001882. https://doi.org/10.1029/2019MS001882

Flemming, J., Benedetti, A., Inness, A., Engelen, R. J., Jones, L., Huijnen, V., Remy, S., Parrington, M., Suttie, M., Bozzo, A., Peuch, V.-H., Akritidis, D., and Katragkou, E.: The CAMS interim Reanalysis of Carbon Monoxide, Ozone and Aerosol for 2003–2015, Atmos. Chem. Phys., 17, 1945–1983, https://doi.org/10.5194/acp-17-1945-2017, 2017.

Horowitz, L. W., Naik, V., Paulot, F.,Ginoux, P. A., Dunne, J. P., Mao, J.,et al. (2020). The GFDL global atmospheric chemistry‐climate model AM4.1: Model description and simulation characteristics.Journal ofAdvances in Modeling Earth Systems,12, e2019MS002032. https://doi.org/10.1029/2019MS002032

Inness, A., Ades, M., Agustí-Panareda, A., Barré, J., Benedictow, A., Blechschmidt, A.-M., Dominguez, J. J., Engelen, R., Eskes, H., Flemming, J., Huijnen, V., Jones, L., Kipling, Z., Massart, S., Parrington, M., Peuch, V.-H., Razinger, M., Remy, S., Schulz, M., and Suttie, M.: The CAMS reanalysis of atmospheric composition,

Atmos. Chem. Phys., 19, 3515–3556, https://doi.org/10.5194/acp-19-3515-2019, 2019.

Griffiths, P. T., Murray, L. T., Zeng, G., Shin, Y. M., Abraham, N. L., Archibald, A. T., Deushi, M., Emmons, L. K., Galbally, I. E., Hassler, B., Horowitz, L. W., Keeble, J., Liu, J., Moeini, O., Naik, V., O'Connor, F. M., Oshima, N., Tarasick, D., Tilmes, S., Turnock, S. T., Wild, O., Young, P. J., and Zanis, P.: Tropospheric ozone in CMIP6 simulations, Atmos. Chem. Phys., 21, 4187–4218, https://doi.org/10.5194/acp-21-4187-2021, 2021.

*iii) The fact that SOFTIO struggles to reproduce CO mixing ratios indicate that at least one or both the AN and BB emission inventories used in this study severely underestimate CO emissions in the tropics. The authors keep claiming that this is mostly due to the AN emission inventory, but with no scientific evidence for it. Their first argument is that "Furthermore, we comment the performance of SOFT-IO when the CO anomalies are attributed entirely to one source (AN) and to one source region. For instance, in the case of Africa (NH and SH) (line 261, page 14 and line 316, page 16 respectively in the original manuscript) and South America (line 445, page 21 of the original manuscript), we discuss the underestimation of the AN emissions during the transition periods, when the fires are suppressed." Looking at their Figure S1 showing GFAS estimations of BB, clearly there is fire activity in Africa that would affect the clusters in April. Their response is in direct contradiction with the data they show. Their second argument is that they performed a sensitivity analysis on emission inventories and found that AN emission inventories weighted more than BB emission inventories. Not surprising: the authors used GFED and GFAS as BB emission inventories, which are extremely similar to one another. On the other hand, they compared CEDS and MACCity for AN emission inventories, which comes down to comparing a Rolls Royce with a Toyota Corolla (and this is meant with no disrespect for Toyota). Of course, they would find a higher sensitivity of SOFTIO results to AN emission inventories. Now, what if the authors used the latest BB emission inventory, FINNv2.5? In their recent paper, Wiedinmyer et al. (2023) clearly show that both GFED and GFAS are lower by a factor of almost two compared to FINNv2.5 or FEER for CO emissions (see their Figure 4 pasted below).*

We disagree, as demonstrated before SOFT-IO does not struggle to reproduce CO anomalies at all! This is just the Referee biased vision. Also we only claim that AN are likely to be underestimated during the month when BB are at their minimum intensity in April. Over Northern Hemisphere Africa, BB emissions are much lower in April, with 1.02e-10 kg/m2/s on average according to GFAS, versus 3.47e-10 kg/m2/s during the fire season. This is also clearly visible on MODIS fire counts (for instance Fig.7 of Yamasoe et al., 2015 Tellus-b paper, cited in our study). During this specific month, AN contributes to more than 85% of the observed anomalies in the LT of West Africa, and as BB is minimum, AN are very likely to be

underestimated. We have clarified this statement in the revised version (see more details at the end of this answer)

In addition, Referee cites a preprint paper (Wiedinmyer et al. in discussion, since Feb 2023) that was not even under discussion when we submitted our manuscript (28 September 2022). This is unfair and irrelevant.

Referee also makes fun of comparisons we did between some anthropogenic emissions inventories (CEDS and MACcity), but at the time of the reference Sauvage et al., 2017 paper and of the current work of our study (Tsivilidou et al.), these emission inventories were state of the art and largely used by the scientific community.

As a result we use state of the art emission inventories, GFAS and CEDS2, that are largely used in many published studies. The majority of the model studies that perform chemistry and transport modeling uses one state of the art BB inventory, same for anthropogenic, or for biogenics, or other sources, depending on the study, and does not provide sensitivity test between BB or anthropogenic emission inventories such as we did in Sauvage et al., 2017 or Tsivlidou et al. current study.

In this Wiedinmyer et al., 2023 paper (currently under revision), Referee just takes one figure stating that Finn v2.5 is larger than GFAS. This is true but on a global and yearly average (their Fig. 4). However there are important regional differences over the tropics in Wiedinmyer et al., 2023 (their Fig. 5) not mentioned by Referee: FINN is larger than GFAS over Africa (NHAF and SHAF), but GFAS is larger than FINN over Asia (EQAS), and there is no proof with this study that FINN or GFAS or another one may be a better emission inventory. There are also differences with other state of the art biomass burning emission inventories not mentioned by Referee (with QFED or FEER). Moreover, the only validation of FINN is through a chemical transport model (CAM chem) by comparing CO simulated tropospheric columns to the ones observed by MOPITT, just for the 2018 year or August 2018. There is no similar comparison in Wiedinmyer et al., 2023 paper using GFAS with the CAM chem model, or using different anthropogenic emission inventories (that also shows large differences) to see which emission inventory would best fit the total tropospheric CO columns.

In our study, we compare SOFT-IO using GFAS and CEDS2 using 17 years of in situ IAGOS high resolution vertical profile observations.In addition to the fact that GFAS is a reference in the community, as others emission inventories, we use GFAS because:

- It is an operational emission inventory that fits the near real time SOFT-IO calculations that we provide for the IAGOS users, which is not the case for other biomass burning emission inventories. GFAS is also used in the CAMS global model simulations.

- GFAS also provides biomass burning injection heights which are important for better simulations of biomass burning plumes (see sensitivity on fire injection in Sauvage et al., 2017). This is not the case of other biomass burning inventories.

All the conclusions we make in the paper concerning the origin of the anomalies are obviously dependent on the emission inventories, the Lagrangian model and the meteorological analysis, as it is for all conclusions done in atmospheric science using models that will depend on the model configuration, version and input parameters.

In the 2nd round revised version, we have revised the affirmations and conclusions (see below for details) when fire and anthropogenic contributions are of the same order, as the influence of other inventories (such as FINNv2.5) could change the results, but not when fire or anthropogenic very largely dominate the other one.
We have modified Section 2.2 lines 172-175 of first round revised version, by adding additional information of SOFT-IO performances:
"On average, SOFT-IO underestimates the observed CO anomalies by 10 ppb in the MT and UT, and by 45 ppb in the LT. A sensitivity test has shown absolute differences of 27% in the LT, 16% in the MT and 10% in the UT between SOFT-IO simulations using AN emissions from MACCity and from CEDS2. This clearly highlights the large uncertainty stemming from uncertainties in AN emissions."
by the following:
"On average, SOFT-IO underestimates the observed CO anomalies by 10 ppb in the MT and UT, and by 45 ppb in the LT. When looking at the differences between IAGOS and SOFT-IO anomalies (taking into account the background not simulated by the model), SOFT-IO reproduces on average more than 93% of the observed anomalies for the 3 layers and 87% in the LT. April is the month where the model gives the largest underestimation, with 78% of the anomalies simulated on average over the 20 clusters (13% standard deviation). The lowest performance is in the LT of Sahel in January, with 52% of the anomaly simulated by SOFT-IO." lines 169-174 of the 2nd round revised version.

We also rephrase: "when the fires are suppressed" line 244 of the 1st round revised version by "when the fires are reduced" line 255 of the 2nd round revised version.

We have suppressed the sentences "The fact that SOFT-IO attributes approximately 80 ppbv of CO to local AN emissions (Figs. 3 panel 3b; A1 panels 1b and 2b), while the observed anomaly reaches 200–250 ppbv and no or few fires are detected by MODIS (Yamasoe et al. (2015); their Fig.7), indicates underestimation of the Northern Hemisphere African AN emissions" lines 245-248
and
"The measured CO maxima reaches 350 ppb, while SOFT-IO attributes 40 ppb to the aforementioned sources. This means that Southern Hemisphere African AN

emissions are likely underestimated" lines 281-283 of the 1st round revised version by the following (lines 258-261 of the 2nd round revised version):
"The annual CO surface maximum in Central Africa occurs also in April. LT CO is attributed by more than 85% to local AN emissions (Fig. 3 panel 4b and Fig. 4a) when few fires are detected by MODIS (Yamasoe et al. (2015); their Fig.7). The measured CO anomaly reaches +200 ppb (after removing the background from the observed CO), while SOFT-IO attributes 40 to 80 ppb to AN.This indicates that African AN emissions might be underestimated".

**Answer to Referee#3 (blue in the text)**

**General comments**

*Tsivlidou et al. present a valuable study adding to the understanding of CO/O3 distributions in the tropics. I very much appreciate the efforts the authors have gone through in their data analysis. I think the trajectory approach in the SOFT-IO model to disentangle the contribution of wildfire and anthropogenic emissions to the observed signals is sound and sufficiently well documented by Sauvage et al. (2017) to be used here.*

*However, I think that the manuscript is not yet suited for publication but needs to be significantly shortened and made more concise. A great part of the result section is dedicated to the lengthy and extremely detailed description of the observed and modelled profiles. Unfortunately the descriptions are difficult to follow, jumping back and forth between regions/locations, altitude regimes and the different figures, with the structure of the discussion often remaining unclear. Please comprise details and have a more structured discussion clearly presenting the similarities/differences between the regions and altitude regimes. The most relevant figures in the draft are Fig. 9 and Fig. 10. Given their importance and content they are not sufficiently discussed although they actually summarize the results of the preceding lengthy and detailed discussion.*

We thank Referee for his/her objective and constructive comments and suggestions.
We have entirely modified the Result Section 3, which has been shortened by around 30% with regards to the previous version.
We have also modified the conclusions to be more concise and precise and to highlight the main results.

**Specific comments**
*– There are too many abbreviations used. I understand that this is an attempt to keep it short, but to a degree that the text gets close to unreadable. Please use full*

*wording more often. In particular, the letter T is used in the abbreviations for tropics and troposphere which makes it even more difficult to keep things sorted.*
*The way abbreviations are embedded into the text sometimes seems strange with regard to grammar.*

The abbreviations NT and ST have been replaced by Northern Tropics and Southern Tropics.

*– I was confused by the term 'observational site' being used for a moving platform. 'Locations' would be more appropriate in my opinion. The term 'site' is usually used for a (temporarily) fixed installation of measurements equipment in one place.*

Site has been replaced by location.

*– I trust the ACP editorial team will eventually take care of this but the usage of italics in subscripts and units is inconsistent and wrong in many instances.*

Corrected

*– it is confusing to have both an appendix and a supplementary document*

We present the figures from both the appendix and the supplementary material in the supplementary section in the 2nd round revised version of the manuscript.

*– Line 522: Table 2 referenced here is not part of the draft.*

We put Table 2 on page 22 in the 2nd round revised version of the manuscript.

**Abstract**

*In my opinion the abstract is too long and not well organized. The main findings are unclear. I suggest to remove some details and make the abstract more concise. It should become clear what the main conclusions of the analysis are and why these are relevant.*

We rewrote the main conclusions in the abstract.

*– L 11: 'in above 6 km' does not make any sense*

We meant that the $O_3$ maxima over the Asian clusters is observed above 6 km in April, between 6.2 and 11.2 km depending on the location (Figure 8, page 23 of the 1st round revised version of the manuscript).

*– L 13: What do you mean by 'The highest amount of transported CO'. Transport to Asia? Overall?*

We mean the overall highest amount of CO (in ppb) exported from the source regions of Northern and Southern Africa.

**Introduction**

*– L 32: Why is stratospheric influence as the least important process mentioned first?*

Stratospheric influence has been mentioned last.

*– L 37: I understand biomass burning throughout the manuscript refers to wildfires excluding usage of biogenic fuels which is attributed to anthropogenic emissions. This should be made clear here.*
This has been specified.

*– L 59 constraint → constrain*
This has been corrected.

*– L 68: 'offered' – the choice of word reads strange here*
Offered has been replaced by allowed.

**Data and Methods**
*– Line 121f: what do you mean by 'a distance criteria of 300-km'?*
We have rephrased by "data are selected within a 300 km radius circle centered on the airport location".

*– Line 165: what do you mean by 'with bias lower than 10-15 ppb'? Please specify what bias exactly refers to here.*
We have added the following in line 161-162, page 7 in the revised manuscript: "SOFT-IO captures the intensity of CO anomalies with a bias lower than 10-15 ppb **with respect to observed anomalies** for most of the regions and tropospheric layers".r

*– Line 173f: percentages are not absolute differences. The statement does not make any sense to me.*
The sentence containing this word have been deleted in the revised version.

*– Line 177: Why two? Which two backgrounds are referred here? The two mentioned pressure surfaces?*
As mentioned in line 175, page 7 in the original manuscript, the definition of the background CO (BG) is a source of uncertainty in SOFT-IO calculations. To quantify the impact of the BG definition on SOFT-IO contributions, we conducted a sensitivity test using 2 different definitions for the background CO:
- the monthly climatological median CO of a remote area away from polluted regions, between 300 and 185 hPa (UTcruise) for the period 2002 to 2020
- the monthly climatological median CO mixing ratio between 600 and 300 hPa for each location.

We have rephrased lines 175-177, page 7 in the revised manuscript as follows:
" To assess this source of uncertainty, we used  an alternative definition for the

background calculated as the median CO mixing ratio between 600 and 300 hPa for each location. The differences between  this background and the one used in the study are within 2.5-60 ppbv."

**Results**
*All vertical profiles are discussed in terms of absolute altitude but in subsection 3.2. the different altitudes of the inbound/outbound airports are mentioned. In particular for the peaking altitudes presented in Fig. 8, I wonder what the results looked like if altitude differences relative to the local ground level were used.*

The novelty of this study is that we analyse the tropical composition using IAGOS as a unified and consistent dataset for the entire tropics, as a globe. For this reason, we think that it is more appropriate to use the same methodology for the 20 locations taken into consideration for our analysis.

*– Line 234: my reading from the figure would be 62+6=68*
We have replaced the correct number.

*– Line 240 and 248: Throughout the manuscript mixing ratios are discussed, not concentrations. Similar on several instances in the following.*
We have replaced concentrations by mixing ratios.

*– Line 246: the 'observed anomaly' to me is not evident in the figure.*
We have rephrased as follows: "The measured CO anomaly reaches +200 ppb (after removing the background from the observed CO), while SOFT-IO attributes 40 to 80 ppb to AN".

*– Line 255f: no need to cite Adon et al. twice within two lines. Skip first one.*
This is done.

*– Line 258: there is no obvious peak in the CO profile in the figures*
We have rephrased by "Windhoek CO enhancement has the smallest magnitude among the African clusters".

*– L265: space missing between brackets*
Done

*– L271: Reference to Fig 2l does not make sense, Fig. 2l shows CO.*
Correct letter has been put.

*– L445: If the vertical layers are defined on pressure as the vertical coordinate then why are km shown in the figures?*
We show km in the figures in order to make it easier for the reader to understand the altitude of the CO and O3 maxima/minima discussed in the results and conclusions.

***Figures***

*Overall, there are too many figures with too many panels and too small fonts.*

We have increased the resolution and the fonts in Figure 3, 5, 6 and 7 in the revised manuscript. In addition, for clarity reasons we moved the O3 and CO verticals profiles over some clusters in the supplementary material (e.g. Sahel, Guinea Gulf (Fig. 3 panels 1-2 ab of the 1st round revised manuscript); Hyderabad and Thailand Gulf (Fig. 5 panels 1-2 be); and Addis Ababa (Fig. 7 panels 1c, 2c and 5).

*The presentation of observations on an absolute mixing ratio scale and the modelled contributions as ∆CO is difficult to compare. Why are the vertical profiles shown not background corrected?*

One of the main goals of the paper is to document the characteristics and seasonality of the tropical O3 and CO vertical distributions (lines 84-86, page 3 of the 1st round revised version). To do so, we present the absolute mixing ratios of O3 and CO based on IAGOS data which has not been analyzed before over the tropical band. As the paper has been commented as too lengthy with too many figures, we think it is not appropriate to add some more panels.

*– Fig. 2: I suggest to have the panel labels in some lighter colour to make them visible.*

Done

*– Figures 3,5,6, 7 are poor resolution and cannot be zoomed which is essential given the small panels and fonts.*

The size of the panels and fonts in Figures 3,5,6,7 are increased in the revised manuscript.

*– Figure 8: I was wondering about the order in which locations are presented on the x-axis. It would be logical to have the locations by longitude which does not seem to be the case.*

Done

***Summary***

*The Summary largely rephrases the detailed discussion from above. Conclusions are presented alongside but are not worked out well. Please be more precise and separate the shortened descriptive discussion of the observations from the conclusions drawn.*

Summary has been entirely rewritten. To separate the descriptive discussion of the observations from the conclusion we added the line 499, page 25 in the revised manuscript:

"**The results of the study indicate that** the highest O3 and CO mixing ratios are observed over Africa during the fire season in January, with anomalies located in the

LT (75 ppb at 2.5 km for O3 and 800 ppb at 0.3 km for CO over Lagos), and explained mainly by anthropogenic emissions, but with a strong contribution from fires."

*– Line 611 should be 'NT' only*

This is removed from the revised version of the manuscript.

---

## Author Response (AR3)

**Minor revision of Tsivlidou et al. "Tropical tropospheric ozone and carbon monoxide distributions: characteristics, origins and control factors, as seen by IAGOS and IASI".**

**Letter to the Editor**

Dear Editor,

We have addressed the comments of Reviewer #3.
Once again, thank you for your time and consideration.
We are looking forward to hearing from you regarding the final decision on the manuscript.
Sincerely,
Maria Tsivlidou

**Answer to Referee#3 (blue in the text)**

We thank Reviewer #3 for his/her comments and suggestions. Below we provide our answers to the comments and the details of the changes made to the revised manuscript.

*I think the authors have greatly improved the manuscript and managed to enhance its readability and make it more concise and focused.*
*I have a few very minor comments left which I assume can easily be considered. Line numbers refer to version 4 of the manuscript as numbered by ACP.*

*L 11: the term 'clusters' has not been explained yet, so it should be 'location clusters' or similar.*
This has been specified.

*L 163: The definition of anomalies should be moved forward to occur prior to first mentioning of CO anomalies in the text.*
We have moved the definition of the CO anomalies in line 154 of the revised manuscript. We adjusted line 153 as follows:

"SOFT-IO estimates the contribution to  **anomalies in CO mixing ratios i.e.** CO emitted by primary sources during the last 20 days, while it does not calculate the background CO. The background CO can be emitted by primary sources older than 20 days, and by secondary sources such as oxidation of methane and non-methane volatile organic compounds. In our study, CO anomalies **(observed by IAGOS)** are defined as the positive difference between the observed and the background CO mixing ratio. The background CO mixing ratio represents a

reference value, not affected by surface emission or pollution events. For this reason, it is computed as the monthly climatological median CO of a remote area away from polluted regions, in the UTcruise (during the whole study period 2002-2020). Overall in the Tropics, depending of the month and region, a 80 ppb to 100 ppb background has to be added to SOFT-IO for direct comparison with IAGOS observations."

*Figure 8: I understand the authors' argument for discussing the profiles in terms of absolute altitude, but it can easily be misinterpreted. Therefore I recommend to stress this point in the figure caption and the related text describing Figure 8.*
We have modified line 393 as follows:
Figure 8 displays the annual maxima/minima of O3 (a) and CO (b) mixing ratios and their corresponding mean height **in terms of absolute altitude**.

We adjusted the caption of Figure 8 as follows:
"O3 (a) and CO (b) annual maximum (higher bar) and minimum (lower bar) mixing ratio observed over the tropical clusters. The annotated number on top of each bar indicates the **absolute** altitude (in km) of the observed annual maximum/minimum mixing ratio. The colour in the bar indicates the month of the maximum/minimum."